# A connectome and analysis of the adult *Drosophila* central brain

Louis K Scheffer[1†]*, C Shan Xu[1†], Michal Januszewski[2†], Zhiyuan Lu[1,3†], Shin-ya Takemura[1†], Kenneth J Hayworth[1†], Gary B Huang[1†], Kazunori Shinomiya[1†], Jeremy Maitlin-Shepard[4], Stuart Berg[1], Jody Clements[1], Philip M Hubbard[1], William T Katz[1], Lowell Umayam[1], Ting Zhao[1], David Ackerman[1], Tim Blakely[2], John Bogovic[1], Tom Dolafi[1], Dagmar Kainmueller[1‡], Takashi Kawase[1], Khaled A Khairy[1§], Laramie Leavitt[2], Peter H Li[2], Larry Lindsey[2], Nicole Neubarth[1#], Donald J Olbris[1], Hideo Otsuna[1], Eric T Trautman[1], Masayoshi Ito[1,5], Alexander S Bates[6], Jens Goldammer[1,7], Tanya Wolff[1], Robert Svirskas[1], Philipp Schlegel[6], Erika Neace[1], Christopher J Knecht[1], Chelsea X Alvarado[1], Dennis A Bailey[1], Samantha Ballinger[1], Jolanta A Borycz[3], Brandon S Canino[1], Natasha Cheatham[1], Michael Cook[1], Marisa Dreher[1], Octave Duclos[1], Bryon Eubanks[1], Kelli Fairbanks[1], Samantha Finley[1], Nora Forknall[1], Audrey Francis[1], Gary Patrick Hopkins[1], Emily M Joyce[1], SungJin Kim[1], Nicole A Kirk[1], Julie Kovalyak[1], Shirley A Lauchie[1], Alanna Lohff[1], Charli Maldonado[1], Emily A Manley[1], Sari McLin[3], Caroline Mooney[1], Miatta Ndama[1], Omotara Ogundeyi[1], Nneoma Okeoma[1], Christopher Ordish[1], Nicholas Padilla[1], Christopher M Patrick[1], Tyler Paterson[1], Elliott E Phillips[1], Emily M Phillips[1], Neha Rampally[1], Caitlin Ribeiro[1], Madelaine K Robertson[3], Jon Thomson Rymer[1], Sean M Ryan[1], Megan Sammons[1], Anne K Scott[1], Ashley L Scott[1], Aya Shinomiya[1], Claire Smith[1], Kelsey Smith[1], Natalie L Smith[1], Margaret A Sobeski[1], Alia Suleiman[1], Jackie Swift[1], Satoko Takemura[1], Iris Talebi[1], Dorota Tarnogorska[3], Emily Tenshaw[1], Temour Tokhi[1], John J Walsh[1], Tansy Yang[1], Jane Anne Horne[3], Feng Li[1], Ruchi Parekh[1], Patricia K Rivlin[1], Vivek Jayaraman[1], Marta Costa[8], Gregory SXE Jefferis[6,8], Kei Ito[1,5,7], Stephan Saalfeld[1], Reed George[1], Ian A Meinertzhagen[1,3], Gerald M Rubin[1], Harald F Hess[1], Viren Jain[4], Stephen M Plaza[1]*

[1]Janelia Research Campus, Howard Hughes Medical Institute, Ashburn, United States; [2]Google Research, Mountain View, United States; [3]Life Sciences Centre, Dalhousie University, Halifax, Canada; [4]Google Research, Google LLC, Zurich, Switzerland; [5]Institute for Quantitative Biosciences, University of Tokyo, Tokyo, Japan; [6]MRC Laboratory of Molecular Biology, Cambridge, United States; [7]Institute of Zoology, Biocenter Cologne, University of Cologne, Cologne, Germany; [8]Department of Zoology, University of Cambridge, Cambridge, United Kingdom

*For correspondence: schefferl@janelia.hhmi.org (LKS); plazas@janelia.hhmi.org (SMP)

†These authors contributed equally to this work

Present address: ‡Max Delbrueck Centre for Developmental Medicine, Berlin, Germany; §Department of Developmental Neurobiology, St. Jude Children's Research Hospital, Memphis, United States; #Two Six Labs, Arlington, United States

**Abstract** The neural circuits responsible for animal behavior remain largely unknown. We summarize new methods and present the circuitry of a large fraction of the brain of the fruit fly *Drosophila melanogaster*. Improved methods include new procedures to prepare, image, align, segment, find synapses in, and proofread such large data sets. We define cell types, refine computational compartments, and provide an exhaustive atlas of cell examples and types, many of them novel. We provide detailed circuits consisting of neurons and their chemical synapses for most of the central brain. We make the data public and simplify access, reducing the effort needed

**eLife digest** Animal brains of all sizes, from the smallest to the largest, work in broadly similar ways. Studying the brain of any one animal in depth can thus reveal the general principles behind the workings of all brains. The fruit fly Drosophila is a popular choice for such research. With about 100,000 neurons – compared to some 86 billion in humans – the fly brain is small enough to study at the level of individual cells. But it nevertheless supports a range of complex behaviors, including navigation, courtship and learning.

Thanks to decades of research, scientists now have a good understanding of which parts of the fruit fly brain support particular behaviors. But exactly how they do this is often unclear. This is because previous studies showing the connections between cells only covered small areas of the brain. This is like trying to understand a novel when all you can see is a few isolated paragraphs.

To solve this problem, Scheffer, Xu, Januszewski, Lu, Takemura, Hayworth, Huang, Shinomiya et al. prepared the first complete map of the entire central region of the fruit fly brain. The central brain consists of approximately 25,000 neurons and around 20 million connections. To prepare the map – or connectome – the brain was cut into very thin 8nm slices and photographed with an electron microscope. A three-dimensional map of the neurons and connections in the brain was then reconstructed from these images using machine learning algorithms. Finally, Scheffer et al. used the new connectome to obtain further insights into the circuits that support specific fruit fly behaviors.

The central brain connectome is freely available online for anyone to access. When used in combination with existing methods, the map will make it easier to understand how the fly brain works, and how and why it can fail to work correctly. Many of these findings will likely apply to larger brains, including our own. In the long run, studying the fly connectome may therefore lead to a better understanding of the human brain and its disorders. Performing a similar analysis on the brain of a small mammal, by scaling up the methods here, will be a likely next step along this path.

to answer circuit questions, and provide procedures linking the neurons defined by our analysis with genetic reagents. Biologically, we examine distributions of connection strengths, neural motifs on different scales, electrical consequences of compartmentalization, and evidence that maximizing packing density is an important criterion in the evolution of the fly's brain.

## Introduction

The connectome we present is a dense reconstruction of a portion of the central brain (referred to here as the hemibrain) of the fruit fly, *Drosophila melanogaster*, as shown in *Figure 1*. This region was chosen since it contains all the circuits of the central brain (assuming bilateral symmetry), and in particular contains circuits critical to unlocking mysteries involving associative learning in the mushroom body, navigation and sleep in the central complex, and circadian rhythms among clock circuits. The largest dense reconstruction to date, it contains around 25,000 neurons, most of which were rigorously clustered and named, with about 20 million chemical synapses between them, plus portions of many other neurons truncated by the boundary of the data set (details in *Figure 1*). Each neuron is documented at many levels - the detailed voxels that constitute it, a skeleton with segment diameters, its synaptic partners and the location of most of their synapses.

Producing this data set required advances in sample preparation, imaging, image alignment, machine segmentation of cells, synapse detection, data storage, proofreading software, and protocols to arbitrate each decision. A number of new tests for estimating the completeness and accuracy were required and therefore developed, in order to verify the correctness of the connectome.

These data describe whole-brain properties and circuits, as well as contain new methods to classify cell types based on connectivity. Computational compartments are now more carefully defined, we conclusively identify synaptic circuits, and each neuron is annotated by name and putative cell type, making this the first complete census of neuropils, tracts, cells, and connections in this portion of the brain. We compare the statistics and structure of different brain regions, and for the brain as a whole, without the confounds introduced by studying different circuitry in different animals.

All data are publicly available through web interfaces. This includes a browser interface, NeuPrint (*Clements et al., 2020*), designed so that any interested user can query the hemibrain connectome even without specific training. NeuPrint can query the connectivity, partners, connection strengths and morphologies of all specified neurons, thus making identification of upstream and downstream partners both orders of magnitude easier, and significantly more confident, compared to existing genetic methods. In addition, for those who are willing to program, the full data set - the gray scale voxels, the segmentation and proofreading results, skeletons, and graph model of connectivity, are also available through publicly accessible application program interfaces (APIs).

This effort differs from previous EM reconstructions in its social and collaborative aspects. Previous reconstructions were either dense in much smaller EM volumes (such as *Meinertzhagen and O'Neil, 1991*; *Helmstaedter et al., 2013*; *Takemura et al., 2017*) or sparse in larger volumes (such as *Eichler et al., 2017* or *Zheng et al., 2018*). All have concentrated on the reconstruction of specific circuits to answer specific questions. When the same EM volume is used for many such efforts, as has occurred in the *Drosophila* larva and the full adult fly brain, this leads to an overall reconstruction that is the union of many individual efforts (*Saalfeld et al., 2009*). The result is inconsistent

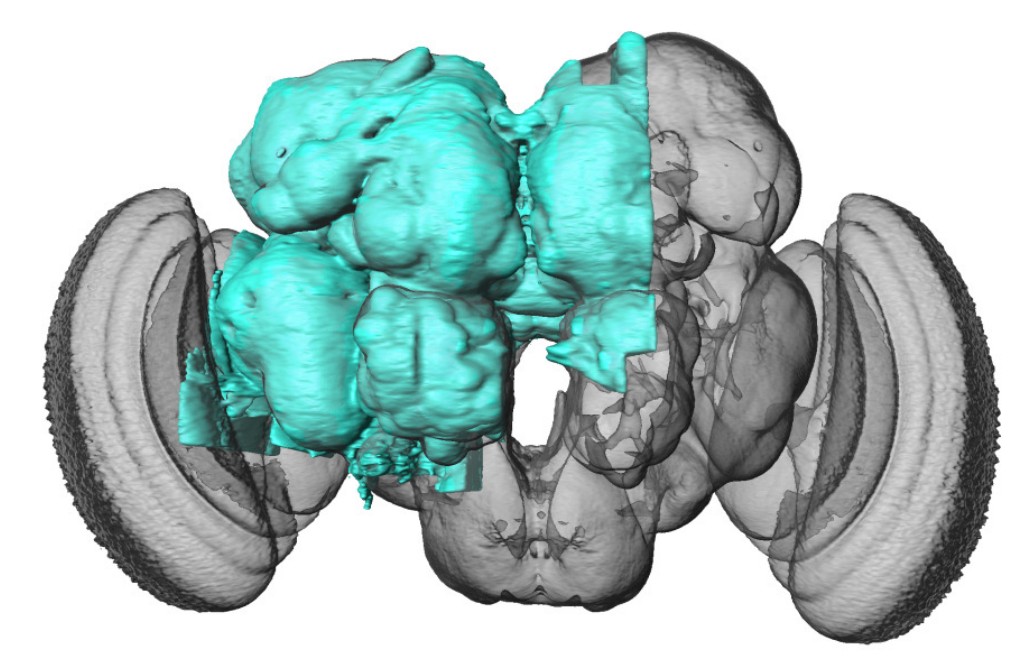

| | |
|---|---|
| Neurons traced, most arbors in volume (uncropped) | 21,734 |
| Neurons traced, large (≥ 1000 connections) but cropped by edge of volume | 4,456 |
| Remaining traced, small (< 1000 connections) and cropped | 71,710 |
| Presynaptic sites (T-Bars) in uncropped/traced/total T-bars | 6M/8.6M/9.5M |
| Postsynaptic densities (PSDs) in uncropped/traced/total | 19M/23M/64M |

**Figure 1.** The hemibrain and some basic statistics. The highlighted area shows the portion of the central brain that was imaged and reconstructed, superimposed on a grayscale representation of the entire *Drosophila* brain. For the table, a neuron is traced if all its main branches within the volume are reconstructed. A neuron is considered uncropped if most arbors (though perhaps not the soma) are contained in the volume. Others are considered cropped. Note: (1) our definition of cropped is somewhat subjective; (2) the usefulness of a cropped neuron depends on the application; and (3) some small fragments are known to be distinct neurons. For simplicity, we will often state that the hemibrain contains ≈25K neurons.

coverage of the brain, with some regions well reconstructed and others missing entirely. In contrast, here we have analyzed the entire volume, not just the subsets of interest to specific groups of researchers with the expertise to tackle EM reconstruction. We are making these data available without restriction, with only the requirement to cite the source. This allows the benefits of known circuits and connectivity to accrue to the field as a whole, a much larger audience than those with expertise in EM reconstruction. This is analogous to progress in genomics, which transitioned from individual groups studying subsets of genes, to publicly available genomes that can be queried for information about genes of choice (*Altschul et al., 1990*).

One major benefit to this effort is to facilitate research into the circuits of the fly's brain. A common question among researchers, for example, is the identity of upstream and downstream (respectively input and output) partners of specific neurons. Previously, this could only be addressed by genetic trans-synaptic labeling, such as trans-Tango (*Talay et al., 2017*), or by sparse tracing in previously imaged EM volumes (*Zheng et al., 2018*). However, the genetic methods may give false positives and negatives, and both alternatives require specialized expertise and are time consuming, often taking months of effort. Now, for any circuits contained in our volume, a researcher can obtain the same answers in seconds by querying a publicly available database.

Another major benefit of dense reconstruction is its exhaustive nature. Genetic methods such as stochastic labeling may miss certain cell types, and counts of cells of a given type are dependent on expression levels, which are always uncertain. Previous dense reconstructions have demonstrated that existing catalogs of cell types are incomplete, even in well-covered regions (*Takemura et al., 2017*). In our hemibrain sample, we have identified all the cells within the reconstructed volume, thus providing a complete and unbiased census of all cell types in the fly's central brain (at least in this single female), and a precise count of the cells of each type.

Another scientific benefit lies in an analysis without the uncertainty of pooling data obtained from different animals. The detailed circuitry of the fly's brain is known to depend on nutritional history, age, and circadian rhythm. Here, these factors are held constant, as are the experimental methods, facilitating comparison between different fly brain regions in this single animal. Evaluating stereotypy across animals will of course eventually require additional connectomes.

Previous reconstructions of compartmentalized brains have concentrated on particular regions and circuits. The mammalian retina (*Helmstaedter et al., 2013*) and cortex (*Kasthuri et al., 2015*), and insect mushroom bodies (*Eichler et al., 2017*; *Takemura et al., 2017*) and optic lobes (*Takemura et al., 2015*) have all been popular targets. Additional studies have examined circuits that cross regions, such as those for sensory integration (*Ohyama et al., 2015*) or motion vision (*Shinomiya et al., 2019*).

So far lacking are systematic studies of the statistical properties of computational compartments and their connections. Neural circuit motifs have been studied (*Song et al., 2005*), but only those restricted to small motifs and at most a few cell types, usually in a single portion of the brain. Many of these results are in mammals, leading to questions of whether they also apply to invertebrates, and whether they extend to other regions of the brain. While there have been efforts to build reduced, but still accurate, electrical models of neurons (*Marasco et al., 2012*), none of these to our knowledge have used the compartment structure of the brain.

## What is included

*Table 1* shows the hierarchy of the named brain regions that are included in the hemibrain. *Table 2* shows the primary regions that are at least 50% included in the hemibrain sample, their approximate size, and their completion percentage. Our names for brain regions follow the conventions of *Ito et al., 2014* with the addition of '(L)' or '(R)' to indicate whether the region (most of which occur on both sides of the fly) has its cell bodies in the left or right, respectively. The mushroom body (*Tanaka et al., 2008*; *Aso et al., 2014*) and central complex (*Wolff et al., 2015*; *Wolff and Rubin, 2018*) are further divided into finer compartments.

*Appendix 1—table 6* provide the list of identified neuron types and their naming schemes. These include newly identified sensory inputs and motor outputs.

The nature of the proofreading process allows us to improve the data even after their initial publication. Our initial data release was version v1.0 (*Xu et al., 2020c*). Version v1.1 is now available, including improvements such as better accuracy, more consistent cell naming and typing, and inclusion of anatomical names for central complex neurons. The old version(s) remain online and

**Table 1.** Brain regions contained and defined in the hemibrain, following the naming conventions of *Ito et al., 2014* with the addition of (R) and (L) to specify the side of the soma for that region.

*Italics* indicate master regions not explicitly defined in the hemibrain. Region LA is not included in the volume. The regions are hierarchical, with the more indented regions forming subsets of the less indented. The only exceptions are dACA, lACA, and vACA which are considered part of the mushroom body but are not contained in the master region MB.

| OL(R) | Optic lobe | CX | Central complex | LH(R) | Lateral horn |
|---|---|---|---|---|---|
| *LA* | lamina | FB | Fan-shaped body | | |
| ME(R) | Medula | FBl1 | Fan-shaped body layer 1 | *SNP(R)/(L)* | Superior neuropils |
| AME(R) | Accessory medulla | FBl2 | Fan-shaped body layer 2 | SLP(R) | Superior lateral protocerebrum |
| LO(R) | Lobula | FBl3 | Fan-shaped body layer 4 | SIP(R)/(L) | Superior intermediate protocerebrum |
| LOP(R) | Lobula plate | FBl4 | Fan-shaped body layer 4 | SMP(R)(L) | Superior medial protocerebrum |
| | | FBl5 | Fan-shaped body layer 5 | | |
| MB(R)/(L) | Mushroom body | FBl6 | Fan-shaped body layer 6 | *INP* | Inferior neuropils |
| CA(R)/(L) | Calyx | FBl7 | Fan-shaped body layer 7 | CRE(R)/(L) | Crepine |
| dACA(R) | Dorsal accessory calyx | FBl8 | Fan-shaped body layer 8 | RUB(R)/(L) | Rubu |
| lACA(R) | Lateral accessory calyx | FBl9 | Fan-shaped body layer 9 | ROB(R) | Round body |
| vACA(R) | Ventral accessory calyx | EB | Ellipsoid body | SCL(R)/(L) | Superior clamp |
| PED(R) | Pedunculus | EBr1 | Ellipsoid body zone r1 | ICL(R)/(L) | Inferior clamp |
| a'L(R)/(L) | Alpha prime lobe | EBr2r4 | Ellipsoid body zone r2r4 | IB | Inferior bridge |
| a'1(R) | Alpha prime lobe compartment 1 | EBr3am | Ellipsoid body zone r3am | ATL(R)/(L) | Antler |
| a'2(R) | Alpha prime lobe compartment 2 | EBr3d | Ellipsoid body zone r3d | | |
| a'3(R) | Alpha prime lobe compartment 3 | EBr3pw | Ellipsoid body zone r3pw | AL(R)/(L) | Antennal lobe |
| aL(R)/(L) | Alpha lobe | EBr5 | Ellipsoid body zone r5 | | |
| a1(R) | Alpha lobe compartment 1 | EBr6 | Ellipsoid body zone r6 | *VMNP* | Ventromedial neuropils |
| a2(R) | Alpha lobe compartment 2 | AB(R)/(L) | Asymmetrical body | VES(R)/(L) | Vest |
| a3(R) | Alpha lobe compartment 3 | PB | Protocerebral bridge | EPA(R)/(L) | Epaulette |
| gL(R)/(L) | Gamma lobe | PB(R1) | PB glomerulus R1 | GOR(R)/(L) | Gorget |
| g1(R) | Gamma lobe compartment 1 | PB(R2) | PB glomerulus R2 | SPS(R)/(L) | Superior posterior slope |
| g2(R) | Gamma lobe compartment 2 | PB(R3) | PB glomerulus R3 | IPS(R)/(L) | Inferior posterior slope |
| g3(R) | Gamma lobe compartment 3 | PB(R4) | PB glomerulus R4 | | |
| g4(R) | Gamma lobe compartment 4 | PB(R5) | PB glomerulus R5 | *PENP* | Pariesophageal neuropils |
| g5(R) | Gamma lobe compartment 5 | PB(R6) | PB glomerulus R6 | SAD | Saddle |
| b'L(R)/(L) | Beta prime lobe | PB(R7) | PB glomerulus R7 | AMMC | Antennal mechanosensory and motor center |
| b'1(R) | Beta prime lobe compartment 1 | PB(R8) | PB glomerulus R8 | FLA(R) | Flange |
| b'2(R) | Beta prime lobe compartment 2 | PB(R9) | PB glomerulus R9 | CAN(R) | Cantle |
| bL(R)/(L) | Beta lobe | PB(L1) | PB glomerulus L1 | PRW | prow |
| b1(R) | Beta lobe compartment 1 | PB(L2) | PB glomerulus L2 | | |
| b2(R) | Beta lobe compartment 2 | PB(L3) | PB glomerulus L3 | GNG | Gnathal ganglia |
| | | PB(L4) | PB glomerulus L4 | | |
| *LX(R)/(L)* | Lateral complex | PB(L5) | PB glomerulus L5 | Major Fiber bundles | |
| BU(R)/(L) | Bulb | PB(L6) | PB glomerulus L6 | AOT(R) | Anterior optic tract |
| LAL(R)/(L) | Lateral accessory lobe | PB(L7) | PB glomerulus L7 | GC | Great commissure |
| GA(R) | Gall | PB(L8) | PB glomerulus L8 | GF(R) | Giant Fiber (single neuron) |
| | | PB(L9) | PB glomerulus L9 | mALT(R)/(L) | Medial antennal lobe tract |
| *VLNP(R)* | Ventrolateral neuropils | NO | Noduli | POC | Posterior optic commissure |
| AOTU(R) | Anterior optic tubercle | NO1(R)/(L) | Nodulus 1 | | |
| AVLP(R) | Anterior ventrolateral protocerebrum | NO2(R)/(L) | Nodulus 2 | | |

*Table 1 continued on next page*

| | | | |
|---|---|---|---|
| PVLP(R) | Posterior ventrolateral protocerebrum | NO3(R)/(L) | Nodulus 3 |
| PLP(R) | Posterior lateral cerebrum | | |
| WED(R) | Wedge | | |

available, to allow reproducibility of older analyses, but we strongly recommend all new analyses use the latest version. The analyses in this article, and in the corresponding articles on the mushroom body and central complex, are based on version v1.1, unless otherwise noted.

## What is not included

This research focused on the neurons of the brain and the chemical synapses between them. Every step in our process, from staining and sample preparation through segmentation and proofreading, has been optimized with this goal in mind. While neurons and their chemical synapses are critical to brain operation, they are far from the full story. Other contributors, known to be important, could not be included in our study, largely for technical reasons. Among these are gap junctions, glia, and structures internal to the cell such as mitochondria. Gap junctions, or electrical connections between neurons, are difficult to reliably detect by FIB-SEM under the best of circumstances and not detectable at the low (for EM) resolution needed to complete this study in a reasonable amount of time.

**Table 2.** Regions with ≥50% included in the hemibrain, sorted by completion percentage.
The approximate percentage of the region included in the hemibrain volume is shown as '%inV'. 'T-bars' gives a rough estimate of the size of the region. 'comp%' is the fraction of the post-synaptic densities (PSDs) contained in the brain region for which both the PSD and the corresponding T-bar are in neurons marked 'Traced'.

| Name | %inV | T-bars | comp% | Name | %inV | T-bars | comp% |
|---|---|---|---|---|---|---|---|
| PED(R) | 100% | 54805 | 85% | aL(R) | 100% | 95375 | 84% |
| b'L(R) | 100% | 67695 | 83% | bL(R) | 100% | 71112 | 83% |
| gL(R) | 100% | 176785 | 83% | a'L(R) | 100% | 39091 | 82% |
| EB | 100% | 164286 | 81% | bL(L) | 56% | 58799 | 81% |
| NO | 100% | 36722 | 79% | b'L(L) | 88% | 57802 | 78% |
| gL(L) | 55% | 133256 | 76% | CA(R) | 100% | 69517 | 73% |
| AB(R) | 100% | 2734 | 65% | aL(L) | 51% | 44803 | 62% |
| FB | 100% | 451031 | 62% | AL(R) | 83% | 501004 | 59% |
| AB(L) | 100% | 572 | 57% | PB | 100% | 46557 | 55% |
| AME(R) | 100% | 6045 | 51% | BU(R) | 100% | 9385 | 46% |
| CRE(R) | 100% | 137946 | 40% | AOTU(R) | 100% | 92578 | 38% |
| LAL(R) | 100% | 234388 | 38% | SMP(R) | 100% | 510937 | 34% |
| PVLP(R) | 100% | 475219 | 30% | ATL(R) | 100% | 25472 | 29% |
| SPS(R) | 100% | 253818 | 29% | ATL(L) | 100% | 28153 | 29% |
| VES(R) | 84% | 157168 | 29% | IB | 100% | 200447 | 28% |
| CRE(L) | 90% | 132656 | 28% | SIP(R) | 100% | 187493 | 26% |
| BU(L) | 52% | 7014 | 26% | GOR(R) | 100% | 27140 | 26% |
| WED(R) | 100% | 232898 | 25% | SMP(L) | 100% | 460784 | 26% |
| EPA(R) | 100% | 31438 | 26% | PLP(R) | 100% | 429949 | 26% |
| AVLP(R) | 100% | 630538 | 23% | ICL(R) | 100% | 202549 | 23% |
| SLP(R) | 100% | 487795 | 23% | LO(R) | 64% | 855251 | 22% |
| SCL(R) | 100% | 189569 | 22% | GOR(L) | 60% | 19558 | 21% |
| LH(R) | 100% | 231662 | 19% | CAN(R) | 68% | 6512 | 16% |

Their contribution to the connectome will need to be established through other means - see the section on future research. Glial cells were difficult to segment, due to both staining differences and convoluted morphologies. We identified the volumes where they exist (a glia 'mask', which allows these regions to be color-coded when viewed in NeuroGlancer) but did not separate them into cells. Structures internal to the neurons, except for synapses, are not considered here even though many are visible in our EM preparation. The most obvious example is mitochondria. Again, we have identified many of them so we could evaluate their effect on segmentation, but they are not included in our connectome. Finally, autapses (synapses from a neuron onto itself) are known to exist in *Drosophila*, but are sufficiently rare that they fall well below the rate of false positives in our automated synapse detection. Therefore most of the putative autapses are false positives, and we do not include them in our connectivity data.

## Differences from connectomes of vertebrates

Most accounts of neurobiology define the operation of the mammalian nervous system with, at most, only passing reference to invertebrate brains. Fly (or other insect) nervous systems differ from those of vertebrates in several aspects (*Meinertzhagen, 2016b*). Some main differences include:

- Most synapses are polyadic. Each synapse structure comprises a single presynaptic release site and, adjacent to this, several neurites expressing neurotransmitter receptors. An element, T-shaped and typically called a T-bar in flies, marks the site of transmitter release into the cleft between cells. This site typically abuts the neurites of several other cells, where a postsynaptic density (PSD) marks the receptor location.
- Most neurites are neither purely axonic nor dendritic, but have both pre- and postsynaptic partners, a feature that may be more prominent in mammalian brains than recognized (*Morgan and Lichtman, 2020*). Within a single brain region, however, neurites are frequently predominantly dendritic (postsynaptic) or axonic (presynaptic).
- Unlike some synapses in mammals, EM imagery (at least as we have acquired and analyzed it here) fails to reveal obvious information about whether a synapse is excitatory or inhibitory.
- The soma or cell body of each fly neuron resides in a rind (the cell body layer) on the periphery of the brain, mostly disjoint from the main neurites innervating the internal neuropil. As a result, unlike vertebrate neurons, no synapses form directly on the soma. The neuronal process between the soma and the first branch point is called the cell body fiber (CBF), which is likewise not involved in the synaptic transmission of information.
- Synapse sizes are much more uniform than those of mammals. Stronger connections are formed by increasing the number of synapses in parallel, not by forming larger synapses, as in vertebrates. In this paper, we will refer to the 'strength' of a connection as the synapse count, even though we acknowledge that we lack information on the relative activity and strength of the synapses, and thus a true measure of their coupling strength.
- The brain is small, about 250 μm per side, and has roughly the same size as the dendritic arbor of a single pyramidal neuron in the mammalian cortex.
- Axons of fly neurons are not myelinated.
- Some fly neurons rely on graded transmission (as opposed to spiking), without obvious anatomical distinction. Some neurons even switch between graded and spiking operation (*Pimentel et al., 2016*).

## Connectome reconstruction

Producing a connectome comprising reconstructed neurons and the chemical synapses between them required several steps. The first step, preparing a fly brain and imaging half of its center, produced a dataset consisting of 26 teravoxels of data, each with 8 bits of grayscale information. We applied numerous machine-learning algorithms and over 50 person-years of proofreading effort over ≈2 calendar years to extract a variety of more compact and useful representations, such as neuron skeletons, synapse locations, and connectivity graphs. These are both more useful and much smaller than the raw grayscale data. For example, the connectivity could be reasonably summarized by a graph with ≈25,000 nodes and ≈3 million edges. Even when the connections were assigned to different brain regions, such a graph took only 26 MB, still large but roughly a million fold reduction in data size.

Many of the supporting methods for this reconstruction have been recently published. Here, we briefly survey each major area, with more details reported in the companion papers. Major advances include:

- New methods to fix and stain the sample, preparing a whole fly brain with well-preserved sub-cellular detail particularly suitable for machine analysis.
- Methods that have enabled us to collect the largest EM dataset yet using Focused Ion Beam Scanning Electron Microscopy (FIB-SEM), resulting in isotropic data with few artifacts, features that significantly sped up reconstruction.
- A coarse-to-fine, automated flood-filling network segmentation pipeline applied to image data normalized with cycle-consistent generative adversarial networks, and an aggressive automated agglomeration regime enabled by advances in proofreading.
- A new hybrid synapse prediction method, using two differing underlying techniques, for accurate synapse prediction throughout the volume.
- New top-down proofreading methods that utilize visualization and machine learning to achieve orders of magnitude faster reconstruction compared with previous approaches in the fly's brain.

Each of these is explained in more detail in the following sections and, where necessary, in the appendix. The companion papers are 'The connectome of the *Drosophila melanogaster* mushroom body: implications for function' (*Li et al., 2020*) and 'A complete synaptic-resolution connectome of the *Drosophila melanogaster* central complex' by Jayaraman, et al.

## Image stack collection

The first steps, fixing and staining the specimen, have been accomplished taking advantage of three new developments. These improved methods allow us to fix and stain a full fly's brain but nevertheless recover neurons as round profiles with darkly stained synapses, suitable for machine segmentation and automatic synapse detection. We started with a 5-day-old female of wild-type Canton S strain G1 x w$^{1118}$, raised on a 12 hr day/night cycle. 1.5 hr after lights-on, we used a custom-made jig to microdissect the brain, which was then fixed and embedded in Epon, an epoxy resin. We then enhanced the electron contrast by staining with heavy metals, and progressively lowered the temperature during dehydration of the sample. Collectively, these methods optimize morphological preservation, allow full-brain preparation without distortion (unlike fast freezing methods), and provide increased staining intensity that speeds the rate of FIB-SEM imaging (*Lu et al., 2019*).

The hemibrain sample is roughly 250 × 250 × 250 µm, larger than we can FIB-SEM without introducing milling artifacts. Therefore, we subdivided our epoxy-embedded samples into 20-µm-thick slabs, both to avoid artifacts and allow imaging in parallel (each slab can be imaged in a different FIB machine) for increased throughput. To be effective, the cut surfaces of the slabs must be smooth at the ultrastructural level and have only minimal material loss. Specifically, for connectomic research, all long-distance processes must remain traceable across sequential slabs. We used an improved version of our previously published 'hot-knife' ultrathick sectioning procedure (*Hayworth et al., 2015*) which uses a heated, oil-lubricated diamond knife, to section the *Drosophila* brain into 37 sagittal slabs of 20 µm thickness with an estimated material loss between consecutive slabs of only ~30 nm – sufficiently small to allow tracing of long-distance neurites. Each slab was re-embedded, mounted, and trimmed, then examined in 3D with X-ray tomography to check for sample quality and establish a scale factor for Z-axis cutting by FIB. The resulting slabs were FIB-SEM imaged separately (often in parallel, in different FIB-SEM machines), and the resulting volume datasets were stitched together computationally.

Connectome studies come with clearly defined resolution requirements – the finest neurites must be traceable by humans and should be reliably segmented by automated algorithms (*Januszewski et al., 2018*). In *Drosophila*, the very finest neural processes are usually 50 nm but can be as little as 15 nm (*Meinertzhagen, 2016a*). This fundamental biological dimension determines the minimum isotropic resolution requirements for tracing neural circuits. To meet the demand for high isotropic resolution and large volume imaging, we chose the FIB-SEM imaging platform, which offers high isotropic resolution (<10 nm in x, y, and z), minimal artifacts, and robust image alignment. The high-resolution and isotropic dataset possible with FIB-SEM has substantially expedited the *Drosophila* connectome pipeline. Compared to serial-section imaging, with its sectioning artifacts and inferior Z-axis resolution, FIB-SEM offers high-quality image alignment, a smaller number of artifacts,

and isotropic resolution. This allows higher quality automated segmentation and makes manual proofreading and correction easier and faster.

At the beginning, deficiencies in imaging speed and system reliability of any commercial FIB-SEM system capped the maximum possible image volume to less than 0.01% of a full fly brain, problems that persist even now. To remedy them, we redesigned the entire control system, improved the imaging speed more than 10x, and created innovative solutions addressing all known failure modes, which thereby expanded the practical imaging volume of conventional FIB-SEM by more than four orders of magnitude from $10^3 \mu m^3$ to $3 \cdot 10^7 \ \mu m^3$, while maintaining an isotropic resolution of $8 \times 8 \times 8$ nm voxels (*Xu et al., 2017*; *Xu et al., 2020a*). In order to overcome the aberration of a large field of view (up to 300 µm wide), we developed a novel tiling approach without sample stage movement, in which the imaging parameters of each tile are individually optimized through an in-line auto focus routine without overhead (*Xu et al., 2020b*). After numerous improvements, we have transformed the conventional FIB-SEM from a laboratory tool that is unreliable for more than a few days of imaging to a robust volume EM platform with effective long-term reliability, able to perform years of continuous imaging without defects in the final image stack. Imaging time, rather than FIB-SEM reliability, is now the main impediment to obtaining even larger volumes.

In our study here, the *Drosophila* 'hemibrain', 13 consecutive hot-knifed slabs were imaged using two customized enhanced FIB-SEM systems, in which an FEI Magnum FIB column was mounted at 90° upon a Zeiss Merlin SEM. After data collection, streaking artifacts generated by secondary electrons along the FIB milling direction were computationally removed using a mask in the frequency domain. The image stacks were then aligned using a customized version of the software platform developed for serial section transmission electron microscopy (*Zheng et al., 2018*; *Khairy et al., 2018*), followed by binning along the z-axis to form the final $8 \times 8 \times 8$ nm$^3$ voxel datasets. Milling thickness variations in the aligned series were compensated using a modified version of the method described by *Hanslovsky et al., 2017*, with the absolute scale calibrated by reference to the MicroCT images.

The 20 µm slabs generated by the hot-knife sectioning were re-embedded in larger plastic tabs prior to FIB-SEM imaging. To correct for the warping of the slab that can occur in this process, methods adapted from Kainmueller (*Kainmueller et al., 2008*) were used to find the tissue-plastic interface and flatten each slab's image stack.

The series of flattened slabs was then stitched using a custom method for large-scale deformable registration to account for deformations introduced during sectioning, imaging, embedding, and alignment (Saalfeld et al. in prep). These volumes were then contrast adjusted using slice-wise contrast limited adaptive histogram equalization (CLAHE) (*Pizer et al., 1987*), and converted into a versioned database (Distributed, Versioned, Image-oriented Database, or DVID) (*Katz and Plaza, 2019*), which formed the raw data for the reconstruction, as illustrated in *Figure 2*.

## Automated segmentation

Computational reconstruction of the image data was performed using flood-filling networks (FFNs) trained on roughly five billion voxels of volumetric ground truth contained in two tabs of the hemibrain dataset (*Januszewski et al., 2018*). Initially, the FFNs generalized poorly to other tabs of the hemibrain, whose image content had different appearances. Therefore, we adjusted the image content to be more uniform using cycle-consistent generative adversarial networks (CycleGANs) (*Zhu et al., 2017*). Specifically, 'generator' networks were trained to alter image content such that a second 'discriminator' network was unable to distinguish between image patches sampled from, for example, a tab that contained volumetric training data versus a tab that did not. A cycle-consistency constraint was used to ensure that the image transformations preserved ultrastructural detail. The improvement is illustrated in *Figure 3*. Overall, this allowed us to use the training data from just two slabs, as opposed to needing training data for each slab.

FFNs were applied to the CycleGAN-normalized data in a coarse-to-fine manner at $32 \times 32 \times 32$ nm$^3$ and $16 \times 16 \times 16$ nm$^3$, and to the CLAHE-normalized data at the native $8 \times 8 \times 8$ nm$^3$ resolution, in order to generate a base segmentation that was largely over-segmented. We then agglomerated the base segmentation, also using FFNs. We aggressively agglomerated segments despite introducing a substantial number of erroneous mergers. This differs from previous algorithms, which studiously avoided merge errors since they were so difficult to fix. Here, advances in proofreading

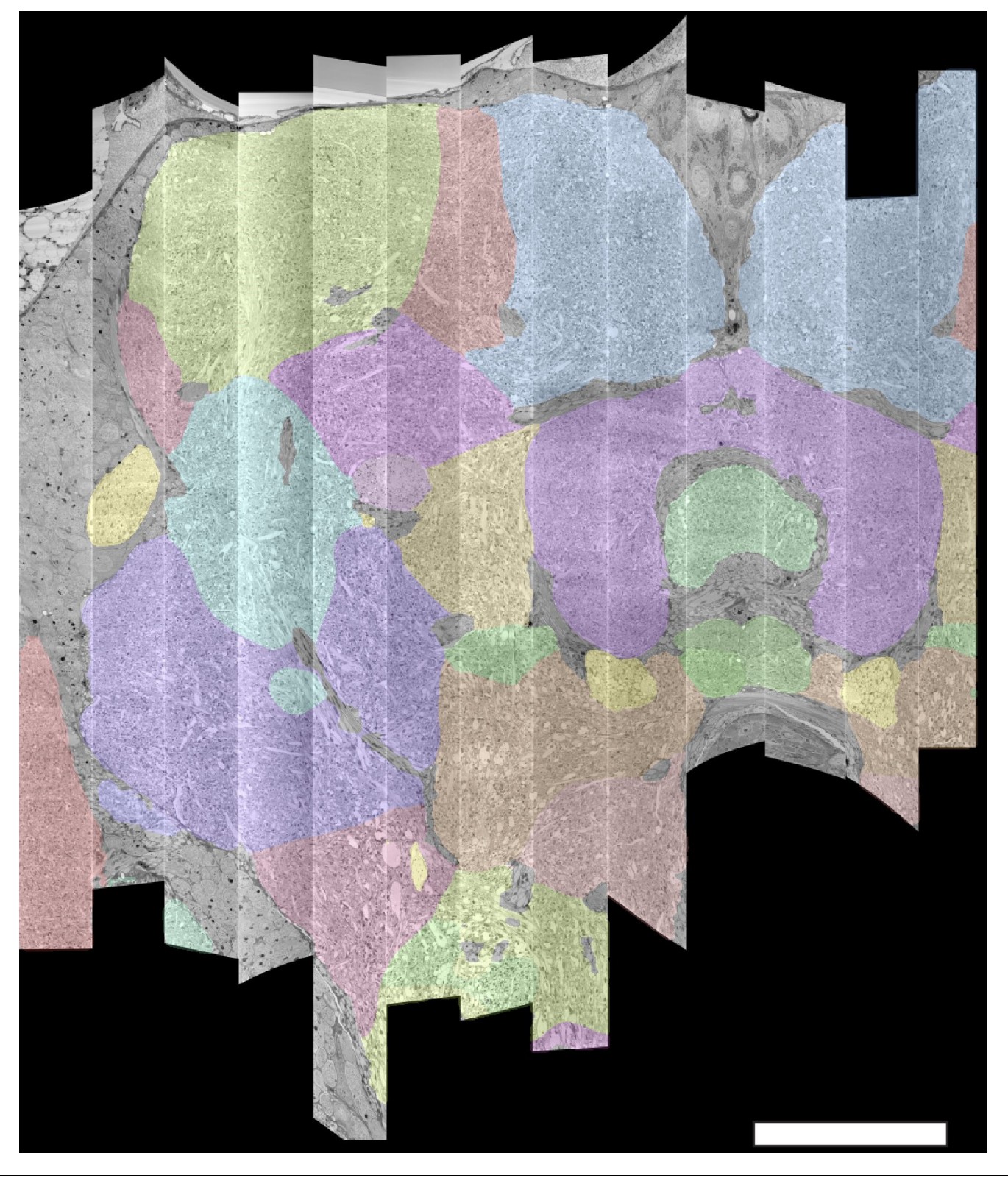

**Figure 2.** The 13 slabs of the hemibrain, each flattened and co-aligned. A vertical section at the level of the fan-shaped body is shown. Colors are arbitrary and added to the monochrome data to show brain regions, as defined below. Scale bar 50 μm.

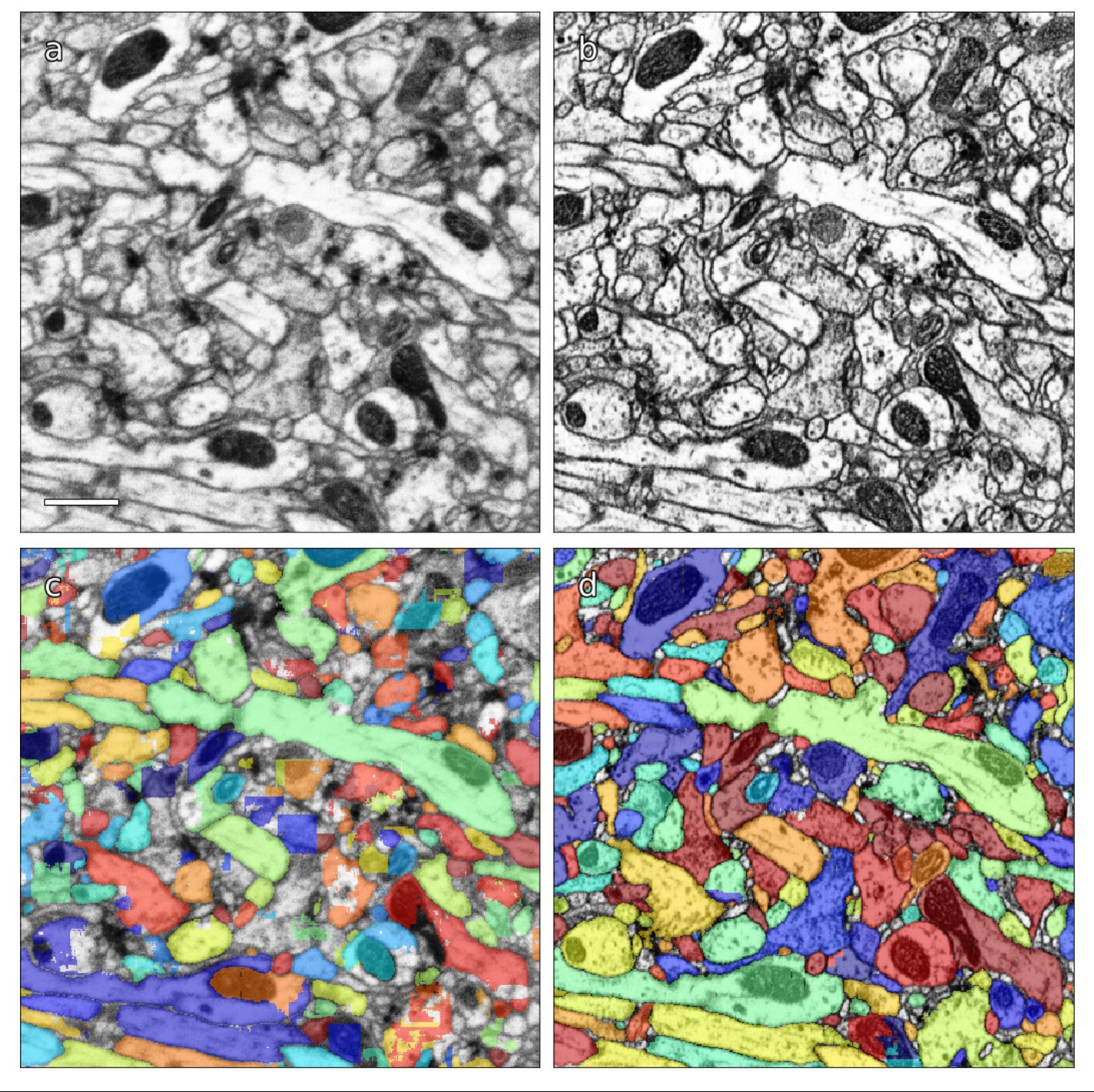

**Figure 3.** Examples of results of CycleGAN processing. (**a**) Original EM data from tab 34 at a resolution of 16 nm / resolution, (**b**) EM data after CycleGAN processing, (**c–d**) FFN segmentation results with the 16 nm model applied to original and processed data, respectively. Scale bar in (**a**) represents 1 µm.

methodology described later in this report enabled efficient detection and correction of such mergers.

We evaluated the accuracy of the FFN segmentation of the hemibrain using metrics for expected run length (ERL) and false merge rate (*Januszewski et al., 2018*). The base segmentation (i.e. the automated reconstruction prior to agglomeration) achieved an ERL of 163 µm with a false merge

rate of 0.25%. After (automated) agglomeration, run length increased to 585 μm but with a false merge rate of 27.6% (i.e. nearly 30% of the path length was contained in segments with at least one merge error). We also evaluated a subset of neurons in the volume, ~500 olfactory PNs and mushroom body KCs chosen to roughly match the evaluation performed in *Li et al., 2019* which yielded an ERL of 825 μm at a 15.9% false merge rate.

## Synapse prediction

Accurate synapse identification is central to our analysis, given that synapses form both a critical component of a connectome and are required for prioritizing and guiding the proofreading effort. Synapses in *Drosophila* are typically polyadic, with a single presynaptic site (a T-bar) contacted by multiple receiving dendrites (most with PSDs) as shown in *Figure 4A*. Initial synapse prediction revealed that there are over 9 million T-bars and 60 million PSDs in the hemibrain. Manually validating each one, assuming a rate of 1000 connections annotated per trained person, per day, would have taken more than 230 working years. Given this infeasibility, we developed machine learning approaches to predict synapses as detailed below. The results of our prediction are shown in *Figure 4B*, where the predicted synapse sites clearly delineate many of the fly brain regions.

Given the size of the hemibrain image volume, a major challenge from a machine learning perspective is the range of varying image statistics across the volume. In particular, model performance can quickly degrade in regions of the data set with statistics that are not well-captured by the training set (*Buhmann et al., 2019*).

To address this challenge, we took an iterative approach to synapse prediction, interleaving model re-training with manual proofreading, all based on previously reported methods (*Huang et al., 2018*). Initial prediction, followed by proofreading, revealed a number of false positive predictions from structures such as dense core vesicles which were not well-represented in the original training set. A second filtering network was trained on regions causing such false positives,

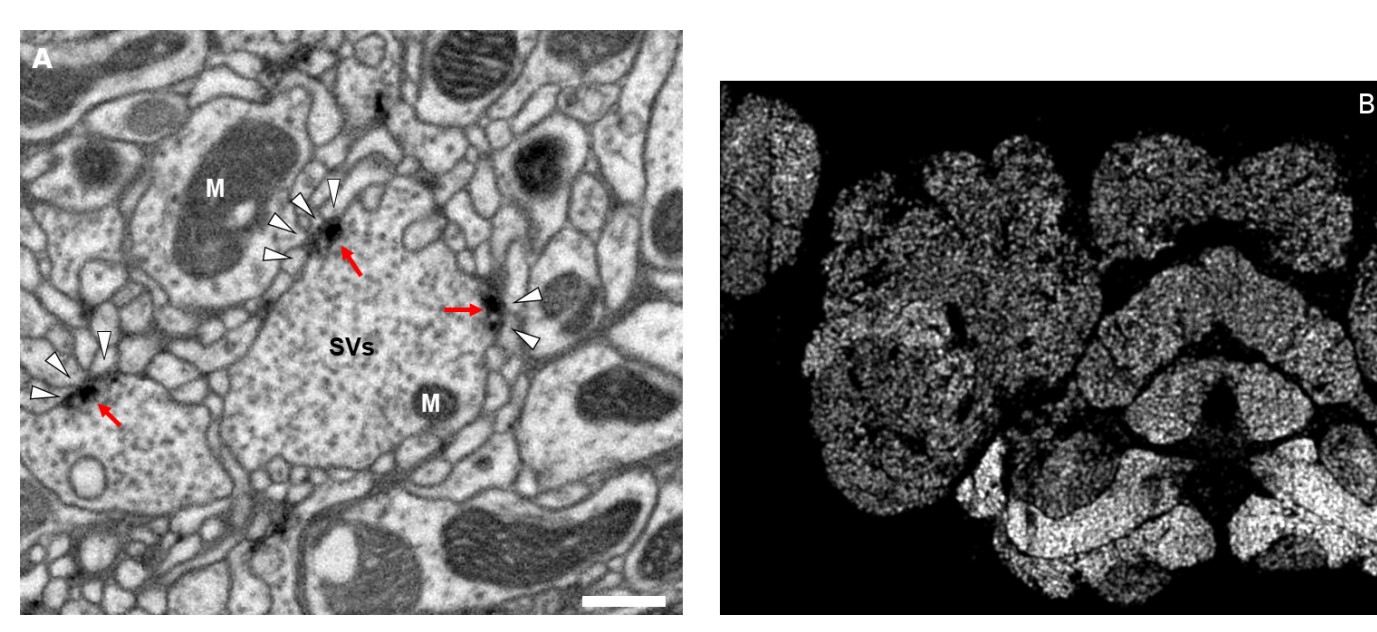

**Figure 4.** Well-preserved membranes, darkly stained synapses, and smooth round neurite profiles are characteristics of the hemibrain sample. Panel (**A**) shows polyadic synapses, with a red arrow indicating the presynaptic T-bar, and white triangles pointing to the PSDs. We identified in total 64 million PSDs and 9.5 million T-bars in the hemibrain volume (*Figure 1*). Thus the average number of PSDs per T-bar in our sample is 6.7. Mitochondria ('M'), synaptic vesicles ('SV'), and the scale bar (0.5 μm) are shown. Panel (**B**) shows a horizontal cross section through a point cloud of all detected synapses. This EM point cloud defines many of the compartments in the fly's brain, much like an optical image obtained using antibody nc82 (an antibody against Bruchpilot, a component protein of T-bars) to stain synapses. This point cloud is used to generate the transformation from our sample to the standard *Drosophila* brain.

and used to prune back the original set of predictions. We denote this pruned output as the 'initial' set of synapse predictions.

Based on this initial set, we began collecting human-annotated dense ground-truth cubes throughout the various brain regions of the hemibrain, to assess variation in classifier performance by brain region. From these cubes, we determined that although many regions had acceptable precision, there were some regions in which recall was lower than desired. Consequently, a subset of cubes available at that time was used to train a new classifier focused on addressing recall in the problematic regions. This new classifier was used in an incremental (cascaded) fashion, primarily by adding additional predictions to the existing initial set. This gave better performance than complete replacement using only the new classifier, with the resulting predictions able to improve recall while largely maintaining precision.

As an independent check on synapse quality, we also trained a separate classifier (*Buhmann et al., 2019*), using a modified version of the 'synful' software package. Both synapse predictors give a confidence value associated with each synapse, a measure of how firmly the classifier believes the prediction to be a true synapse. We found that we were able to improve recall by taking the union of the two predictor's most confident synapses, and similarly improve precision by removing synapses that were low confidence in both predictions. *Figure 5A and B* show the results, illustrating the precision and recall obtained in each brain region.

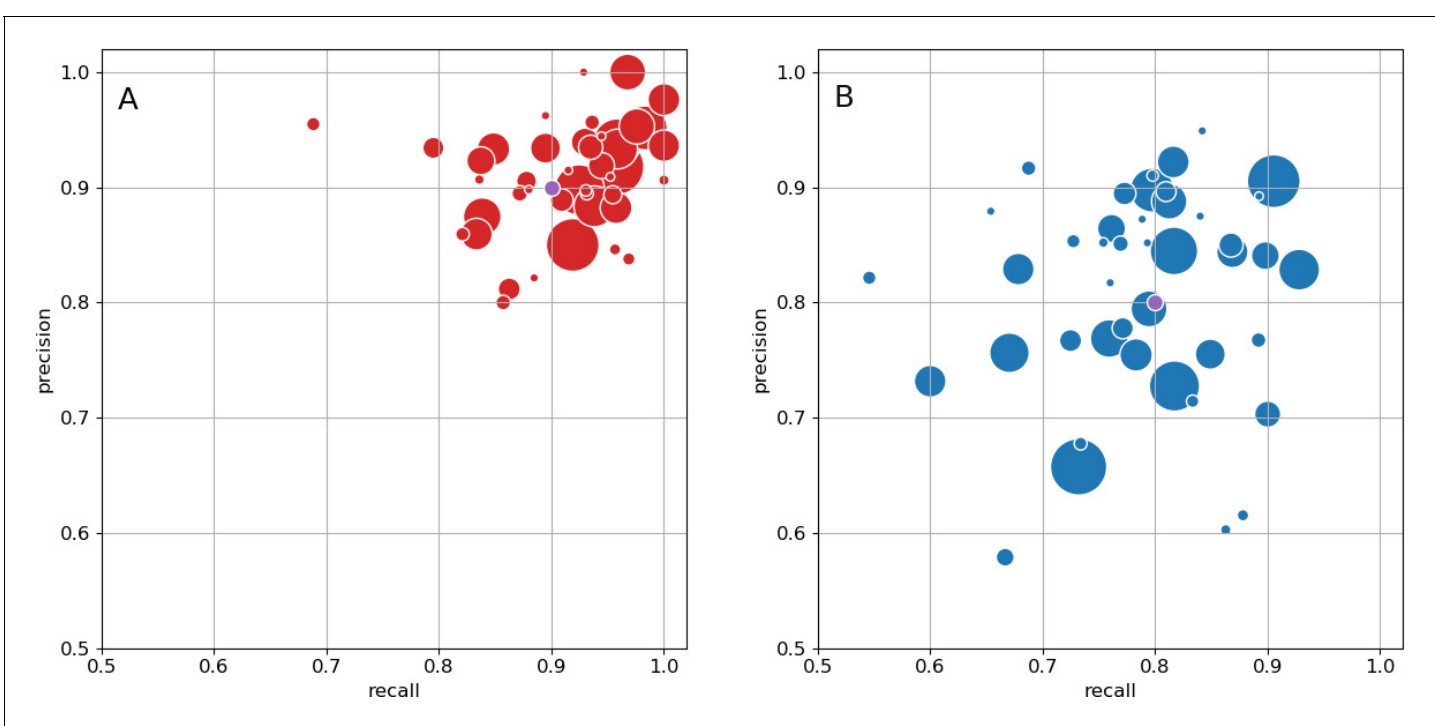

**Figure 5.** Precision and recall for synapse prediction, panel (**A**) for T-bars, and panel (**B**) for synapses as a whole including the identification of PSDs. T-bar identification is better than PSD identification since this organelle is both more distinct and typically occurs in larger neurites. Each dot is one brain region. The size of the dot is proportional to the volume of the region. Humans proofreaders typically achieve 0.9 precision/recall on T-bars and 0.8 precision/recall on PSDs, indicated in purple. Data available in *Figure 5—source datas 1–2*.

The online version of this article includes the following source data for figure 5:

**Source data 1.** Data for *Figure 5A*.

**Source data 2.** Data for *Figure 5B*.

## Proofreading

Since machine segmentation is not perfect, we made a concerted effort to fix the errors remaining at this stage by several passes of human proofreading. Segmentation errors can be roughly grouped into two classes - 'false merges', in which two separate neurons are mistakenly merged together, and 'false splits', in which a single neuron is mistakenly broken into several segments. Enabled by advances in visualization and semi-automated proofreading using our Neu3 tool (*Hubbard et al., 2020*), we first addressed large false mergers. A human examined each putative neuron and determined if it had an unusual morphology suggesting that a merge might have occurred, a task still much easier for humans than machines. If judged to be a false merger, the operator identified discrete points that should be on separate neurons. The shape was then resegmented in real time allowing users to explore other potential corrections. Neurons with more complex problems were then scheduled to be re-checked, and the process repeated until few false mergers remained.

In the next phase, the largest remaining pieces were merged into neuron shapes using a combination of machine-suggested edits (*Plaza, 2014*) and manual intuition, until the main shape of each neuron emerged. This requires relatively few proofreading decisions and has the advantage of producing an almost complete neuron catalog early in the process. As discussed below, in the section on validation, emerging shapes were compared against genetic/optical image libraries (where available) and against other neurons of the same putative type, to guard against large missing or superfluous branches. These procedures (which focused on higher-level proofreading) produced a reasonably accurate library of the main branches of each neuron, and a connectome of the stronger neuronal pathways. At this point, there was still considerable variations among the brain regions, with greater completeness achieved in regions where the initial segmentation performed better.

Finally, to achieve the highest reconstruction completeness possible in the time allotted, and to enable confidence in weaker neuronal pathways, proofreaders connected remaining isolated fragments (segments) to already constructed neurons, using NeuTu (*Zhao et al., 2018*) and Neu3 (*Hubbard et al., 2020*). The fragments that would result in largest connectivity changes were considered first, exploiting automatic guesses through focused proofreading where possible. Since proofreading every small segment is still prohibitive, we tried to ensure a basic level of completeness throughout the brain with special focus in regions of particular biological interest such as the central complex and mushroom body.

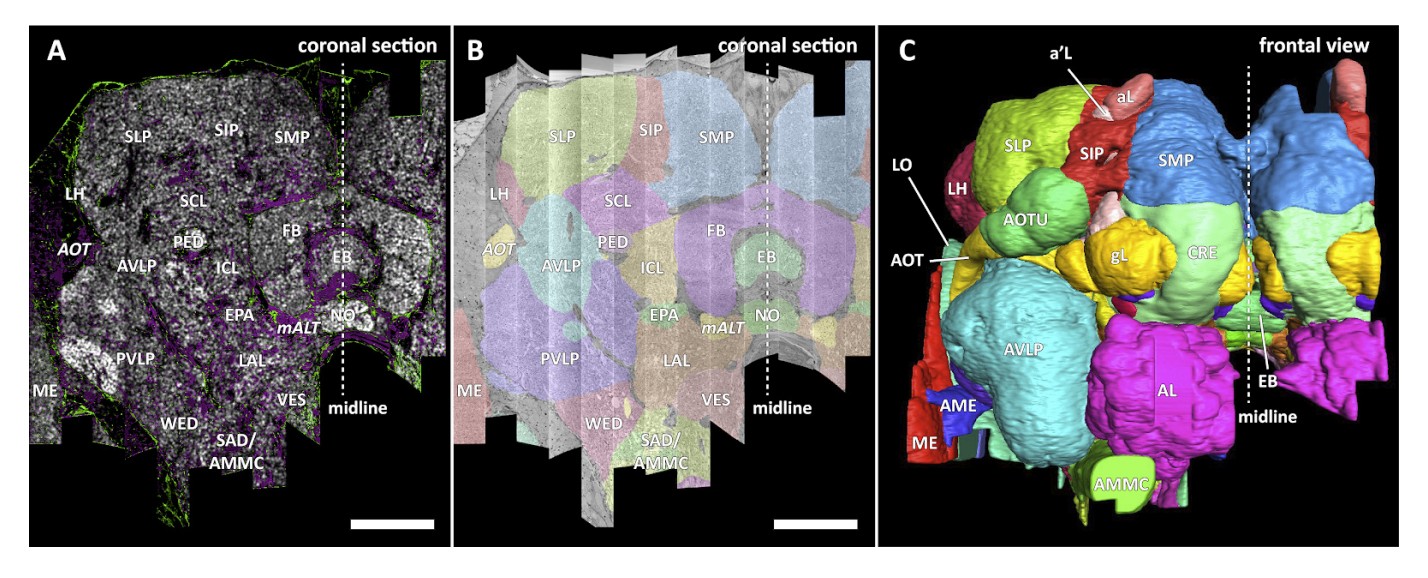

**Figure 6.** Division of the sample into brain regions. (A) A vertical section of the hemibrain dataset with synapse point clouds (white), predicted glial tissue (green), and predicted fiber bundles (magenta). (B) Grayscale image overlaid with segmented neuropils at the same level as (A). (C) A frontal view of the reconstructed neuropils. Scale bar: (A, B) 50 μm.

## Defining brain regions

In a parallel effort to proofreading, the sample was annotated with discrete brain regions. Our progression in mapping the cells and circuits of the fly's brain bears formal parallels to the history of mapping the earth, with many territories that are named and with known circuits, and others that still lack all or most of these. For the hemibrain dataset, the regions are based on the brain atlas in *Ito et al., 2014*. The dataset covers most of the right hemisphere of the brain, except the optic lobe (OL), periesophageal neuropils (PENP) and gnathal ganglia (GNG), as well as part of the left hemisphere (*Table 2*). It covers about 36% of all synaptic neuropils by volume, and 54% of the central brain neuropils. We examined innervation patterns, synapse distribution, and connectivity of reconstructed neurons to define the neuropils as well as their boundaries on the dataset. We also made necessary, but relatively minor, revisions to some boundaries by considering anatomical features that had not been known during the creation of previous brain maps, while following the existing structural definitions (*Ito et al., 2014*). We also used information from synapse point clouds, a predicted glial mask, and a predicted fiber bundle mask to determine boundaries of the neuropils (*Figure 6A*). The brain regions of the fruit fly (*Figure 6*, B and C) include synaptic neuropils and non-synaptic fiber bundles. The non-synaptic cell body layer on the brain surface, which contains cell bodies of the neurons and some glia, surrounds these structures. The synaptic neuropils can be further categorized into two groups: delineated and diffuse neuropils. The delineated neuropils have distinct boundaries throughout their surfaces, often accompanied by glial processes, and have clear internal structures in many cases. They include the antennal lobe (AL), bulb (BU), as well as the neuropils in the optic lobe (OL), mushroom body (MB), and central complex (CX). Remaining are the diffuse neuropils, sometimes referred to as *terra incognita*, since most have been less investigated than the delineated neuropils.

### Diffuse (*terra incognita*) neuropils

In the previous brain atlas of 2014, boundaries of some *terra incognita* neuropils were somewhat arbitrarily determined, due to a lack of precise information of the landmark neuronal structures used for the boundary definition. In the hemibrain data, we adjusted these boundaries to trace more faithfully the contours of the structures that are much better clarified by the EM-reconstructed data. Examples include the lateral horn (LH), ventrolateral neuropils (VLNP), and the boundary between the crepine (CRE) and lateral accessory lobe (LAL). The LH has been defined as the primary projection target of the olfactory projection neurons (PNs) from the antennal lobe (AL) via several antennal lobe tracts (ALTs) (*Ito et al., 2014*; *Pereanu et al., 2010*). The boundary between the LH and its surrounding neuropils is barely visible with synaptic immunolabeling such as nc82 or predicted synapse point clouds, as the synaptic contrast in these regions is minimal. The olfactory PNs can be grouped into several classes, and the projection sites of the uniglomerular PNs that project through the medial ALT (mALT), the thickest fiber bundle between the AL and LH, give the most conservative and concrete boundary of the 'core' LH (*Figure 7A*). Multiglomerular PNs, on the other hand, project to much broader regions, including the volumes around the core LH (*Figure 7B*). These regions include areas which are currently considered parts of the superior lateral protocerebrum (SLP) and posterior lateral protocerebrum (PLP). Since the 'core' LH roughly approximates the shape of the traditional LH, and the boundaries given by the multiglomerular PNs are rather diffused, in this study we assumed the core to be the LH itself. Of course, the multiglomerular PNs convey olfactory information as well, and therefore the neighboring parts of the SLP and PLP to some extent also receive inputs from the antennal lobe. These regions might be functionally distinct from the remaining parts of the SLP or PLP, but they are not explicitly separated from those neuropils in this study.

The VLNP is located in the lateral part of the central brain and receives extensive inputs from the optic lobe through various types of the visual projection neurons (VPNs). Among them, the projection sites of the lobula columnar (LC), lobula plate columnar (LPC), lobula-lobula plate columnar (LLPC), and lobula plate-lobula columnar (LPLC) cells form characteristic glomerular structures, called optic glomeruli (OG), in the AOTU, PVLP, and PLP (*Klapoetke et al., 2017*; *Otsuna and Ito, 2006*; *Panser et al., 2016*; *Wu et al., 2016*). We exhaustively identified columnar VPNs and found 41 types of LC, two types of LPC, six types of LLPC, and three types of LPLC cells (including sub-types of previously identified types). The glomeruli of these pathways were used to determine the medial boundary of the PVLP and PLP, following existing definitions (*Ito et al., 2014*), except for a few LC types

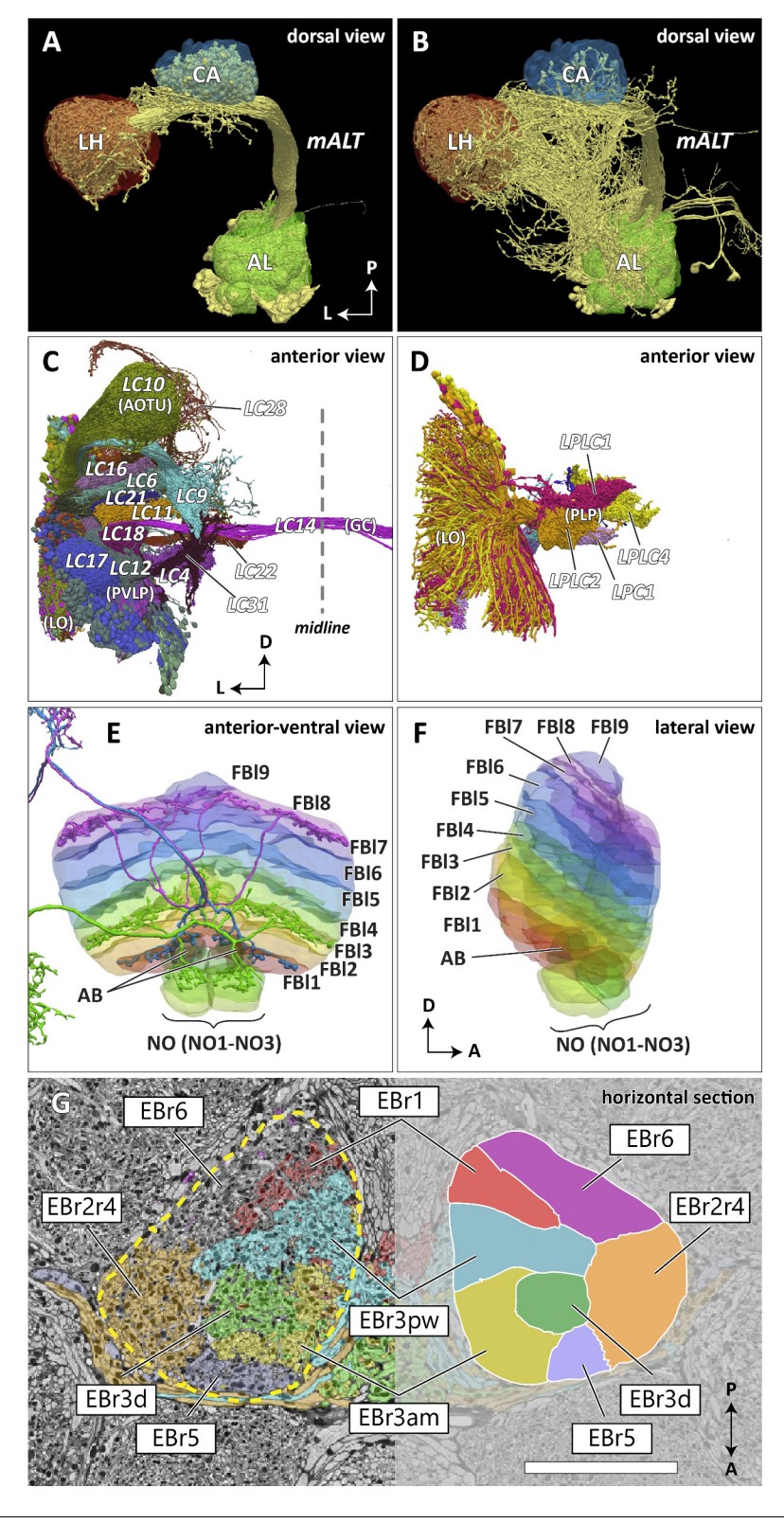

**Figure 7.** Reconstructed brain regions and substructures. (**A, B**) Dorsal views of the olfactory projection neurons (PNs) and the innervated neuropils, AL, CA, and LH. Uniglomerular PNs projecting through the mALT are shown in (**A**), and multiglomerular PNs are shown in (**B**). (**C, D**) Columnar visual projection neurons. Each subtype of cells is color coded. LC cells are shown in (**C**), and LPC, LLPC, and LPLC cells are shown in (**D**). (**E, F**) The nine layers of the fan-shaped body (FB), along with the asymmetrical bodies (AB) and the noduli (NO), displayed as an anterior-ventral view (**E**), and a lateral

*Figure 7 continued on next page*

*Figure 7 continued*

view (F). In (E), three FB tangential cells (FB1D (blue), FB3A (green), FB8H (purple)) are shown as markers of the corresponding layers (FBl1, FBl3, and FBl8, respectively). (G) Zones in the ellipsoid body (EB) defined by the innervation patterns of different types of ring neurons. In this horizontal section of the EB, the left side shows the original grayscale data, and the seven ring neuron zones (see *Table 1*) are color-coded. The right side displays the seven segmented zones based on the innervation pattern, in a slightly different section. Scale bar: 20 μm.

which do not form glomerular terminals. The terminals of the reconstructed LC cells and other lobula complex columnar cells (LPC, LLPC, LPLC) are shown in *Figure 7C and D*, respectively.

In the previous paper (*Ito et al., 2014*), the boundary between the CRE and LAL was defined as the line roughly corresponding to the posterior-ventral surface of the MB lobes, since no other prominent anatomical landmarks were found around this region. In this dataset, we found several glomerular structures surrounding the boundary both in the CRE and LAL. These structures include the gall (GA), rubus (RUB), and round body (ROB). Most of them turned out to be projection targets of several classes of central complex neurons, implying the ventral CRE and dorsal LAL are closely related in their function. We re-determined the boundary so that each of the glomerular structures would not be divided into two, while keeping the overall architecture and definition of the CRE and LAL. The updated boundary passes between the dorsal surface of the GA and the ventral edge of the ROB. Other glomerular structures, including the RUB, are included in the CRE.

## Delineated neuropils

Substructures of the delineated neuropils have also been added to the brain region map in the hemibrain. The asymmetrical bodies (AB) were added as the fifth independent neuropil of the CX (*Wolff and Rubin, 2018*). The AB is a small synaptic volume adjacent to the ventral surface of the fan-shaped body (FB) that has historically been included in the FB (*Ito et al., 2014*). The AB has been described as a Fasciclin II (FasII)-positive structure that exhibits left-right structural asymmetry by *Pascual et al., 2004*, who reported that most flies have their AB only in the right hemisphere, while a small proportion (7.6%) of wild-type flies have their AB on both sides. In the hemibrain dataset, the pair of ABs is situated on both sides of the midline, but the left AB is notably smaller than the right AB (right: 1679 μm$^3$, left: 526 μm$^3$), still showing an obvious left-right asymmetry. The asymmetry is consistent with light microscopy data (*Wolff and Rubin, 2018*), though the absolute sizes differ, with the light data showing averages (n = 21) of 522 μm$^3$ for the right and 126 μm$^3$ on the left. The AB is especially strongly connected to the neighboring neuropil, the FB, by neurons including vDeltaA_a (anatomical name AF in *Wolff and Rubin, 2018*), while it also houses both pre- and postsynaptic terminals of the CX output neurons such as the subset of FS4A and FS4B neurons that project to AB. These anatomical observations imply that the AB is a ventralmost annexed part of the FB, although this possibility is neither developmentally nor phylogenetically proven.

The round body (ROB) is also a small round synaptic structure situated on the ventral limit of the crepine (CRE), close to the β lobe of the MB (*Lin et al., 2013*; *Wolff and Rubin, 2018*). It is a glomerulus-like structure and one of the foci of the CX output neurons, including the PFR (protocerebral bridge – fan-shaped body – round body) neurons. It is classified as a substructure of the CRE along with other less-defined glomerular regions in the neuropil, many of which also receive signals from the CX. Among these, the most prominent one is the rubus (RUB). The ROB and RUB are two distinct structures; the RUB is embedded completely within the CRE, while the ROB is located on the ventrolateral surface of the CRE. The lateral accessory lobe (LAL), neighboring the CRE, also houses similar glomerular terminals, and the gall (GA) is one of them. While the ROB and GA have relatively clear boundaries separating them from the surrounding regions, they may not qualify as independent neuropils because of their small size and the structural similarities with the glomerulus-like terminals around them. They may be comparable with other glomerular structures such as the AL glomeruli and the optic glomeruli in the lateral protocerebrum, both of which are considered as substructures of the surrounding neuropils.

Substructures of independent neuropils are also defined using neuronal innervations. The five MB lobes on the right hemisphere are further divided into 15 compartments (α1–3, α′1–3, β1–2, β′1–2, and γ1–5) (*Tanaka et al., 2008*; *Aso et al., 2014*) by the mushroom body output neurons (MBONs) and dopaminergic neurons (DANs). Our compartment boundaries were defined by approximating the innervation of these neurons. Although the innervating regions of the MBONs and DANs do not

perfectly tile the entire lobes, the compartments have been defined to tile the lobes, so that every synapse in the lobes belongs to one of the 15 compartments.

The anatomy of the central complex is discussed in detail in the companion paper 'A complete synaptic-resolution connectome of the *Drosophila melanogaster* central complex'. Here, we summarize the division of its neuropils into compartments.

The FB is subdivided into nine horizontal layers (FBl1-9) (*Figure 7E and F*) as already illustrated (*Wolff et al., 2015*). The layer boundaries in our dataset were determined by the pattern of innervation of 574 FB tangential cells, which form nine groups depending on the dorsoventral levels they innervate in the FB. Since tangential cells overlap somewhat, and do not entirely respect the layer boundaries, these boundaries were chosen to maximize the containment of the tangential arbors within their respective layers.

The EB is likewise subdivided into zones by the innervating patterns of the EB ring neurons, the most prominent class of neurons innervating the EB. The ring neurons have six subtypes, ER1-ER6, and each projects to specific zones of the EB. Among them, the regions innervated by ER2 and ER4 are mutually exclusive but highly intermingled, so these regions are grouped together into a single zone (EBr2r4). ER3 has the most neurons among the ring neuron subtypes and is further grouped into five subclasses (ER3a, d, m, p, and w). While each subclass projects to a distinct part of the EB, the innervation patterns of the subclasses ER3a and ER3m, and also ER3p and ER3w, are very similar to each other. The region innervated by ER3 is, therefore, subdivided into three zones, including EBr3am, EBr3pw, and EBr3d. Along with the other three zones, EBr1, EBr5, and EBr6 (innervated by ER1, ER5, and ER6), the entire EB is subdivided into seven non-overlapping zones (*Figure 7G*). Unlike other zones, EBr6 is innervated only sparsely by the ER6 cells, with the space filled primarily by synaptic terminals of other neuron types, including the extrinsic ring neurons (ExR). *Omoto et al., 2017* segmented the EB into five domains (EBa, EBoc, EBop, EBic, EBip) by the immunolabeling pattern of DN-cadherin, and each type of the ring neurons may innervate more than one domain in the EB. Our results show that the innervation pattern of each ring neuron subtype is highly compartmentalized at the EM level and the entire neuropil can be sufficiently subdivided into zones based purely on the neuronal morphologies. The neuropil may be subdivided differently if other neuron types, such as the extrinsic ring neurons (ExR) (*Omoto et al., 2018*), are recruited as landmarks.

## Quality of the brain region boundaries

Since many of the *terra incognita* neuropils are not clearly partitioned from each other by solid boundaries such as glial walls, it is important to evaluate if the current boundaries reflect anatomical and functional compartments of the brain. To check our definitions, which are mostly based on morphology, we compute metrics for each boundary between any two adjacent neuropil regions. The first is the area of each boundary, in square microns, as shown in *Figure 8A*. The map shows results for brain regions that are over 75% in the hemibrain region, restricted to right regions with exception to the asymmetric AB(L). By restricting our analysis to the right part of the hemibrain, we hopefully minimize the effect of smaller, traced-but-truncated neuron fragments on our metric.

Next, for each boundary, we compute the number of 'excess' neuron crossings by traced neurons, where excess crossings are defined as 0 for a neuron that does not cross the boundary, and $n - 1$ for a neuron crosses the same boundary $n$ times. There is no contribution to the metric from neurons that cross a boundary once, since most such crossings are inevitable no matter where the boundary is placed. *Figure 8B* shows the number of excess crossings normalized by the area of boundary. A bigger dot indicates a potentially less well-defined boundary.

We spot checked many of the instances and in general note that the brain regions with high excess crossings per area, such as those in SNP, INP and VLNP, tend to have less well-defined boundaries. In particular, the boundaries at SMP/CRE, CRE/LAL, SMP/SIP, and SIP/SLP have worse scores, indicating these boundaries may not reflect actual anatomical and functional segregation of the neuropils. These brain regions were defined based on the arborization patterns of characteristic neuron types, but because neurons in the *terra incognita* neuropils tend to be rather heterogeneous, there are many other neuron types that do not follow these boundaries. The boundary between the FB and the AB also has a high excess crossing score, suggesting the AB is tightly linked to the neighboring FB.

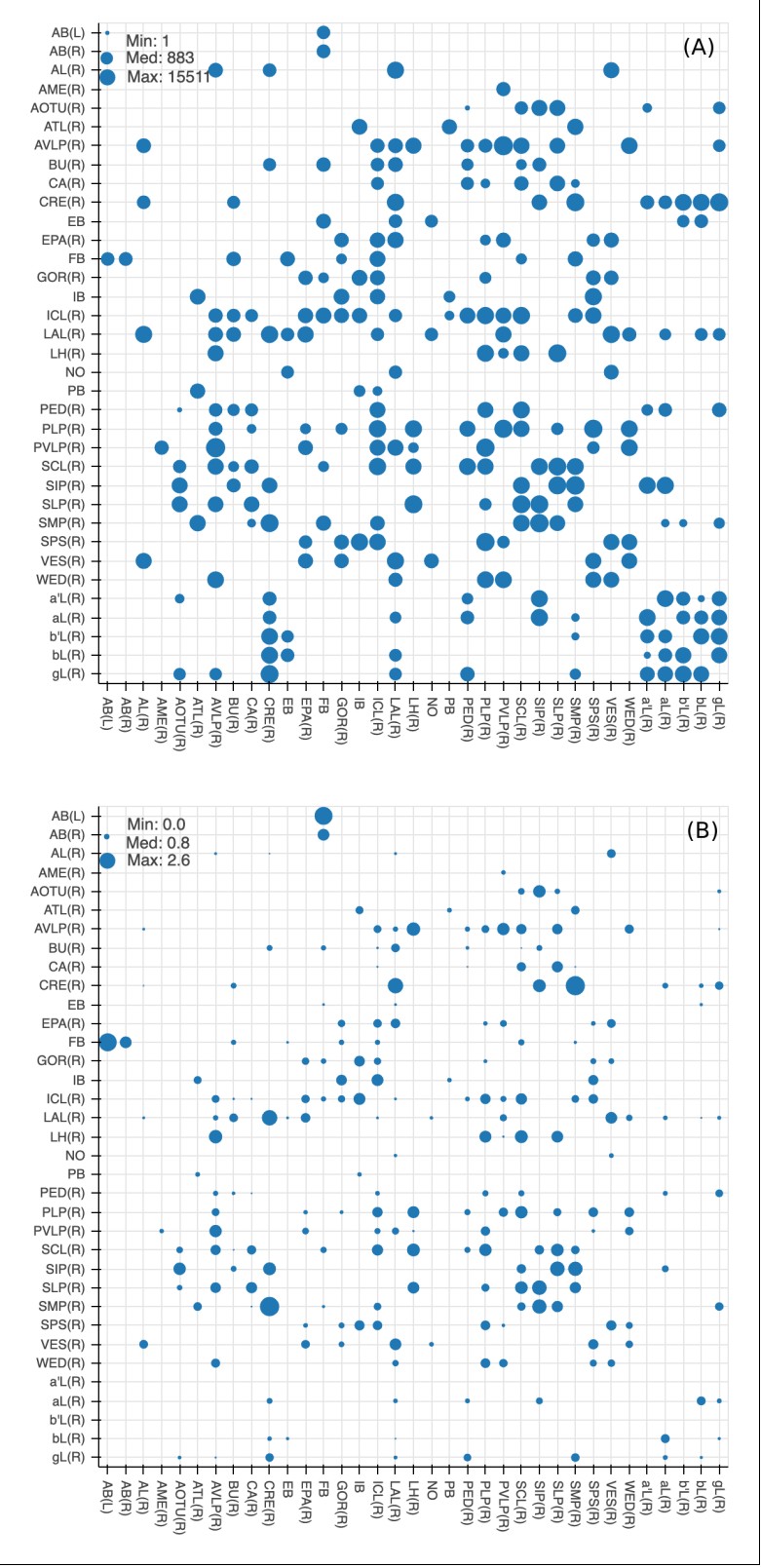

**Figure 8.** Quality checks of the brain compartments. (**A**) Areas of the boundaries (in square microns) between adjacent neuropils, indicated on a log scale. (**B**) The number of excess crossings normalized by the area of neuropil boundary. Larger dots indicate a more uncertain boundary. Data available in *Figure 8—source data 1*.

*Figure 8 continued on next page*

*Figure 8 continued*

The online version of this article includes the following source data for figure 8:

**Source data 1.** Data for *Figure 8*.

## Insights for a whole-brain remapping

The current brain regions based on *Ito et al., 2014* contain a number of arbitrary determinations of brain regions and their boundaries in the *terra incognita* neuropils. In this study, we tried to solidify the ambiguous boundaries as much as possible using the information from the reconstructed neurons. However, large parts of the left hemisphere and the subesophageal zone (SEZ) are missing from the hemibrain dataset, and neurons innervating these regions are not sufficiently reconstructed. This incompleteness of the dataset is the main reason that we did not alter the previous map drastically and kept all the existing brain regions even if their anatomical and functional significance is not obvious. Once a complete EM volume of the whole fly brain is imaged and most of its 100,000 neurons are reconstructed, the entire brain can be re-segmented from scratch with more comprehensive anatomical information. Arbitrary or artificial neuropil boundaries will thereby be minimized, if not avoided, in a new brain map. Anatomy-based neuron segmentation strategies such as NBLAST may be used as neutral methods to revise the neuropils and their boundaries. Any single method, however, is not likely to produce consistent boundaries throughout the brain, especially in the *terra incognita* regions. It may be necessary to use different methods and criteria to segment the entire brain into reasonable brain regions. Such a new map would need discussion in a working group, and approval from the community in advance (as did the previous map [*Ito et al., 2014*]), insofar as it would replace the current map and therefore require a major revision of the neuron mapping scheme.

## Cell type classification

Defining cell types for groups of similar neurons is a time-honored means to help to understand the anatomical and functional properties of a circuit. Presumably, neurons of the same type have similar circuit roles. However, the definition of what is a distinct cell type and the exact delineation between one cell type and another remains inherently subjective and represents a classic taxonomic challenge, pitting 'lumpers' against 'splitters'. Therefore, despite our best efforts, we recognize that our typing of cells may not be identical to that proposed by other experts. We expect future revisions to cell type classification, especially as additional dense connectome data become available.

One common method of cell type classification, used in flies, exploits the GAL4 system to highlight the morphology of neurons having similar gene expression (*Jenett et al., 2012*). Since these genetic lines are imaged using fluorescence and confocal microscopy, we refer to them as 'light lines'. Where they exist and are sufficiently sparse, light lines provide a key method for identifying types by grouping morphologically similar neurons together. However, there are no guarantees of coverage, and it is difficult to distinguish between neurons of very similar morphology but different connectivity.

We enhanced the classic view of morphologically distinct cell types by defining distinct cell types (or sub-types) based on both morphology and connectivity. Connectivity-based clustering often reveals clear cell type distinctions, even when genetic markers have yet to be found, or when the neuronal morphologies of different types are hardly distinguishable in optical images. For example, the two PEN (protocerebral bridge - ellipsoid body - noduli) neurons have very similar forms but quite distinct inputs (*Figure 9*; *Turner-Evans et al., 2019*) Confirming their differences, PEN1 and PEN2 neurons, in fact, have been shown to have different functional activity (*Green et al., 2017*).

Based on our previous definition of cell type, many neurons exhibit a unique morphology or connectivity pattern at least within one hemisphere of the brain (with a matching type in the other hemisphere in most cases). Because our hemibrain volume covers only the right-side examples of ipsilaterally-projecting neurons, and the contralateral arborizations of bilaterally-projecting neurons arising from the left side of the brain were in practice very difficult to match to neurons in the right side, many partial neurons were therefore left uncategorized. As a result, many neuron types consisting of a distinct morphology and connectivity have only a single example in our reconstruction.

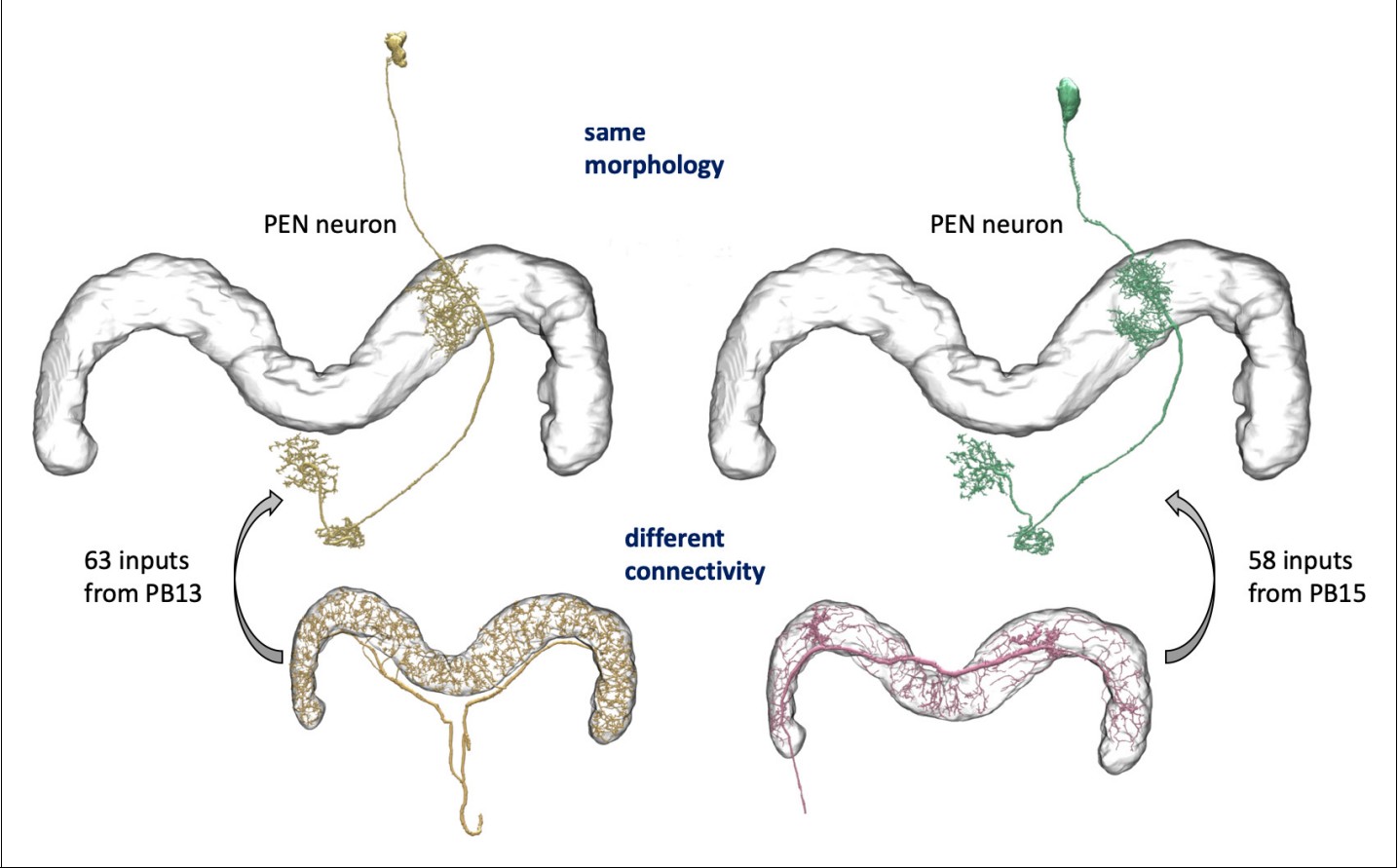

**Figure 9.** An example of two neurons with very similar shapes but differing connectivities. PEN1 is on the left, PEN2 on the right.

It is possible to provide coarser groupings of neurons. For instance, most cell types are grouped by their cell body fiber representing a distinct clonal unit, which we discuss in more detail below. Furthermore, each neuron can be grouped with neurons that innervate similar brain regions. In this paper, we do not explicitly formalize this higher level grouping, but data on the innervating brain regions can be readily mined from the dataset.

## Methodology for assigning cell types and nomenclature

Assigning types and names to the more than 20,000 reconstructed cells was a difficult undertaking. Less than 20% of neuron types found in our data have been described in the literature, and half of our neurons have no previously annotated type. Adding to the complexity, prior work focused on morphological similarities and differences, but here we have, for the first time, connectivity information to assist in cell typing as well.

Many cell types in well-explored regions have already been described and named in the literature, but existing names can be both inconsistent and ambiguous. The same cell type is often given differing names in different publications, and conversely, the same name, such as PN for projection neuron, is used for many different cell types. Nonetheless, for cell types already named in the literature (which we designate as published cell types, many indexed, with their synonyms, at http://virtualflybrain.org), we have tried to use existing names. In a few cases, using existing names created conflicts, which we have had to resolve. 'R1', for example, has long been used both for photoreceptor neurons innervating the lamina and medulla, and ring neurons in the ellipsoid body of the central complex. Similarly, 'LN' has been used to refer to lateral neurons in the circadian clock system, 'local neurons' in the antenna lobe, and LAL-Nodulus neurons in the central complex. To resolve these conflicts, the ellipsoid body ring neurons are now named 'ER1' instead of 'R1', and the nodulus neurons are now 'LNO' and 'GLNO' instead of 'LN' and 'GLN'. The names of the antennal lobe local

neuron are always preceded by lowercase letters for their cell body locations to differentiate them from the clock neuron names, for example, lLN1 versus LNd. Similarly, 'dorsal neurons' of the circadian clock system and 'descending neurons' in general, both previously abbreviated as 'DN', are distinguished by the following characters - numbers for the clock neurons (e.g. DN1) and letters for descending neurons (e.g. DNa01).

Overall, we defined a 'type' of neurons as either a single cell or a group of cells that have a very similar cell body location, morphology, and pattern of synaptic connectivity. We were able to trace from arborizations to the cell bodies for 15,912 neurons in the hemibrain volume, ≈85% of which are located in the right side of the brain while the rest are in the medialmost part of the left-side brain.

We classified these neurons in several steps. The first step classified all cells by their lineage, grouping neurons according to their bundle of cell body fibers (CBFs). Neuronal cell bodies are located in the cell body layer that surrounds the brain, and each neuron projects a single CBF towards synaptic neuropils. In the central brain, cell bodies of clonally related neurons deriving from a single stem cell (called a neuroblast in the insect brain) tend to form clusters, from each of which arises one or several bundles of CBFs. Comparing the location, trajectory, and the combined arborization patterns of all the neurons that arise from a particular CBF with the light microscopy (LM) image data of the neuronal progeny that derive from single neuroblasts (*Ito et al., 2013*; *Yu et al., 2013*), we confirmed that the neurons of each CBF group belong to a single lineage.

We carefully examined the trajectory and origins of CBFs of the 15,752 neurons on the right central brain and identified 192 distinct CBF bundles. Neurons arising from four specific CBF bundles arborize primarily in the contralateral brain side, which is not fully covered in the hemibrain volume. We characterized these neurons using the arborization patterns in the right-side brain that are formed by the neurons arising from the left-side CBFs.

The CBF bundles and associated neuronal cell body clusters were named according to their location (split into eight sectors of the brain surface with the combination of Anterior/Posterior, Ventral/Dorsal, and Medial/Lateral) and a number within the sector given according to the size of cell population. Thus, CBF group ADM01 is the group with the largest number of neurons in the Anterior Dorsal Medial sector of the brain's surface (see the cellBodyFiber field of the Neuprint database explained later). For the neurons of the four CBF bundles that arborize primarily in the contralateral brain side - AVM15, 18, 19, and PVM10 - we indicated CBF information in the records of the left-side neurons.

Among the 192 bundles, 155 matched the CBF bundles of 92 known and six newly identified clonal units (*Ito et al., 2013*; *Yu et al., 2013*), a population of neurons and neuronal circuits derived from a single stem cell. The remaining 37 CBF bundles are minor populations and most likely of embryonic origin. In addition, we found 80 segregated cell body fiber bundles (SCB001-080, totalling 112 cells) with only one or two neurons per bundle. Many of them are also likely of embryonic origin.

We were able to identify another 6682 neurons that were not traced up to their cell bodies. For the neurons that arise from the contralateral side, we gave matching neuron names and associated CBF information, provided their specific arborization patterns gave us convincing identity information by comparison with cells that we identified in the right side of the brain. For the neurons arising from the ventralmost part of the brain outside of the hemibrain volume, we identified and gave them names if we could find convincingly specific arborization patterns, even if the CBF and cell body location data were missing. Sensory neurons that project to the specific primary sensory centers were also identified insofar as possible. In total, we typed and named 22,594 neurons.

Different stem cells sometimes give rise to neurons with very similar morphologies. We classified these as different types because of their distinct developmental origin and slightly different locations of their cell bodies and CBFs. Thus, the next step in neuron typing was to cluster neurons within each CBF group. This process consisted of three further steps, as shown in *Figure 10*. First, we used NBLAST (*Costa et al., 2016*) to subject all the neurons of a particular CBF group to morphology-based clustering. Next, we used CBLAST, a new tool to cluster neurons based on synaptic connectivity (see the next section). This step is an iterative process, using neuron morphology as a template, regrouping neurons after more careful examination of neuron projection patterns and their connections. Neurons with similar connectivity characteristics but with distinguishable shapes were categorized into different morphology types. Those with practically indistinguishable shapes but with

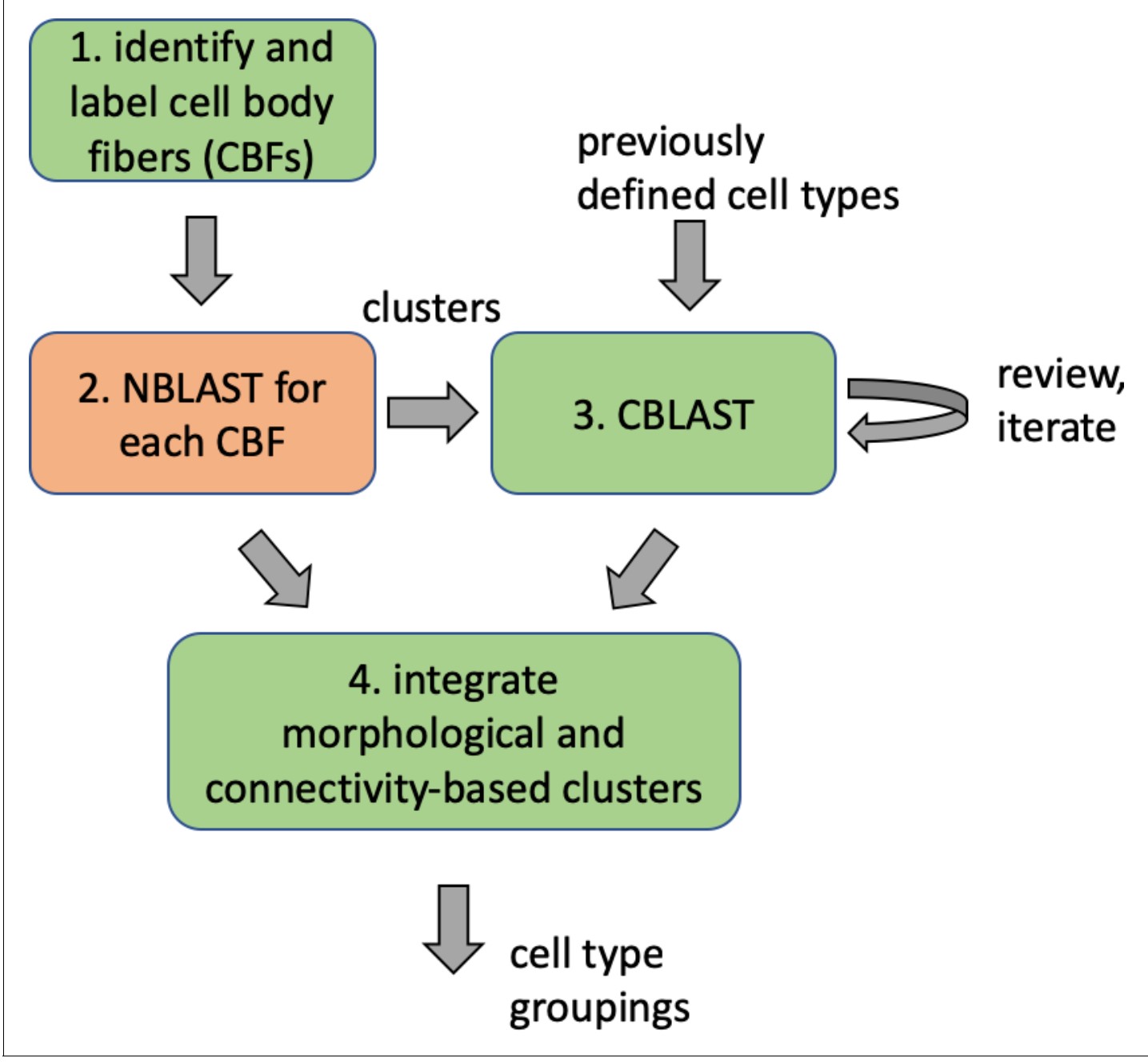

**Figure 10.** Workflow for defining cell types.

different connectivity characteristics were categorized into connectivity types within a morphology type. Finally, we validated the cell typing with extensive manual review and visual inspection. This review allowed us both to confirm cell type identity and help ensure neuron reconstruction accuracy. In total we identified 5229 morphology types and 5609 connectivity types in the hemibrain dataset. (See *Table 3* for the detailed numbers and *Appendix 1—table 6* for naming schemes for various neuron categories.)

In spite of this general rule, we assigned the same neuron type name for the neurons of different lineages in the following four cases.

- Mushroom body intrinsic neurons called Kenyon cells, which are formed by a set of four near-identical neuroblasts (*Ito et al., 1997*) (see also the accompanying MB paper).

**Table 3.** Summary of the numbers and types of the neurons in the hemibrain EM dataset.

*m-types* is the number of morphology types; *c-types* the number of connectivity types; and *c/t* the average number of cells per connectivity type. Brain regions with repetitive array architecture tend to have higher average numbers of cells per type (see **Figure 12**). The cell number includes ≈ 4000 neurons on the contralateral side, and the percentage of contralateral cells varies between 0 and ≈ 50% depending on the category. For example, the central complex includes neurons on both sides of the brain, the mushroom body neurons are identified mostly on the right side, and many left-side antennal lobe sensory neurons are included as they tend to terminate bilaterally. Because of these differences, the figures shown above do not indicate the number of cells (or cell number per type) per brain side.

| Brain regions (neuropils) or neuron types | Cells | m-types | c-types | C/t | Notes |
|---|---|---|---|---|---|
| Central complex neuropil neurons | 2826 | 224 | 262 | 10.8 | |
| Mushroom body neuropil neurons | 2315 | 72 | 80 | 28.9 | Including MB-associated DANs |
| Mushroom body neuropil neurons | 2003 | 51 | 51 | 39.3 | Excluding MB-associated DANs |
| Dopaminergic neurons (DANs) | 335 | 35 | 43 | 7.8 | Including MB-associated DANs |
| Dopaminergic neurons (DANs) | 23 | 14 | 14 | 1.7 | Excluding MB-associated DANs |
| Octopaminergic neurons | 19 | 10 | 10 | 1.9 | |
| Serotonergic (5HT) neurons | 9 | 5 | 5 | 1.8 | |
| Peptidergic and secretory neurons | 51 | 12 | 14 | 3.6 | |
| Circadian clock neurons | 27 | 7 | 7 | 3.9 | |
| Fruitless gene expressing neurons | 84 | 29 | 30 | 2.8 | |
| Visual projection neurons and lobula intrinsic neurons | 3723 | 160 | 160 | 23.3 | |
| Descending neurons | 103 | 51 | 51 | 2.0 | |
| Sensory associated neurons | 2768 | 67 | 67 | 41.3 | |
| Antennal lobe neuropil neurons | 604 | 284 | 294 | 2.1 | |
| Lateral horn neuropil neurons | 1496 | 517 | 683 | 2.2 | |
| Anterior optic tubercle neuropil neurons | 243 | 77 | 80 | 3.0 | |
| Antler neuropil neurons | 81 | 45 | 45 | 1.8 | |
| Anterior ventrolateral protocerebrum neuropil neurons | 1276 | 596 | 629 | 2.0 | |
| Clamp neuropil neurons | 746 | 364 | 382 | 2.0 | |
| Crepine neuropil neurons | 333 | 108 | 115 | 2.9 | |
| Inferior bridge neuropil neurons | 264 | 119 | 119 | 2.2 | |
| Lateral accessory lobe neuropil neurons | 429 | 204 | 206 | 2.1 | |
| Posterior lateral protocerebrum neuropil neurons | 480 | 255 | 260 | 1.8 | |
| Posterior slope neuropil neurons | 621 | 303 | 311 | 2.0 | |
| Posterior ventrolateral protocerebrum neuropil neurons | 348 | 151 | 156 | 2.2 | |
| Saddle neuropil and antennal mechanosensory and motor center neurons | 219 | 96 | 99 | 2.2 | |
| Superior lateral protocerebrum neuropil neurons | 1096 | 468 | 494 | 2.2 | |
| Superior intermediate protocerebrum neuropil neurons | 220 | 90 | 92 | 2.4 | |
| Superior medial protocerebrum neuropil neurons | 1494 | 605 | 629 | 2.4 | |
| Vest neuropil neurons | 137 | 84 | 85 | 1.6 | |
| Wedge neuropil neurons | 559 | 212 | 230 | 2.4 | |
| Total | 22,594 | 5229 | 5609 | 4.0 | |

- Columnar neurons of the central complex, where neurons arising from different stem cells form repetitive column-like arrangement and are near identical in terms of connectivity with tangential neurons (*Hanesch et al., 1989*; *Wolff et al., 2015*; *Wolff and Rubin, 2018*) (and the accompanying CX paper).
- The PAM cluster of the dopaminergic neurons, where one of the hemilineages of the two clonal units forms near identical set of neurons (*Lee et al., 2020*) (accompanying MB paper).

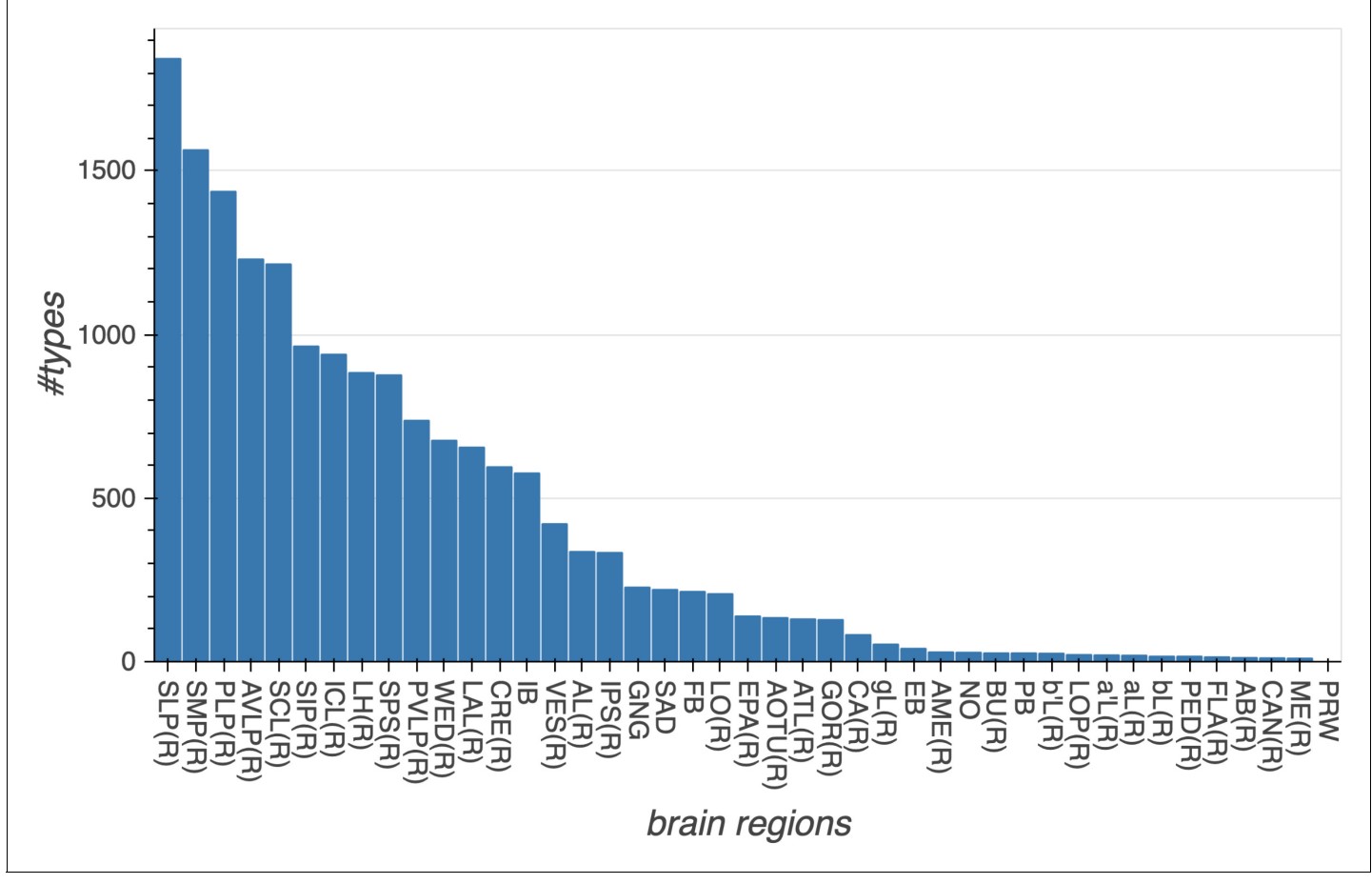

**Figure 11.** The number of cell types in each major brain region. The total number of cell types shown in this graph is larger than the total number of cell types shown in *Table 3*, because types that arborize in multiple regions are counted in each region in which they occur. Data available in *Figure 11—source data 1*.

The online version of this article includes the following source data for figure 11:

**Source data 1.** Data for *Figure 11*.

- Cell body fiber groupings for neurons of the lateral horn, where systematic neuron names have already been given based on the light microscopy analysis (*Frechter et al., 2019*), which did not allow for the precise segregation of very closely situated CBF bundles. Individual cell types exist within the same lineage, however.

'Lumping' versus 'splitting' is a difficult problem for classification. Following the experiences of taxonomy, we opted for splitting when we could not obtain convincing identity information, a decision designed to ease the task of future researchers. If we split two similar neuron types into Type 1 and Type 2, then there is a chance future studies might conclude that they are actually subsets of a common cell type. If so, then at that time we can simply merge the two types as Type 1, and leave the other type name unused, and publish a lookup table of the lumping process to keep track of the names that have been merged. The preceding studies can then be re-interpreted as the analyses on the particular subsets of a common neuron type. If, on the contrary, we lump the two similar neurons into a common type, then a later study finds they are actually a mixture of two neuron types, then it would not be possible to determine which of the two neuron types, or a mixture of them, was analyzed in preceding studies.

In the hemibrain, using the defined brain regions (neuropils) and reference to known expression driver strains, we were able to assign a cell type to many cells. Where possible, we matched previously defined cell types with those labeled in light data using a combination of Neuprint, an interactive analysis tool (described later), Color_MIP_mask search (*Otsuna et al., 2018*), and human

recognition to find the matching cell types, especially in well-explored neuropils such as the mushroom body and central complex, where abundant cell type information was already available and where we are more confident in our anatomical expertise (see the accompanying MB and CX papers). Even though most of the cell types in the MB and CX were already known, we still found new cell types in these regions, an important vindication of our methods. In these cases, we tried to name them using the existing schemes for these regions, and further refined these morphological groupings with relevant information on connectivity.

To give names to neuron types, we categorized neurons that share certain characteristics into groups and distinguished individual types by adding identifiers (IDs) with numbers, uppercase letters, or combinations of these. (See *Appendix 1—table 6* for the summary of the naming schemes of all the neuron types). For example, the tangential neurons of the fan-shaped body (FB) of the central complex were grouped as 'FB', and an ID of their primary innervating FB layer was added with numbers 1–9. Different types of neurons that arborize in each layer were further distinguished by uppercase letters. Thus, for example the FB7B neurons are the second type of tangential neurons that arborize in the seventh layer of FB. We also used uppercase letters to subdivide the neuron types that have previously been reported as a single type to keep naming consistency. For example, a population of antennal lobe local neurons that has been known as LN2L was divided into five morphology subtypes as lLN2F, 2P, 2R, 2S, and 2T for their full, patchy, regional, star-like and tortuous arborization patterns while still indicating that they are part of the LN2 population. The letter 'L' at the end of the previous name, which referred to the cell body location on the lateral side of the AL, was moved in front of LN to keep consistency with the established naming scheme for the olfactory projection neurons (e.g., DA1_lPN).

Neuron types that are known to exist were sometimes not identified in the particular brain sample used for the hemibrain EM dataset. In such cases, the corresponding ID numbers were kept blank. For example, the MBON08 neurons were not identified in the current sample and the number was therefore skipped.

Although the morphology type names generally end with either numbers or uppercase letters, in a few cases lower case letters were used for distinguishing morphological subtypes to keep the naming convention of that cell group consistent. For example, subtypes of the neurons in the optic lobes were distinguished as, for example LC28a and LC28b, because such subtypes of the optic lobe neurons have historically been distinguished by lowercase letters.

If neurons of near-identical morphology could be further subdivided into different connectivity types, they were suffixed with an underscore and a lowercase letter, for example FB2F_a, FB2F_b, and FB2F_c. A neuron type without such a suffix consists of a single connectivity type.

The cell type names are indicated in the 'type' field of the NeuPrint database. In the 'instance' field, information about the side of the neuronal cell body, when it is known, is added as _R and _L after the cell type name. The name of the CBF group is indicated in the 'cellBodyFiber' field of the right-side neurons except for those that belong to AVM15, 18, 19, and PVM10 groups, and in the same field of the left-side neurons for those four CBF groups. For the rest of the neurons, the CBF information is shown in the 'instance' field in parentheses when it is known.

Across the brain, we looked for neurons that correspond to already known cell types, and as far as possible gave them consistent names. These include: olfactory projection neurons and local neurons associated with the antennal lobe (*Tanaka et al., 2012*; *Bates et al., 2020*; *Marin et al., 2020*), neurons associated with the lateral horn (*Dolan et al., 2019*; *Frechter et al., 2019*; *Bates et al., 2020*), aminergic and peptidergic neurons (*Bergland et al., 2012*; *Busch et al., 2009*; *Mao and Davis, 2009*; *Martelli et al., 2017*; *Pech et al., 2013*; *Pooryasin and Fiala, 2015*; *Shao et al., 2017*; *White et al., 2010*), neurons associated with the circadian clock (*Helfrich-Förster et al., 2007*), and neurons that express the fruitless gene (*Cachero et al., 2010*; *Yu et al., 2010*; *Zhou et al., 2014*; *Wang et al., 2020*).

In some cases, we found candidate neurons that do not precisely match previously identified neurons. For example, in addition to the three cell types that match the octopaminergic (OA) neurons OA-ASM1, 2 and 3 (*Busch et al., 2009*), we found two neuron types in the same location that appear to match some of the tdc2-Gal4 expressing neurons in the FlyCircuit database of single-cell labeling images (*Chiang et al., 2011*). Because of the remaining uncertainty we gave them the canonical names SMP143 and SMP149, but added 'Tdc2 (OA)-ASM candidates' in the *Notes* field. We also found that the FB2B neurons share the same cell body location and appear to match

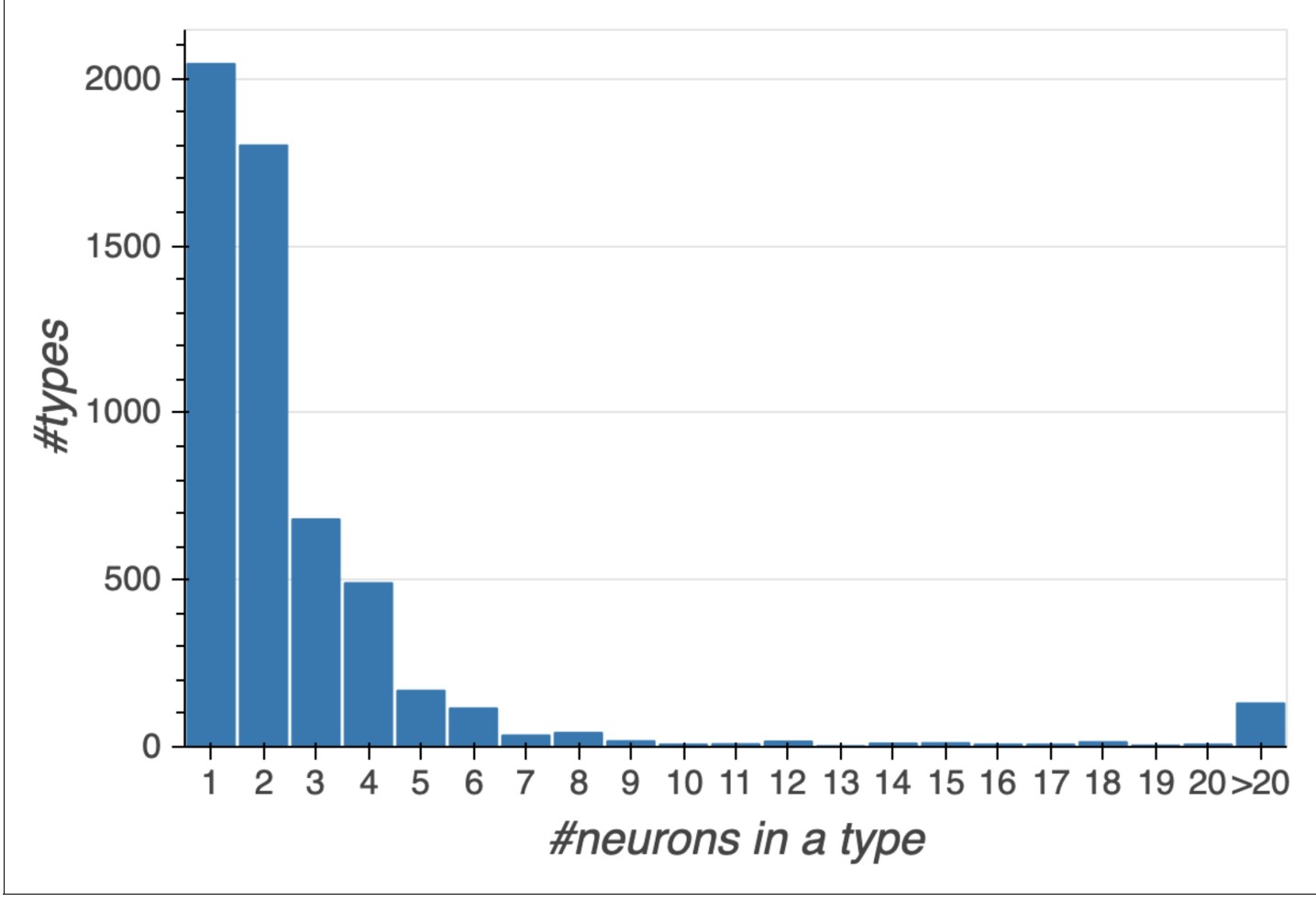

**Figure 12.** Histogram showing the number of cell types with a given number of constituent cells. Data available in *Figure 12—source data 1*. The online version of this article includes the following source data for figure 12:

**Source data 1.** Data for *Figure 12*.

another type of tdc2-Gal4 expressing neurons in the FlyCircuit database. Although OA-immunoreactive neurites have been observed in the FB (*Sinakevitch et al., 2005*), it is not known from where they are derived. Considering that the particular neurons may produce only tyramine (TA) but not OA, we added 'Tdc2 (TA)-ASM candidates' in the Notes. Due to similar considerations, the number of candidate neurons may not match the actual known numbers for many neuron types.

For the multiglomerular olfactory projection neurons and local interneurons of the antennal lobe, we devised new naming schemes by expanding the naming scheme of uniglomerular projection neurons, which consists of the contributing antennal lobe glomerulus and the location of the cell body cluster (*Bates et al., 2020*; *Marin et al., 2020*). Because the list of contributing glomeruli is not a useful designator for the multiglomerular projection neurons, we used information about the antennal lobe tract (ALT) projection pathways instead. Unique type ID numbers were then added at the end of the names of the multiglomerular projection neurons (1-92) and local neurons (1-50). For the local neurons LN1-6 the numbers were kept consistent with the published neuron names (*Tanaka et al., 2012*); for the newly identified local neurons and for the multiglomerular projection neurons, ID numbers were sorted according to the cell body location from dorsal to ventral.

For the neurons associated with the lateral horn, we expanded the existing naming scheme (names such as PV5a1) based on the cell body cluster location (uppercase letters and first number), anatomically associated groups (lower case letter), and individual neuron type (last number), which has previously been applied for ≈30% of the lateral horn neurons (*Frechter et al., 2019*;

*Bates et al., 2020*). The neuron types that have been defined in the lateral horn sometimes contain slightly larger morphological varieties of neurons than would be categorized as different types in the hemibrain volume. To reconcile this slight discrepancy while keeping the published neuron type names as consistent as possible, in some cases we used suffices _a, _b, etc., for distinguishing not only the neurons that are different in their connectivity but also those that have minute but distinct morphological differences. Because of this technical issue more neurons are distinguished by suffices in the lateral horn than in other brain regions.

In cases where we gave new neuron names to the already known ones, or slightly modified the existing names for the sake of naming scheme consistency, we indicated the most commonly used previous names in the *notes* field, from where users can look for further synonyms using the *Virtual Fly Brain* database (http://virtualflybrain.org).

For the optic lobe neurons, we categorized only the VPNs based primarily on the specific projection patterns of their axon terminals in the central brain. Newly identified neuron types were given higher numbers than those already used (*Fischbach and Dittrich, 1989*; *Panser et al., 2016*; *Otsuna and Ito, 2006*; *Hausen, 1984*). Neurons that arborize only in the optic lobe are not classified, except for several intrinsic neurons in the lobula, because the hemibrain dataset does not provide enough information about their projection patterns in the optic lobe for conclusive cell typing.

Olfactory-, thermo-, and hygro-receptor (sensory) neurons were named according to their target glomeruli in the antennal lobe (*Fishilevich and Vosshall, 2005*; *Couto et al., 2005*; *Gallio et al., 2011*; *Enjin et al., 2016*; *Frank et al., 2017*; *Marin et al., 2020*). Some of the auditory receptor neurons (Johnston's organ neurons) were also identified, but their precise target zones in the antennal mechanosensory and motor center (*Kamikouchi et al., 2006*) were not determined because of the insufficient information in the hemibrain image volume.

The neurons associated with the ocellar ganglion (OCG), a detached ganglion just beneath the ocelli, were categorized into eight types based on the morphology of their terminals in the central brain. Precise classification of OCG neurons is not possible without the projection pattern information in the OCG. To remedy this problem the neurons that share the common projection patterns within the brain were classified as OCG1, OCG2, etc., and when the projection pattern information in the OCG is available they will be classified in more detail as OCG1A, OCG1B, etc.

Outside the heavily studied regions, and the neuron types explained above, the fly's circuits are largely composed of cells of so-far unknown type. Because such neurons, in what is called the *terra incognita* of the fly brain, account for nearly 70% of the total neuron types, it was necessary to devise a systematic naming scheme to give them names that annotate reasonable morphological characteristics and are easy to pronounce. About 40% of these neurons extend their projections to regions outside of the imaged volume of the hemibrain EM dataset, such as the contralateral brain side, the ventralmost parts of the brain, and the optic lobes. Since whole brain reconstructions of such neurons will soon become available, the naming scheme should provide reasonable names for the neurons that are not fully traceable within the hemibrain image volume.

To address this problem, we tested various naming schemes using single-cell LM images of about 500 neuron types in these regions. LM images have much lower spatial resolution but visualize entire projection patterns across the brain compared to the EM data. We found the regions (neuropils) of the central brain with the most extensive arborization by counting the voxel numbers of the three-dimensional LM data. We also simulated the numbers of output and input synapses available in the EM data by assessing the number of boutons and spines - characteristic morphology of output and input synaptic sites - in the LM images. Regions with the largest number of output synapses tend to lie on the contralateral side of the brain, out of the hemibrain volume, making it difficult to use EM information as a primary determining factor. Regions with the largest number of input synapses often showed discrepancies between EM and LM images, mainly due to the varying completeness of fine dendritic fragments in the EM data. We found the names based on the neuropils with the largest number of voxels gave the most consistent names, regardless of whether we used the information of the entire brain or only the image area that corresponds to the hemibrain volume. Because the still unmapped fragments of input dendritic arborizations are thin and tiny, with much smaller volumes compared to the already mapped major branches, we found the voxel counts of dendrites are much less affected by potential incompleteness than the counts of input synapses.

We then applied the above LM-based naming scheme to the EM data of terra incognita neurons, and found that naming based on EM voxel count matched with either the neuropils with the largest

or second-largest number of output or input synapses for more than 95% of the neuron types. For the remaining types, we took the neuropil names with the second largest voxel numbers, which resulted in near-perfect match with the neuron type name and either the region of the most major or second major output/input synapses, making the names reasonable for connectivity analysis.

There is one more factor we had to consider. Certain groups of neuron types tend to share common core projection patterns and differ slightly only in the extent of arbors in each neuropil. For functional interpretation it would be more convenient if such neurons were classified into the same category of neuropils. If we gave names simply to individual neuron types, however, such neurons tend to be scattered into various neuropil categories affected by the slight differences of arborization patterns. To address this problem, we performed NBLAST morphological clustering with a higher threshold than used for individual neuron typing, to group the neurons that share the same CBF bundle and rather similar morphology into a common neuropil category. This additional process, however, sometimes caused mismatches between the resulting neuropil name and the most major or second major output/input synapses if the arborization pattern of that neuron type deviates too much from the rest of the group. In these cases we split such neuron types from the group and assigned them into more appropriate neuropil categories.

Between 45 and 630 neuron types were assigned into each neuropil category and distinguished with three-digit ID numbers, for example SLP153 and WED048, using the standard nomenclature abbreviations of the neuropils (*Ito et al., 2014*). We gave sequential numbers to the neuron types that share the same CBF bundles and common core projection patterns so that neurons with similar appearance would be assigned similar names, as far as possible. Within each CBF group, neurons are sorted from the ones with broader and more extensive projections to the ones with restricted local arborizations. Because of this numbering scheme, broadly arborizing neurons have scattered numbers within the number range of each neuropil category, depending on the CBF groups they belong to.

## Results of cell typing

Using the workflow of *Figure 10*, we identified 22,594 neurons with 5229 morphological types and 5609 connectivity types (*Table 3*). Over 2000 of these are types with only a single instance, although presumably, for a whole brain reconstruction, most of these types would have partners on the opposite side of the brain.

*Figure 11* shows the number of distinct neuron types found in different brain regions. *Figure 12* shows the distribution of the number of neurons in each cell type.

In spite of our extensive efforts, the assignment of type names to neurons is still ongoing. Because we opted for splitting rather than lumping of hard to differentiate cell types, it is possible that some of the neuron types may be merged with others in the future. In such cases, the number that is unused after the merger should not be re-used for other later-discovered neuron types, in order to avoid confusion. There may also be cases where neuron types could be split, or that neuron types that are missing in the current brain sample might be identified in EM or LM images of other brain samples. In such cases the newly identified neurons are expected to be given numbers above the current number range.

Although cell types and names may change, and indeed have already changed between versions v1.0 and v1.1 of our reconstruction, what will not change are the unique body ID numbers given in the database that refer to a particular (traced) cell in this particular image dataset. We strongly advise that such body IDs be included in any publications based on our data to avoid confusion as cell type names evolve.

## CBLAST

As part of our effort to assign cell types, we built a tool for cell type clustering based on neuron connectivity, called CBLAST (by analogy with the existing NBLAST [*Costa et al., 2016*], which forms clusters based on the shapes of neurons). The overall flow of the tool is described in *Figure 13*, and the code and instructions on how to install and run it can be found at https://github.com/connectome-neuprint/CBLAST (*Plaza and Dreher, 2020*; copy archived at https://github.com/elifesciences-publications/CBLAST).

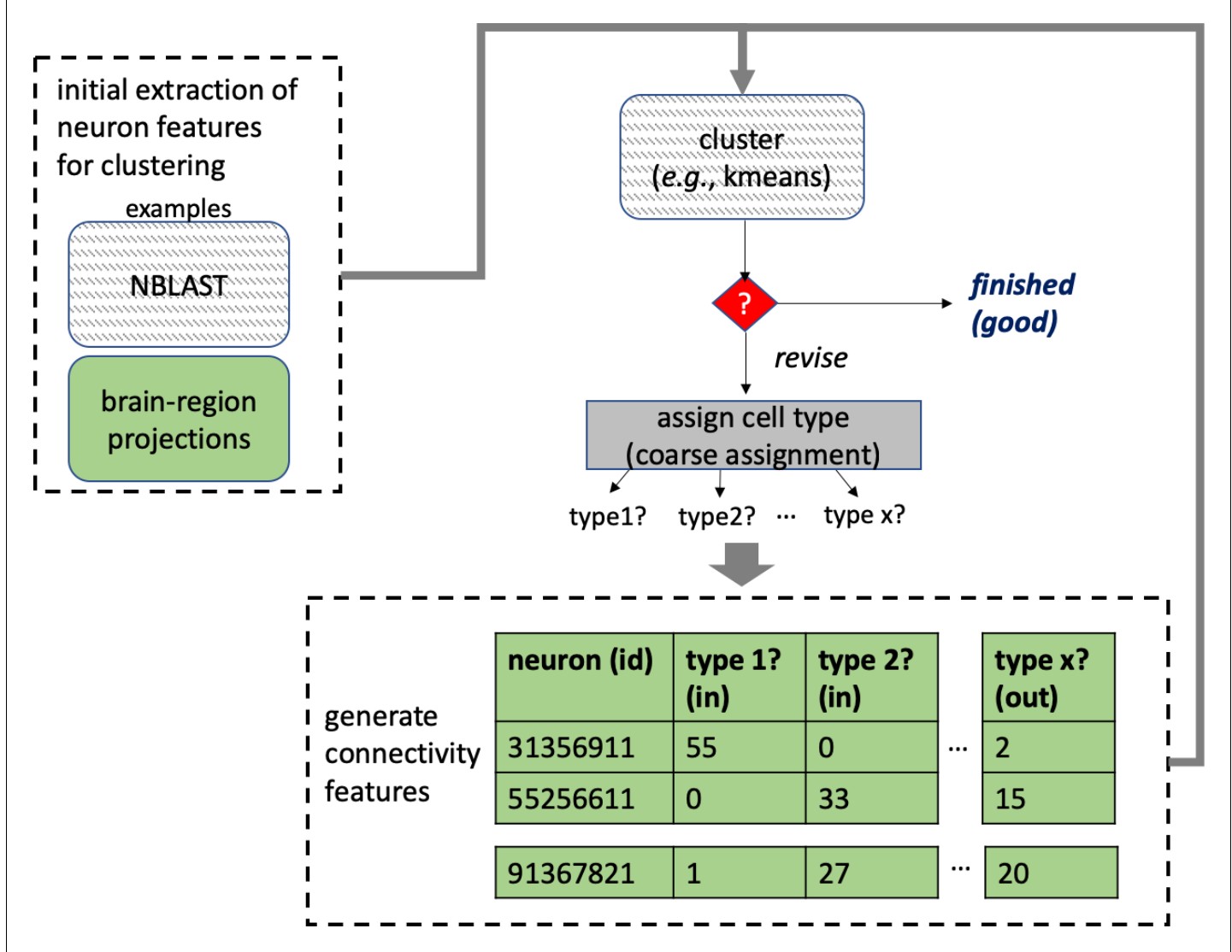

**Figure 13.** Overview of the operation of CBLAST.

Partitioning a network into clusters of nodes that exhibit similar connectivity is known as community detection or graph clustering (*Fortunato and Hric, 2016*). Numerous methods have been proposed for selecting such partitions, the best known being the stochastic block model. To non-theoreticians, the process by which most methods choose a partitioning is not intuitive, and the results are not easily interpretable. Furthermore, most approaches do not readily permit a domain expert to guide the partitioning based on their intuition or on other features of the nodes that are not evident in the network structure itself. In contrast, CBLAST is based on traditional data clustering concepts, leading to more intuitive results. Additionally, users can apply their domain expertise by manually refining the partitioning during successive iterations of the procedure. This is especially useful in the case of a network like ours, in which noise and missing data make it difficult to rely solely on connectivity to find a good partitioning automatically. Additionally, other graph clustering methods do not accommodate the notion of left-right symmetry amongst communities, a feature that is critical for assigning cell types in a connectome.

CBLAST clusters neurons together using a similarity feature score defined by how the neuron distributes inputs and outputs to different neuron types. However, this is a circular requirement since neuron types must already be defined to use this technique. CBLAST therefore uses an iterative approach, refining cell type definitions successively. Initial cell type groups are putatively defined

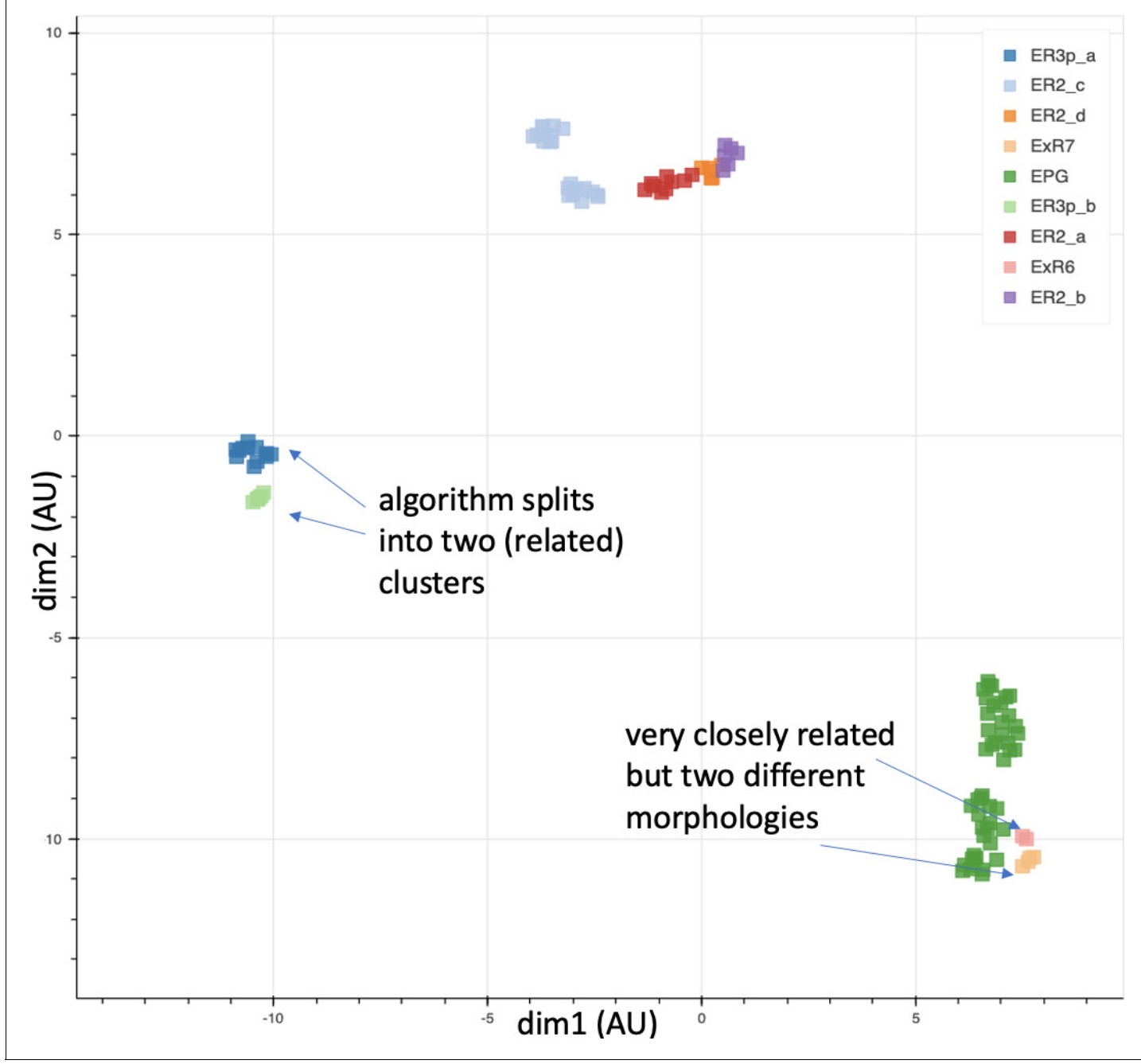

**Figure 14.** Cells of nine types plotted according to their connectivities. Coordinates are in arbitrary units after dimensionality reduction using UMAP (*McInnes et al., 2018*). The results largely agree with those from morphological clustering but in some cases show separation even between closely related types.

using an initial set of features based on morphological overlap as in NBLAST and/or based on the distribution of inputs and outputs in defined brain regions. These initial groups are fed into CBLAST in which the user can visualize and analyze the results using plots such as that in *Figure 14*. Given the straightforward similarity measure, the user can look at the input and output connections for each neuron to better understand the decision made by the clustering algorithm. As the definitions of cell type definitions are improved, the clustering becomes more reliable. In some cases, this readily exposes incompleteness (e.g., due to the boundary of the hemibrain sample) in some neurons which would complicate clustering even for more computationally intensive strategies such as a

stochastic block model. Based on these interactions, the user makes decisions and refines the clusters manually, iterating until further changes are not observed.

Our large, dense connectome is a key requirement for CBLAST. Unless a significant fraction of a neuron's inputs and outputs is known, neurons that are in fact similar may not cluster together correctly. This requirement is not absolute, as we note that CBLAST is often able to match left and right symmetric neurons, despite some of these left side neurons being truncated by the boundaries of the dataset. Nonetheless, reconstruction incompleteness and any noise in the reconstruction can contribute to noise in clustering results.

CBLAST usually generates clusters that are consistent with the morphological groupings of the neurons, with CBLAST often suggesting new sub-groupings as intended. This agreement serves as some validation of the concepts behind CBLAST. In some cases it can be preferable to NBLAST, since the algorithm is less sensitive to exact neuron location, and for many applications the connectivity is more important than the morphology. In *Figure 14*, we show the results of using CBLAST on a few neuron types extracted from the ellipsoid body. The clusters are consistent with the morphology, with exception to a new sub-grouping for R3p being suggested as a more distinct group than type ExR7/ExR6.

## Assessing morphologies and cell types

Verifying correctness and completeness in these data is a challenging problem because no existing full brain connectome exists against which our data might be compared. We devised a number of tests to check the main features: Are the morphologies correct? Are the regions and cell types correctly defined? Are the synaptic connection counts representative?

Assessing completeness is much easier than assessing correctness. Since the reconstruction is dense, we believe the census of cells, types, and regions should be essentially complete. The main arbors of every cell within the volume are reconstructed, and almost every cell is assigned a cell type. Similarly, since the identified brain regions nearly tile the entire brain, these are complete as well.

For checking morphologies, we searched for major missing or erroneous branches using a number of heuristics. Each neuron was reviewed by multiple proofreaders. The morphology of each neuron was compared with light microscopy data whenever it was available. When more than one cell of a given type was available (either left and right hemisphere, or multiple cells of the same type in one hemisphere), a human examined and compared them. This helped us find missing or extra branches, and also served as a double check on the cell type assignment. In addition, since the reconstruction is dense, all sufficiently large 'orphan' neurites were examined manually until they were determined to form part of a neuron, or they left the volume. To help validate the assigned cell types, proofreaders did pairwise checks of every neuron with types that had been similarly scored.

For subregions in which previous dense proofreading was available (such as the alpha lobes of the mushroom body), we compared the two connectomes. We were also helped by research groups using both sparse tracing in the full fly brain TEM dataset (*Zheng et al., 2018*), and our hemibrain connectome. They were happy to inform us of any inconsistencies. There are limits to this comparison, as the two samples being compared were of different ages and raised under different conditions, then prepared and imaged by different techniques, but this comparison would nevertheless have revealed any gross errors. Finally, we generated a 'probabilistic connectome' based on a different segmentation, and systematically visited regions where the two versions differed.

## Assessing synapse accuracy

As discussed in the section on finding synapses, we evaluated both precision (the fraction of found synapses that are correct) and recall (fraction of true synapses that were correctly predicted) on sample cubes in each brain region. We also double checked by comparing our findings with a different, recently published, synapse detection algorithm (*Buhmann et al., 2019*).

As a final check, we also evaluated the end-to-end correctness of given connections between neurons for different cell types and across brain regions. Specifically, for each neuron, we sampled 25 upstream connections (T-bar located within the neuron) and 25 downstream connections (PSD located within the neuron), and checked whether the annotations were correct, meaning that the pre/post annotation was valid and assigned to the correct neuron.

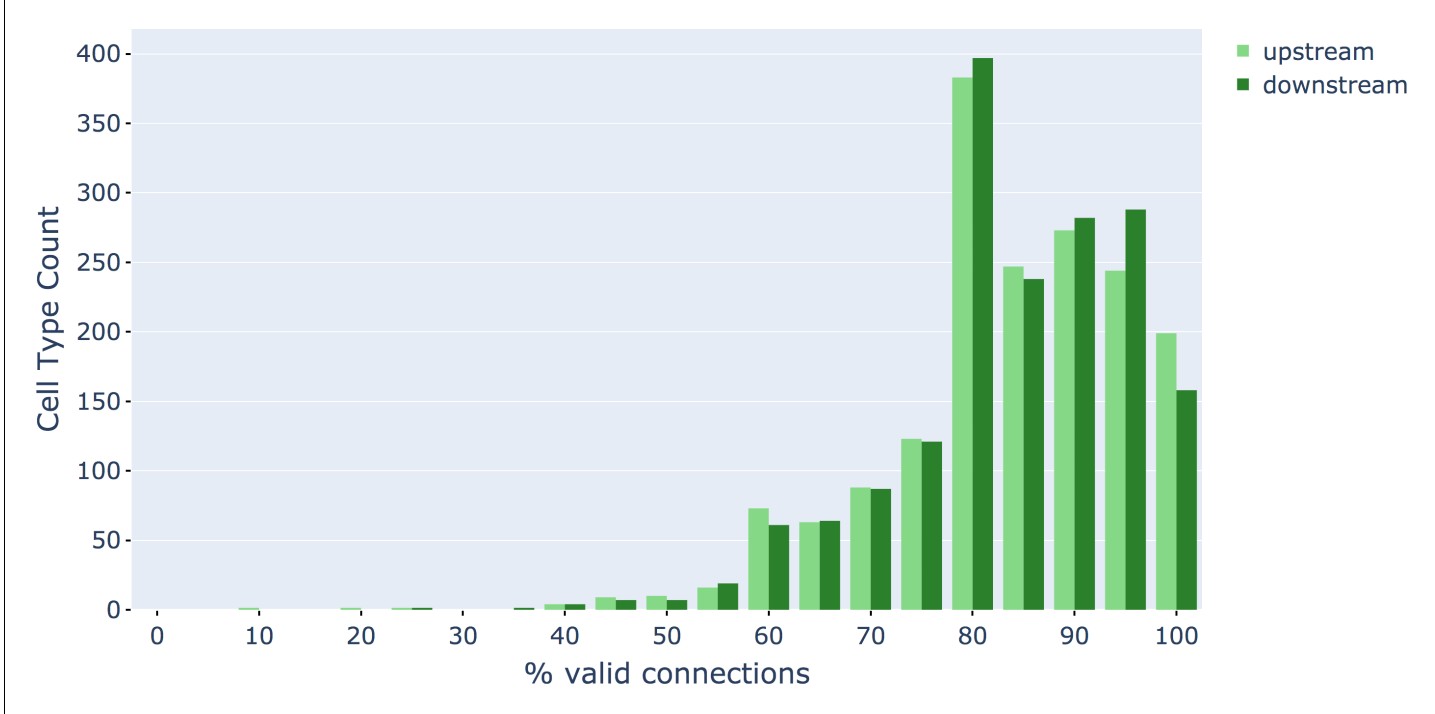

**Figure 15.** Connection precision of upstream and downstream partners for ≈ 1000 cell types. Data available in *Figure 15—source data 1*. The online version of this article includes the following source data for figure 15:

**Source data 1.** Data on 1735 neurons, one per row.

In total, we examined 1735 traced neurons spanning 1518 unique cell types (therefore examining roughly 43,000 upstream connections and 43,000 downstream connections). The histogram of synapse accuracy (end-to-end precision of predicted synapses) is given in *Figure 15*. Median precision for upstream connections, as well as for downstream connections, is 88%. Additionally, 90% of cell types have an accuracy of at least 70%. For the few worst cases, we manually refined the synapse predictions afterwards. We note that the worst outlier, having an upstream connection accuracy of 12%, is both a case involving few total connections (17 T-bars), and some ambiguity in the ground-truth decisions (whether the annotated location is an actual T-bar).

We also evaluated single-connection pathways across each brain region. In the fly, functionally important connections are thought typically to have many synapses, with the possible exception of cases where many neurons of the same type synapse onto the same downstream partner. However, the presence of connections represented by few synapses is also well known, even if the biological importance of these is less clear. Regardless, we wanted to ensure that even single connection pathways were mostly correct. We sampled over 5500 single-connection pathways, distributed across 57 brain regions. Mean synapse precision per brain region was 76.1%, suggesting that single-connection accuracy is consistent with overall synapse prediction accuracy.

We also undertook a preliminary evaluation of two-connection pathways (two synapses between a single pair of neurons). We sampled 100 such two-connection pathways within the FB. Overall synapse precision (over the 200 synapses) is 79%, consistent with the single-edge accuracy. Moreover, the results also suggest that synapse-level accuracy is largely uncorrelated with pathway/bodies, implying that the probability that both synapses in a two-connection pathway were incorrect is 4.4% $(1 - 0.79^2)$, close to the observed empirical value of 3%. (Applying a $\chi^2$ goodness of fit test with a null hypothesis of independence gives a *p* value of 0.7.)

## Assessing connection completeness

A synapse in the fly's brain consists of a presynaptic density (with a characteristic T-bar) and typically several postsynaptic partners (PSDs). The T-bars are contained in larger neurites, and most (>90%)

of the T-bars in our dataset were contained in identified neurons. The postsynaptic densities are typically in smaller neurites, and it is these that are difficult for both machine and human to connect with certainty.

With current technology, tracing all fine branches in our EM images is impractical, so we sampled among them (at completeness levels typically ranging from 20% to 85%) and traced as many as practical in the allotted time. The goal is to provide synapse counts that are representative, since completeness is beyond reach and largely superfluous. Assuming the missing PSDs are independent (which we try to verify), then the overall circuit emerges even if a substantial fraction of the connections are missing. If a connection has a synapse count of 10, for example, then it will be found in the final circuit with more than 99.9% probability, provided at least half the individual synapses are traced.

If unconnected small twigs are the main source of uncertainty in our data (as we believe to be the case), then as the proofreading proceeds the synapse counts of existing connections should only increase. Of course corrections resulting in lower synapse counts, such as correcting a false

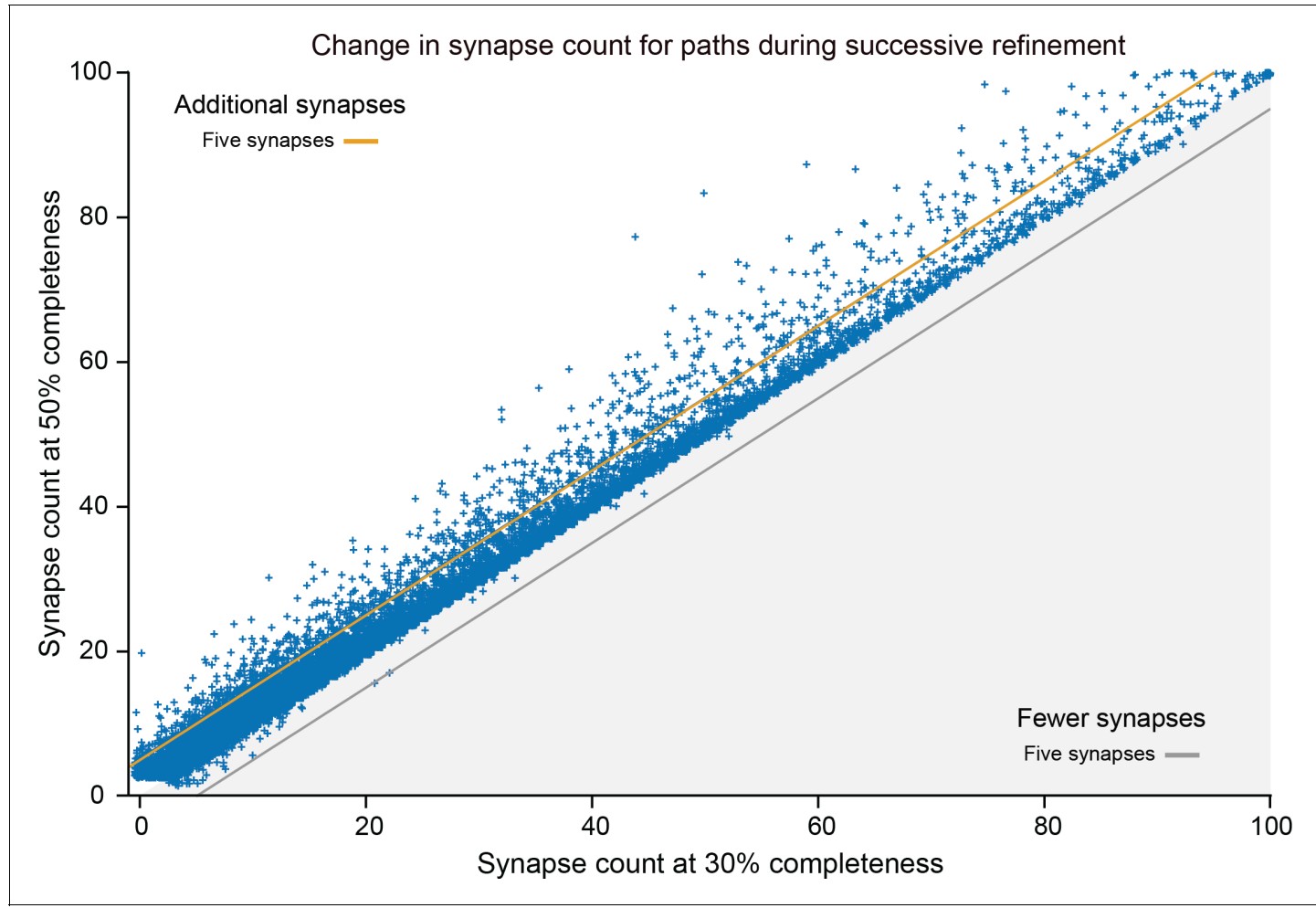

**Figure 16.** Difference between synapse counts in connections of the Ellipsoid Body, with increased completeness in proofreading. Roughly 40,000 connection strengths are shown. Almost all points fall above the line Y = X, showing that almost all connections increased in synapse count, with very few decreasing. In particular, no path decreased by more than five synapses. Only two new strong (count >10) paths were found that were not present in the original. As proofreading proceeds, this error becomes less and less common since neuron fragments (orphans) are added in order of decreasing size (see text). Data available in *Figure 16—source data 1*.

The online version of this article includes the following source data for figure 16:

**Source data 1.** Data for *Figure 16*.

connection or removing an incorrect synapse, are also possible, but are considerably less likely. To see if our proofreading process worked as expected, we took a region that had been read to a lower percentage completion and then spent the manual effort to reach a higher percentage, and compared the two circuits. (A versioned database such as DVID is enormously helpful here.) If our efforts were successful, ideally what we see is that almost all connections that changed had more synapses, very few connections got fewer synapses, and no new strong (many synapse) connections appeared (since all strong connections should already be present even in low coverage proofreading). If this is the behavior we find, we could be reasonably certain that the circuits found are representative for all many-synapse connections.

*Figure 16* shows such an analysis. The results support our view that the circuits we report reflect what would be observed if we extrapolated to assign all pre- and postsynaptic elements.

## Interpreting the connection counts

Given the complexity of the reconstruction process, and the many different errors that could occur, how confident should the user be that the returned synapse counts are valid? This section gives a quick guide in the absence of detailed investigation. The number of synapses we return is the number we found. The true number could range from slightly less, largely due to false synapse predictions, to considerably more, in the regions with low percentage reconstructed. For connections known to be in a specific brain region, the reciprocal of the completion percentage (as shown in *Table 1*) gives a reasonable estimate of the undercount.

If we return a count of 0 (the neurons are not connected), there are two cases. If the neurons do not share any brain regions, then the lack of connections is real. If they do share a brain region or regions, then a count of 0 is suspect. It is possible that there might be a weak connection (count 1–2) and less likely there is a connection of medium strength (3–9 synapses). Strong connections can be confidently ruled out, minus the small chance of a mis- or un-assigned branch with many synapses.

If we report a weak connection (1–2 synapses), then the true strength might range from 0 (the connection does not exist) through a weak connection (3–9 synapses). If your model or analysis relies on the strength of these weak connections, it is a good idea to manually check our reconstruction. If your analysis does not depend on knowledge of weak connections, we recommend ignoring connections based on three or fewer synapses.

If we report a medium strength connection (3–9 synapses) then the connection is real. The true strength could range from weak to the lower end of a strong connection.

If we report a strong connection (10 or more synapses), the connection not only exists, but is strong. It may well be considerably stronger than we report.

## Data representation

The representation of connectomics data is a significant problem for all connectomics efforts. The raw image data on which our connectome is based is larger than 20 TB, and takes 2 full days to download even at a rate of 1 gigabit/second. Looking forward, this problem will only get worse. Recent similar projects are generating petabytes worth of data (*Yin et al., 2019*), and a mouse brain of 500 mm$^3$, at a typical FIB-SEM resolution of 8 nm isotropic, would require almost 1000 petabytes.

In contrast, most users of connectivity information want a far smaller amount of much more specific information. For example, a common query is 'what neurons are downstream (or upstream) of a given target neuron?'. This question can be expressed in a few tens of characters, and the desired answer, the top few partners, fits on a single page of text.

Managing this wide range of data, from the raw gray-scale through the connectivity graph, requires a variety of technologies. An overview of the data representations we used to address these needs is shown in *Figure 17*.

This organization offers several advantages. In most cases, instead of transferring files, the user submits queries for the portion of data desired. If the user needs only a subset of the data (as almost all users do) then they need not cope with the full size of the data set. Different versions of the data can be managed efficiently behind the scenes with a versioned database such as DVID (*Katz and Plaza, 2019*) that keeps track of changes and can deliver data corresponding to any previous version. The use of existing software infrastructure, such as Google buckets or the graph package

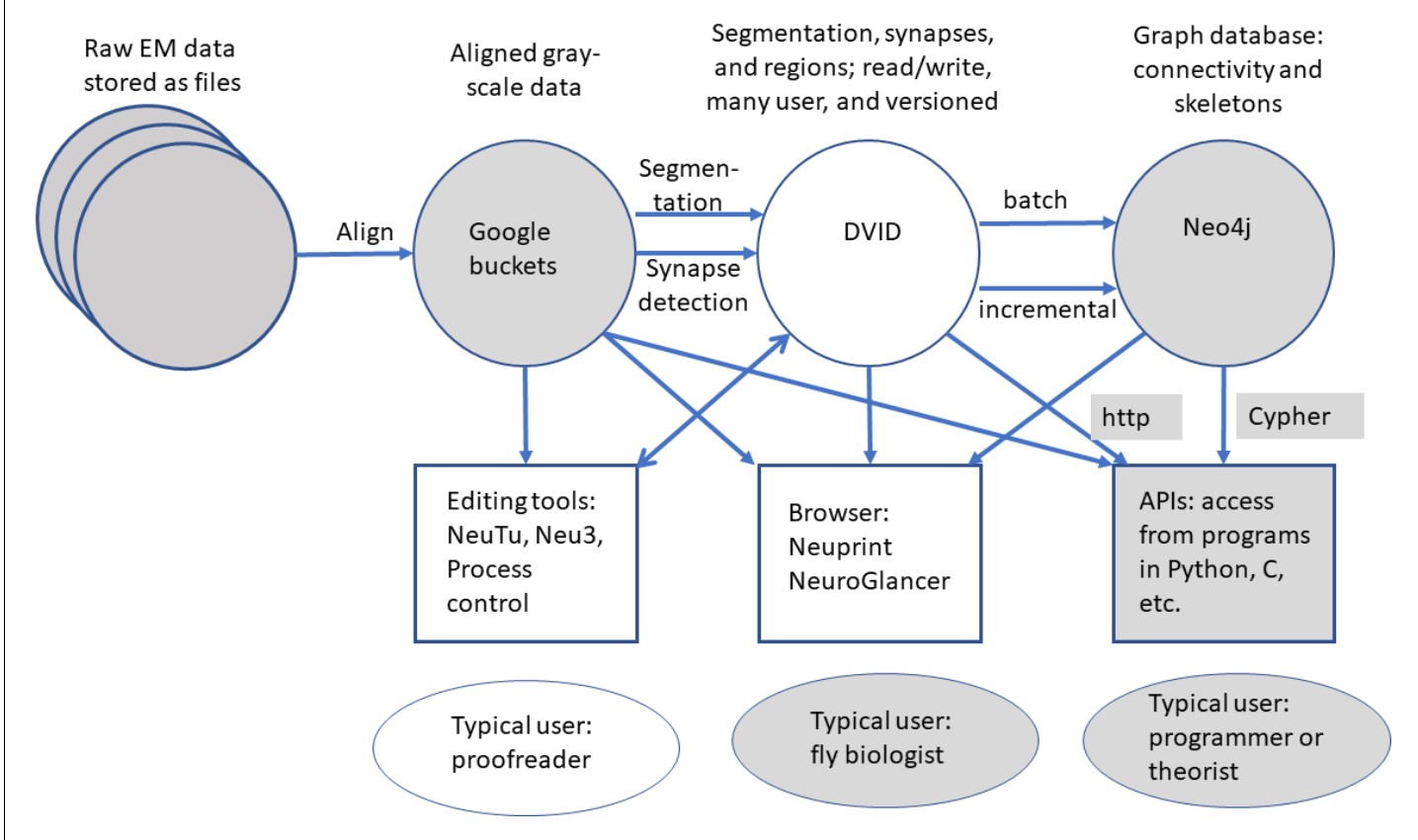

**Figure 17.** Overview of data representations of our reconstruction. Circles are stored data representations, rectangles are application programs, ellipses represent users, and arrows indicate the direction of data flow labeled with transformation and/or format. Filled areas represent existing technologies and techniques; open areas were developed for the express purpose of EM reconstruction of large circuits.

neo4j, which are already optimized for large data, helps with both performance and ease of development. The advanced user is not limited to these interfaces - for those who may wish to validate or extend our results; we have provided procedures whereby the user can make personal copies of each representation, including the grayscale, the DVID data storage, and our editing and proofreading software. These allow other researchers to establish an entirely independent version of all we have done, completely under their control. Contact the authors for the details of how to copy all the underlying data and software.

## What are the data types?

Grayscale data correspond to traditional electron microscope images. This is written only once, after alignment, but often read, because it is required for segmentation, synapse finding, and proofreading. We store the grayscale data, eight bits per voxel, in Google buckets, which facilitates access from geographically distributed sites.

Segmentation, synapses, and identifying regions annotate and give biological meaning to the grayscale data. For segmentation, we assign a 64 bit neuron ID to each voxel. Despite the larger size per voxel (64 vs 8 bits) compared with the grayscale, the storage required is much smaller (by a factor of more than 20) since segmentation compresses well. Although the voxel level segmentation is not needed for connectivity queries, it may be useful for tasks such as computing areas and cross-sections at the full resolution available, or calculating the distance between a feature and the boundary.

Synapses are stored as point annotations - one point for a presynaptic T-bar, and one point for each of its postsynaptic densities (or PSDs). The segmentation can then be consulted to find the identity of the neurons containing their connecting synapses.

The compartment map of the brain is stored as a volume specified at a lower resolution, typically a 32 × 32 × 32 voxel grid. At 8 nm voxels, this gives a 256 nm resolution for brain regions, comparable to the resolution of confocal laser scanning microscopy.

Unlike the grayscale data, segmentation, synapses, and regions are all modified during proofreading. This requires a representation that must cope with many users modifying the data simultaneously, log all changes, and be versioned. We use DVID (*Katz and Plaza, 2019*), developed internally, to meet these requirements.

Neuron skeletons are computed from the segmentation (*Zhao and Plaza, 2014*), and not entered or edited directly. A skeleton representation describes each neuron with (branching) centerlines and diameters, typically in the SWC format popularized by the simulator *Neuron* (*Carnevale and Hines, 2006*). These are necessarily approximations, since it is normally not possible (for example) to match both the cross-sectional area and the surface area of each point along a neurite with such a representation. But SWC skeletons are a good representation for human viewing, adequate for automatic morphology classification, and serve as input to neural simulation programs such as 'Neuron'. SWC files are also well accepted as an interchange format, used by projects such as NeuroMorpho (*Ascoli et al., 2007*) and FlyBrain (*Shinomiya et al., 2011*).

The connectivity graph is also derived from the data and is yet more abstract, describing only the identity of neurons and a summary of how they connect - for example, Neuron ID1 connects to neuron ID2 through a certain number of synapses. In our case, it also retains the brain region information and the location of each synapse. Such a connectivity graph is both smaller and faster than the geometric data, but sufficient for most queries of interest to biologists, such as finding the upstream or downstream partners of a neuron. A simple connectivity graph is often desired by theorists, particularly within brain regions, or when considering neural circuits in which each neuron can be represented as a single node.

A final, even more abstract form is the adjacency matrix: This compresses the connectivity between each pair of neurons to a single number. Even this most economical form requires careful treatment in connectomics. As our brain sample contains more than 25K traced neurons as well as many unconnected fragments, the adjacency matrix has more than a billion entries (most of which are zero). Sparse matrix techniques, which report only the non-zero coefficients, are necessary for practical use of such matrices.

## Accessing the data

For the hemibrain project, we provide access to the data through a combination of a software interface (*Clements et al., 2020*) and a server (https://neuprint.janelia.org, also accessible through https://doi.org/10.25378/janelia.12818645). Login is via any Google account; users who wish to remain anonymous can create a separate account for access purposes only. Data are available in the form of gray-scale, pixel-level segmentation, skeletons, and a graph representation. Two previous connectomics efforts are available as well (a seven-column optic lobe reconstruction [*Takemura et al., 2015*] and the alpha lobe of the mushroom body [*Takemura et al., 2017*]). These can be found at https://neuprint-examples.janelia.org .

The most straightforward way to access the hemibrain data is through the Neuprint (*Clements et al., 2020*) interactive browser. This is a web-based application that is intended to be usable by biologists with minimal or no training. It allows the selection of neurons by name, type, or brain region, displays neurons, their partners, and the synapses between these in a variety of forms, and provides many of the graphs and summary statistics that users commonly want.

Neuprint also supports queries from languages such as Python (*Sanner, 1999*) and R, as used by the neuroanatomy tool NatVerse (*Manton et al., 2019*). Various formats are supported, including SWC format for the skeletons. In particular, the graph data can be queried through an existing graph query language, Cypher (*Francis et al., 2018*), as seen in the example below. The schema for the graph data is shown in *Figure 18*.

```
MATCH (n:Neuron) - [c:ConnectsTo] -> (t:Neuron) WHERE t.
type = `MBON18'
RETURN n.type, n.bodyId, c.weight ORDER BY c.weight DESCENDING
```

This query looks for all neurons that are presynaptic to any neuron of type 'MBON18'. For each such neuron it returns the types and internal identities of the presynaptic neuron, and the count of

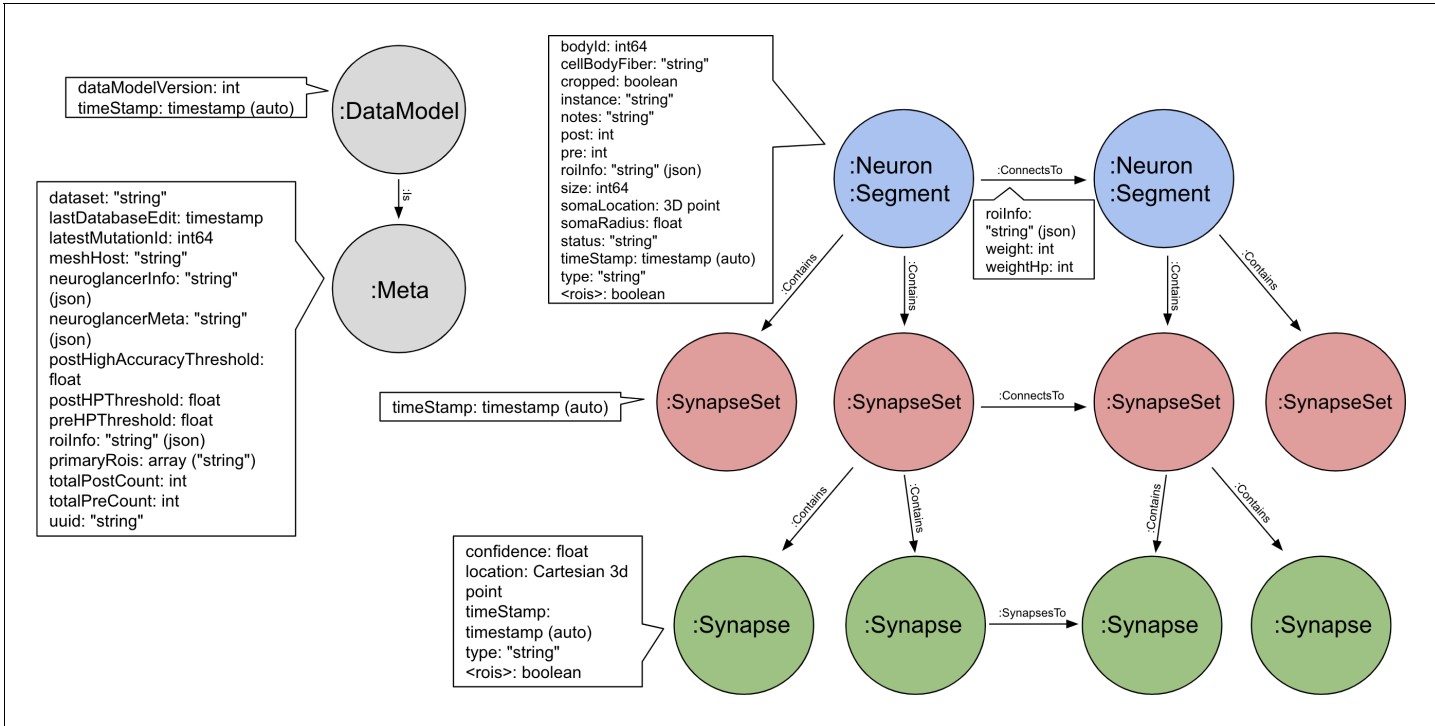

**Figure 18.** Schema for the neo4j graph model of the hemibrain. Each neuron contains 0 or more SynapseSets, each of which contains one or more synapses. All the synapses in a SynapseSet connect the same two neurons. If the details of the synapses are not needed, the neuron-to-neuron weight can be obtained as a property on the 'ConnectsTo' relation, as can the distribution of this weight across different brain regions (the roiInfo).

synapses between them. The whole list is ordered in order of decreasing synapse count. This is just an illustration for a particular query that is quite common and supported in Neuprint without the need for any programming language.

Adjacency matrices, if needed, can be derived from the graph representation. We provide a small demonstration program that queries the API and generates such matrices, either with or without the brain regions. The two matrices themselves are available in gzipped Python format.

The raw greyscale images, with overlays of segmentation and feature masks (such as glia and mitochondria), can be viewed in the publicly available tool NeuroGlancer (*Perlman, 2019*). This viewer can be selected from the Neuprint browser.

For more information on accessing data and other hemibrain updates, please see https://www.janelia.org/project-teams/flyem/hemibrain .

## Matching EM and light microscopy data

No two flies are identical, and brain samples differ in size and orientation. Furthermore, different preparation methods cause tissues to swell and shrink by varying amounts. Therefore, the first step when comparing the features of different brains is registration to a common reference frame.

Some of these differences are illustrated in *Figure 19*. Compared to the hemibrain EM data (*Figure 19(a)*), the confocal laser scanning microscopy images of the previous brain atlas (*Ito et al., 2014*) are about 17% smaller (*Figure 19(b)*), and the JRC2018 unisex template brain used for the registration of EM and light microscopy brain images (*Bogovic et al., 2020*) is about 30% smaller (*Figure 19(c)*). Since unfixed brains right after dissection in saline are 15–20% larger than the antibody-labeled brains mounted in 80% glycerol – similar to *Figure 19(b)* – a raw female brain will be nearly the same size as the hemibrain EM stack.

The orientation of the brain samples may also vary. There is about 18.5˚ of tilt between the hemibrain EM stack and the 2014 brain atlas, and about 14˚ of tilt between hemibrain EM and the JRC2018 template. To create matching vertical or horizontal sections, therefore, each image stack should be re-sliced after applying the corresponding rotation.

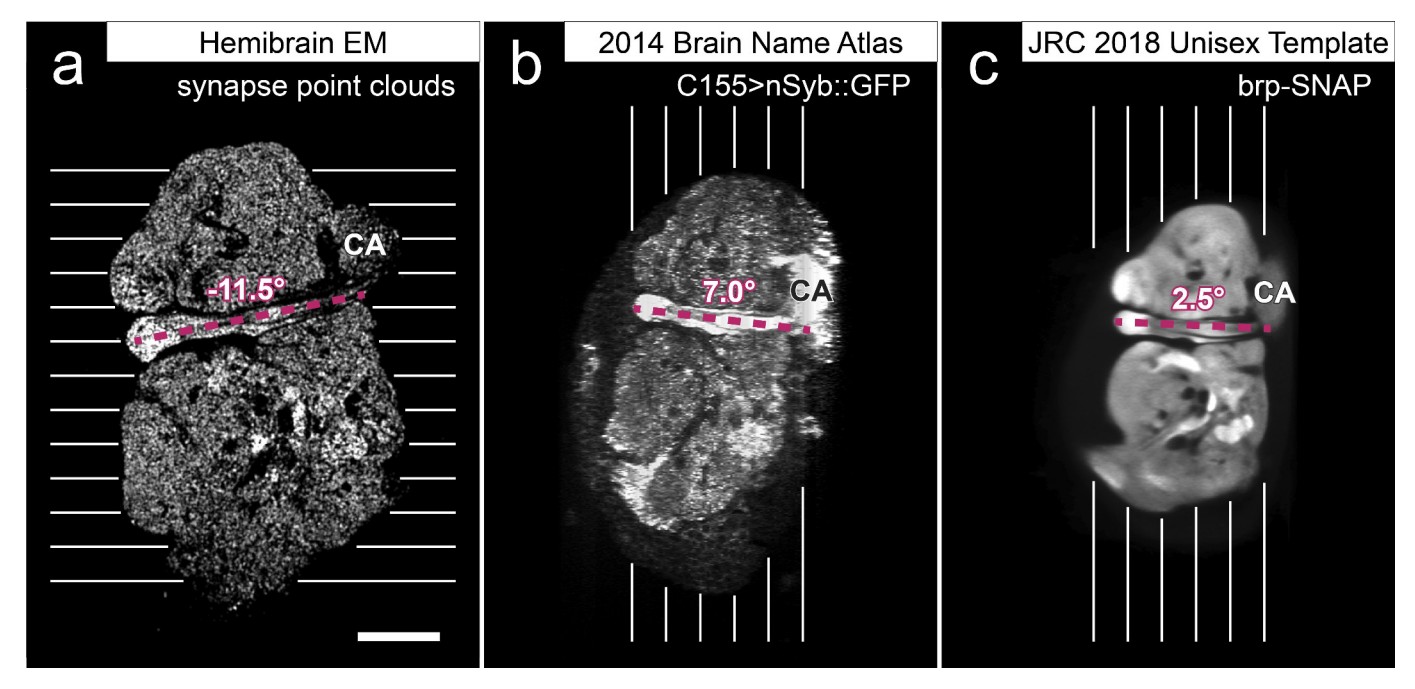

**Figure 19.** Comparison of the size and orientation of brain images. Sagittal section images at the plane of the mushroom body pedunculus are shown. Parallel lines indicate the direction of serial sectioning. Purple dotted lines indicate the axes of the pedunculus to show the sample orientation. Numbers indicate the angles of the pedunculus axes relative to the horizontal axis. Scale bar: 50 μm for all images. CA: calyx of the mushroom body. Panel (a) Hemibrain EM image stack. Grayscale indicates the density of the points of the presynaptic T-bars (point clouds). (b) Confocal light microscopy image stack provided by the Insect Brain Name Working Group (*Ito et al., 2014*), of a female brain mounted in 80% glycerol after antibody labeling. Presynaptic sites are labeled by GFP fused with the synaptic vesicle-associated protein neuronal synaptobrevin (nSyb), driven by the pan-neuronal expression driver line elav-GAL4 C155. (c) JRC2018 Unisex brain template (*Bogovic et al., 2020*), which is an average of 36 female and 26 male brains mounted in DPX plastic after dehydration with ethanol and clearization with xylene. Presynaptic sites are labeled with the SNAP chemical tag knock-in construct inserted into the genetic locus of the active zone protein bruchpilot (brp). The relative sizes of the brains, measured as the height along the lines that are perpendicular to the pedunculus axes, are 100:83:70 for (a), (b), and (c). These differences in size and orientation must be taken into account when comparing the sections and reconstructed neurons of the hemibrain EM and registered light microscopy images.

The raw EM data, segmentation, and skeletons (as displayed in Neuprint) were all computed in a reference frame corresponding to *Figure 19(a)*, whereas the light lines and tools such as Color_-MIP_mask search (*Otsuna et al., 2018*) use the reference frame of *Figure 19(c)*. Therefore, registration is required to map between the EM and light representations.

We registered the hemibrain EM data to the JRC2018 *Drosophila* template brain using an automatic registration algorithm followed by manual correction. We began by using the automated T-bar predictions (described in section on synapse prediction) to generate a T-bar density volume rendered at a resolution comparable to those from light microscopic images. This hemibrain synapse density volume was automatically registered to the template brain using elastix (*Klein et al., 2010*). The resulting registration was manually fine-tuned using BigWarp (*Bogovic et al., 2016*). The total transform is the composition of the elastix and BigWarp transformations, and can be found at https://www.janelia.org/open-science/jrc-2018-brain-templates. We estimated a corresponding inverse transformation and make that available as well.

Using these transformations, an implementation that matches EM to light lines, and vice versa, is publicly accessible at https://neuronbridge.janelia.org/. This matching software is accessible directly from the Neuprint browser, where it can be launched from the tabular display of selected neurons. For those not familiar with NeuronBridge, an explanatory video explains the matching process. The details of the underlying algorithm will be covered in a separate paper by Otsuna et al., but are briefly sketched here.

If starting from an EM neuron of interest, researchers can use NeuronBridge to identify GAL4 lines labeling that neuron. First, the EM representation of the neuron is spatially transformed into

the JRC 2018 unisex template space where GAL4 driver line images are registered. The EM neuron is then used to create a mask (*Otsuna et al., 2018*) that narrows the search space considerably, making it easier to find corresponding neurons even in crowded GAL4 driver line images.

The opposite direction, finding an EM neuron that corresponds to a light neuron, is also supported. In this case the scoring of a potential match must be modified, since the light image contains the entire neuron, but many EM neurons are trunctated by the limits of our reconstructed volume. Both of these cases are discussed in the upcoming paper, with examples.

As another option, since hemibrain neurons are skeletonized, users can query GAL4 neuronal skeleton databases using NBLAST (*Costa et al., 2016*).

## Longer term storage of data, and archival references

Historically, archival data from biology data have been expressed as files that are included with supplementary data. However, for connectivity data this practice has two main problems. First, the data are large, and hard to store. Journals, for example, typically limit supplemental data to a few 10s of megabytes. The data here are about 6 orders of magnitude larger. Second, connectome data are not static, during proofreading and even after initial publication. As proofreading proceeds, the data improve in their completeness and quality. The question then is how to refer to the data as they existed at some point in time, required for reproducibility of scientific results. If represented as files, this would require many copies, checkpointed at various times - the 'as submitted' version, the 'as published' version, the 'current best version', and so on.

We resolve this, at least for now, by hosting the data ourselves and making them available through query mechanisms. Underlying our connectome data is a versioned database (DVID) so it is technically possible to access every version of the data as it is revised. However, as it requires effort to host and format this data for the Neuprint browser and API, only selected versions (called named versions) are available by default from the website, starting with the initial versions, which are 'hemibrain:v1.0' and the much improved 'hemibrain:v1.1'. Since multiple versions are available, when reproducibility is required (such as when referencing the data in a paper) it is best to refer explicitly to the version used by name (such as 'hemibrain:v1.1') because we expect new milestone versions every few months, at least at first. We will supply a DOI for each of these versions, and each is archived, can be viewed and queried through the web browser and APIs at any time, and will not change.

The goal of multiple versions is that later versions should be of higher quality. Towards this end we have implemented several systems for reporting errors so we can correct them. Users can add annotations in NeuroGlancer (*Perlman, 2019*), the application used in conjunction with Neuprint to view image data, where they believe there are such errors. To make this process easier, we provide a video explaining it. We will review these annotations and amend those that we agree are problems. Users can also contact us via email about problems they find.

Archival storage is an issue since, unlike genetic data, there is not yet an institutional repository for connectomics data and the data are too large for journals to archive. We pledge to keep our data available for at least the next 10 years.

## Analysis

Of necessity, most previous analyses have concentrated on particular circuits, cell types, or brain regions with relevance to specific functions or behaviors. For example, a classic paper about motifs (*Song et al., 2005*) sampled the connections between one cell type (layer five pyramidal neurons) in one brain region (rat visual cortex), and found a number of non-random features, such as over-represented reciprocal connections and a log-normal strength distribution. However, it has never been clear which of these observations generalize to other cell types, other brain regions, and the brain as a whole. We are now in a position to make much stronger statements, ranging over all brain regions and cell types.

In addition, many analyses are best performed (or can only be performed) on dense connectomes. Type-wide observations depend on a complete census of that cell type, and depending on the observation, a complete census of upstream and downstream partners as well. Some analyses, such as null observations about motifs (where certain motifs do not occur in all or portions of the fly's brain) can only be undertaken on dense connectomes.

**Table 4.** Regions with minimum or maximum characteristics, picked from those regions lying wholly within the reconstructed volume and containing at least 100 neurons.
Yellow indicates a minimum value; blue a maximal value. Volume is in cubic microns. N is the number of neurons in the region, L the number of connections between those neurons, ⟨k⟩ the average number of partners (in the region), D the network diameter (the maximum length of the shortest path between neurons), ⟨str⟩ the average connection strength, broken up into non-reciprocal and reciprocal. fracR is the fraction of connections that are reciprocal, and AvgDist the average number of hops (one hop corresponding to a direct synaptic connection) between any two neurons in the compartment.

| Name | Volume | N | L | ⟨k⟩ | D | ⟨str⟩ | ⟨non-r⟩ | ⟨r⟩ | fracR | AvgDist |
|---|---|---|---|---|---|---|---|---|---|---|
| MB(R) | 309371 | 3514 | 574732 | 163.555 | 8 | 3.275 | 3.081 | 3.388 | 0.632 | 2.215 |
| bL(R) | 29695 | 1171 | 108250 | 92.442 | 8 | 2.019 | 1.856 | 2.122 | 0.613 | 2.090 |
| EB | 93932 | 555 | 58789 | 105.926 | 5 | 10.087 | 4.610 | 12.215 | 0.720 | 1.798 |
| AB(L) | 526 | 100 | 1250 | 12.500 | 4 | 2.182 | 1.765 | 2.687 | 0.453 | 1.938 |
| PLP(R) | 367711 | 6913 | 244182 | 35.322 | 15 | 2.791 | 2.479 | 3.866 | 0.225 | 3.148 |
| SNP(R) | 1076257 | 9130 | 811279 | 88.859 | 13 | 3.026 | 2.552 | 4.539 | 0.239 | 2.724 |
| RUB(L) | 834 | 128 | 623 | 4.867 | 6 | 7.313 | 2.766 | 20.253 | 0.260 | 2.727 |
| EPA(R) | 29947 | 1483 | 18848 | 12.709 | 13 | 2.224 | 2.152 | 2.700 | 0.131 | 3.471 |

## Compartment statistics

One analysis enabled by a dense whole-brain reconstruction involves the comparison between the circuit architectures of different brain areas within a single individual.

The compartments vary considerably. *Table 4* shows the connectivity statistics of compartments that are completely contained within the volume, have at least 100 neurons, and have the largest or smallest value of various statistics. Across regions, the number of neurons varies by a factor of 74, the average number of partners of each neuron by a factor of 36, the network diameter (defined as the maximum length of the shortest path between any two neurons) by a factor of 4, the average strength of connection between partner neurons by a factor of 5, and the fraction of reciprocal connections by a factor of 5. The average graph distance between neurons is more conserved, differing by a factor of only 2.

## Paths in the fly brain are short

Neurons in the fly brain are tightly interconnected, as shown in *Figure 20*, which plots what fraction of neuron pairs are connected as a function of the number of interneurons between them. Three quarters of all possible pairs are connected by a path with fewer than three interneurons, even when only connections with ≥5 synapses are included. If weaker connections are allowed, the paths become shorter yet. These short paths and tight coupling are very different from human designed systems, which have much longer path lengths connecting node pairs. As an example, a standard electrical engineering benchmark (S38584 from *Brglez et al., 1989*) is shown alongside the hemibrain data in *Figure 20A–B*. The connection graph for this example has roughly the same number of nodes as the graph of the fly brain, but pair-to-pair connections involve paths more than an order of magnitude longer – a typical node pair is separated by 60 intervening nodes. This is because a typical computational element in a human designed circuit (a gate) connects only to a few other elements, whereas a typical neuron receives input from, and sends outputs to, hundreds of other neurons.

## Distribution of connection strength

The distribution of connection strengths has been studied in mammalian tissue, looking at specific cell types in specific brain areas. These findings, such as the log-normal distribution of connection strengths in rat cortex, do not appear to generalize to flies. Assuming the strength of a connection is proportional to the number of synapses in parallel, we can plot the distribution of connection strengths, summing over the whole central brain, as shown in *Figure 21*. We find a nearly pure power law with an exponential cutoff, very different from the log-normal distribution of strengths found by *Song et al., 2005* in pyramidal cells in the rat cortex, or the bimodal distribution found for

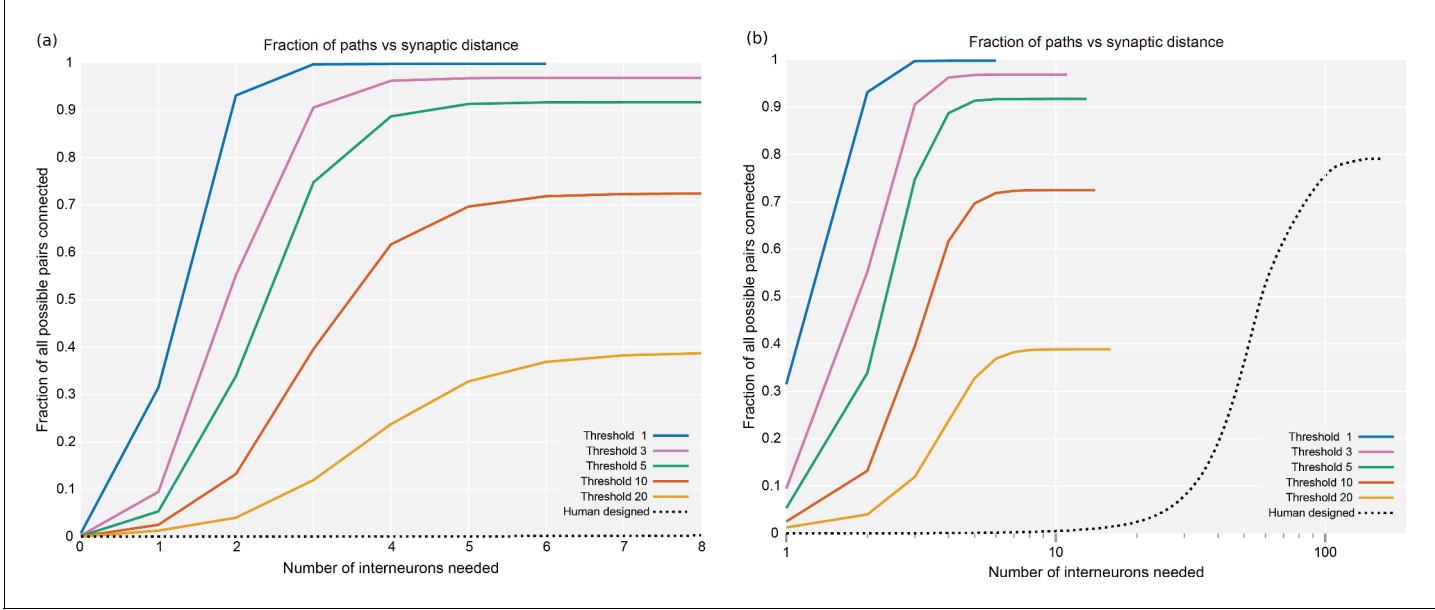

**Figure 20.** Plots of the percentage of pairs connected (of all possible) versus the number of interneurons required. (**a**) It shows the data from the whole hemibrain, for up to eight interneurons. (**b**) It is a much wider view of the same data, shown on a log scale so the curve from a human designed system is visible. Data available in *Figure 20—source datas 1–6*.

The online version of this article includes the following source data for figure 20:

**Source data 1.** Data for threshold 1 trace.
**Source data 2.** Data for threshold 3.
**Source data 3.** Data for threshold 5 trace.
**Source data 4.** Data for threshold 10 trace.
**Source data 5.** Data for threshold 20 trace.
**Source data 6.** Data for human designed trace.

pyramidal cells in the mouse by *Dorkenwald et al., 2019*. However, we caution that these analyses are not strictly comparable. Even aside from the very different species examined, the three analyses differ. Both Song and Dorkenwald looked at only one cell type, with excitatory connections only, but one looked at electrical strength while the other looked at synapse area as a proxy for strength. In our analysis, we use synapse count as a proxy for connection strength, and look at all cell types, including both excitatory and inhibitory synapses.

## Small motifs

As mentioned earlier, there have been many studies of small motifs, usually involving limited circuits, cell types, and brain regions. We emphatically confirm some traditional findings, such as the over-representation of reciprocal connections. We observe this in all brain regions and among all cell types, confirming similar findings in the antennal lobe (*Horne et al., 2018*). This can now be assumed to be a general feature of the fly's brain, and possibly all brains. In the fly, the incidence varies somewhat by compartment, however, as shown in *Table 4*.

## Large motifs

We define a large motif as a graph structure that involves every cell of an abundant type (N ≥ 20). The most tightly bound motif is a clique, in which every cell of a given type is connected to every other cell of that type, with synapses in both directions. Such connections, as illustrated in *Figure 22 (a)*, are extremely unlikely in a random wiring model. Consider, for example, the clique of ER4d cells found in the ellipsoid body, as shown in *Table 5*. In the ellipsoid body, two cells are connected with an average probability of 0.19. Therefore, the odds of finding all 600 possible connections between ER4d cells, assuming a random wiring model, is $0.19^{600} \approx 10^{-432}$.

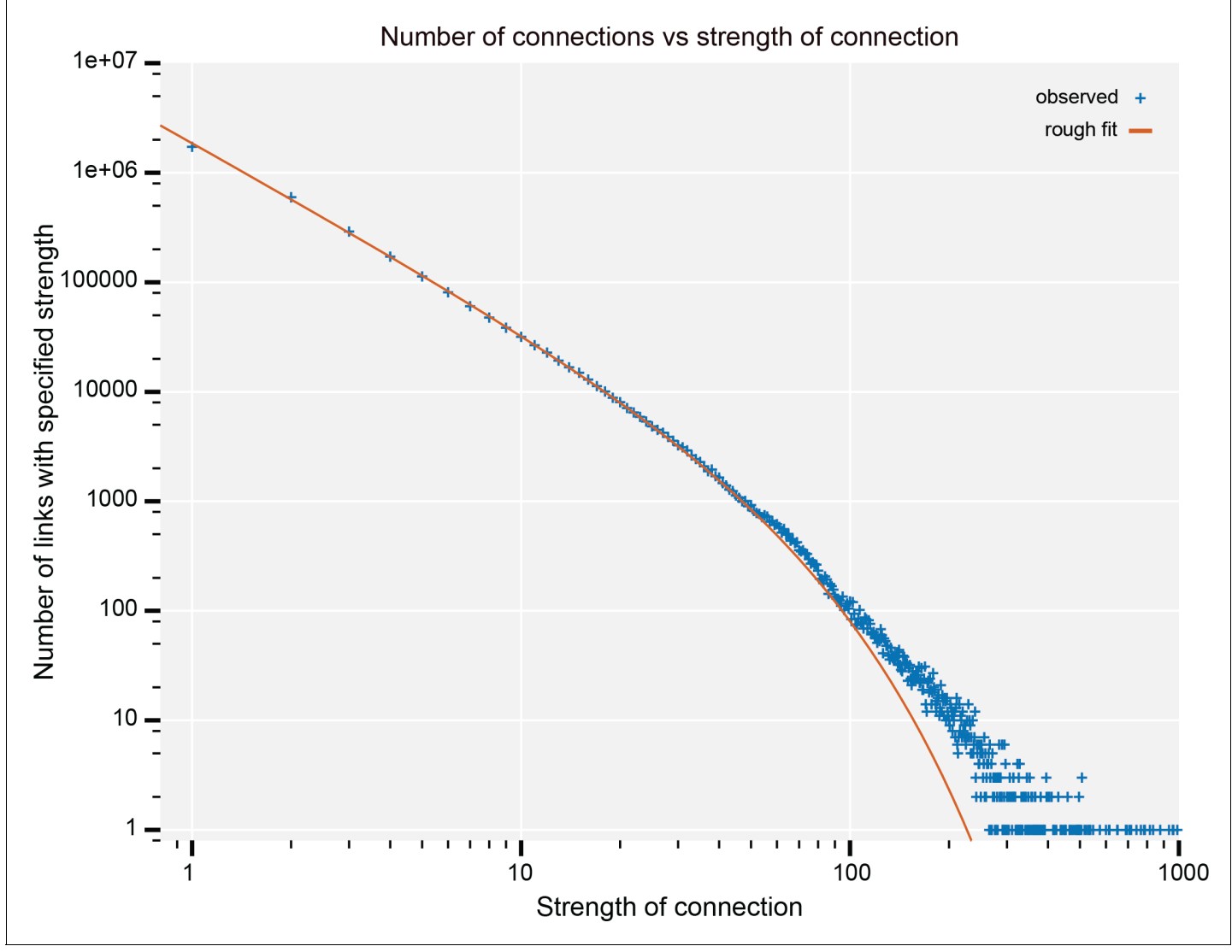

**Figure 21.** The number of connections with a given strength. Up to a strength of 100, this is well described by a power law (exponent −1.67) with exponential cutoff (at N = 42). Data available in *Figure 21—source data 1*.
The online version of this article includes the following source data for figure 21:

**Source data 1.** Data for *Figure 21*.

In the fly's brain, only a few cell types form large cliques, as shown in *Table 5*. All true cliques are among the ring neurons in the central complex, with a near-clique among the KCab-p cells of the mushroom body. The cell types PFNa and PFNd are included although they do not form a clique as shown in *Figure 22(a)*. However, these neurons are part of symmetrical structures, the noduli, that occur on both sides of the brain. Within each side, the cells form a clique, as shown in *Figure 22(d)*. The cliques within the central complex, and their potential operation, are discussed in detail in the companion paper on the central complex by Jayaraman et al.

The next most tightly bound motifs are individual cells that connect both to and from all cells of a given type, but are themselves of a different type. This is illustrated in *Figure 22(b)*. Such a motif is often speculated to be a gain or sparseness controlling circuit, where the single neuron reads the collective activation of a population and then controls their collective behavior. A well-known example is the APL neuron in the mushroom body, which connects both to and from all the Kenyon cells, and is thought to regulate the sparseness of the Kenyon cell activation (*Lin et al., 2014*).

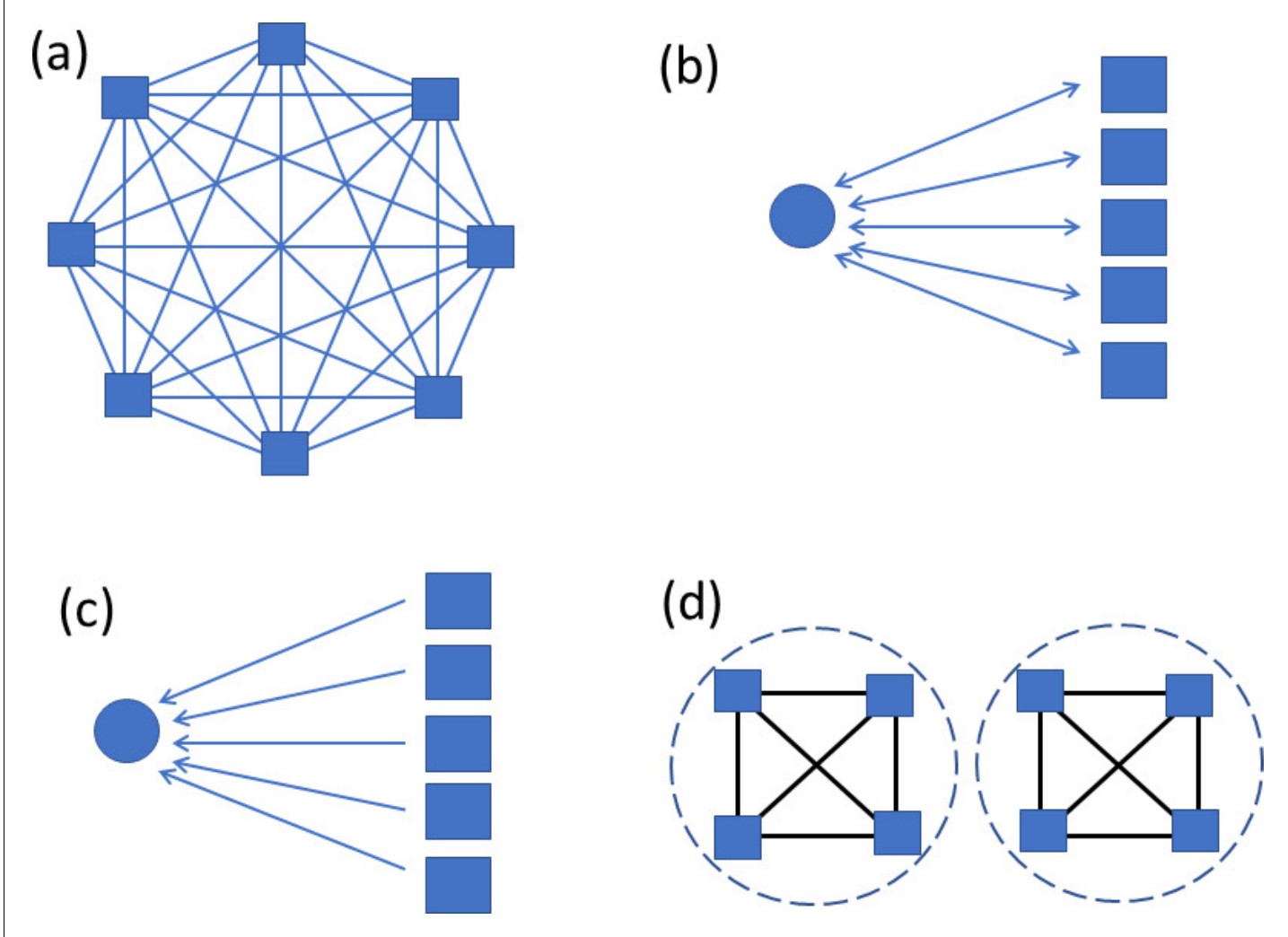

**Figure 22.** Large motifs searched for. Squares represent abundant types with at least 20 instances. Circles represent sparse types with at most two instances. Panel (a) shows a clique, where all possible connections are present. (b) It shows bidirectional connections between a sparse type and all instances of an abundant type. (c) It shows unidirectional connections from all of an abundant type to a sparse type. Panel (d) illustrates a cell type that does not form a clique overall, but does within each of two compartments.

We search for this motif by looking at cells with few instances (one or two) connecting bidirectionally to almost all cells (at least 90%) of an abundant type (N ≥ 20). We find this motif in three regions of the brain – it is common in the CX (73 different cells overseeing 22 cell types), the optic lobe circuits (19 cells overseeing 14 types), and somewhat in the MB (12 types overseeing nine types). A spreadsheet containing these cell types, who they connect to, and the numbers and strengths of their connections is described in the appendix and included as supplementary data. We only analyze the optical circuits here, since the mushroom body and central complex are the subjects of companion papers. We observe three variations on this motif - a single cell connected to all of a type (*Figure 23(a)*, found five times), a single cell with bidirectional connections to many types (*Figure 23(b)*, found once), and multiple cells all connected bidirectionally to a single type (*Figure 23(c)*), found three times. We find one circuit that is a combination: There is one cell that connects bidirectionally to all the LC17 neurons, and then a higher order cell that connects bidirectionally to a larger set (LPLC1, LPLC2, LLP1, LPC1, and LC17). In this case, these are all looming-sensitive cells and hence these circuits may regulate the features of the overall looming responses. It is tempting to speculate that the more complex structures of *Figure 23 (b) and (c)* arose from the simpler structures of (a)

**Table 5.** Cell types that form cliques and near-cliques in the hemibrain data.
To be included, a cell type must have at least 20 cell instances, 90% or more of which have bidirectional connections to at least 90% of cells of the same type. Coverage is the fraction of all possible edges in the clique that are present with any synapse count >0. Average strength is the average number of synapses in each connection. Synapses is the total number of synapses in the clique.

| Type | Region | Cells | Coverage | Avg. strength | Synapses |
|---|---|---|---|---|---|
| KCab-p | MB | 59/60 | 3455/3540 | 5.13 | 17722 |
| Delta7 | PB, CX | 42/42 | 1719/1722 | 14.21 | 24433 |
| ER2_c | EB, CX | 21/21 | 420/420 | 33.76 | 14180 |
| ER3w | EB, CX | 20/20 | 380/380 | 28.00 | 10639 |
| ER4d | EB, CX | 25/25 | 600/600 | 54.94 | 32961 |
| ER5 | EB, CX | 20/20 | 380/380 | 26.61 | 10111 |
| PFNa | NO(R) | 29/29 | 811/812 | 6.74 | 5467 |
| PFNa | NO(L) | 29/29 | 811/812 | 7.22 | 5858 |
| PFNd | NO(R) | 20/20 | 377/380 | 7.69 | 2899 |
| PFNd | NO(L) | 20/20 | 378/380 | 7.60 | 2874 |

through cell type duplication followed by divergence, but the connectomes of many more related species will be needed before this argument could be made quantitative.

The least tightly bound large motif is a cell that connects either to or from (but not both) all cells of a given type, as shown in *Figure 22(c)*. Examples include the mushroom body output neurons (*Takemura et al., 2017*). This is a very common motif, found in many regions. We find more than 500 examples of this in the fly's brain.

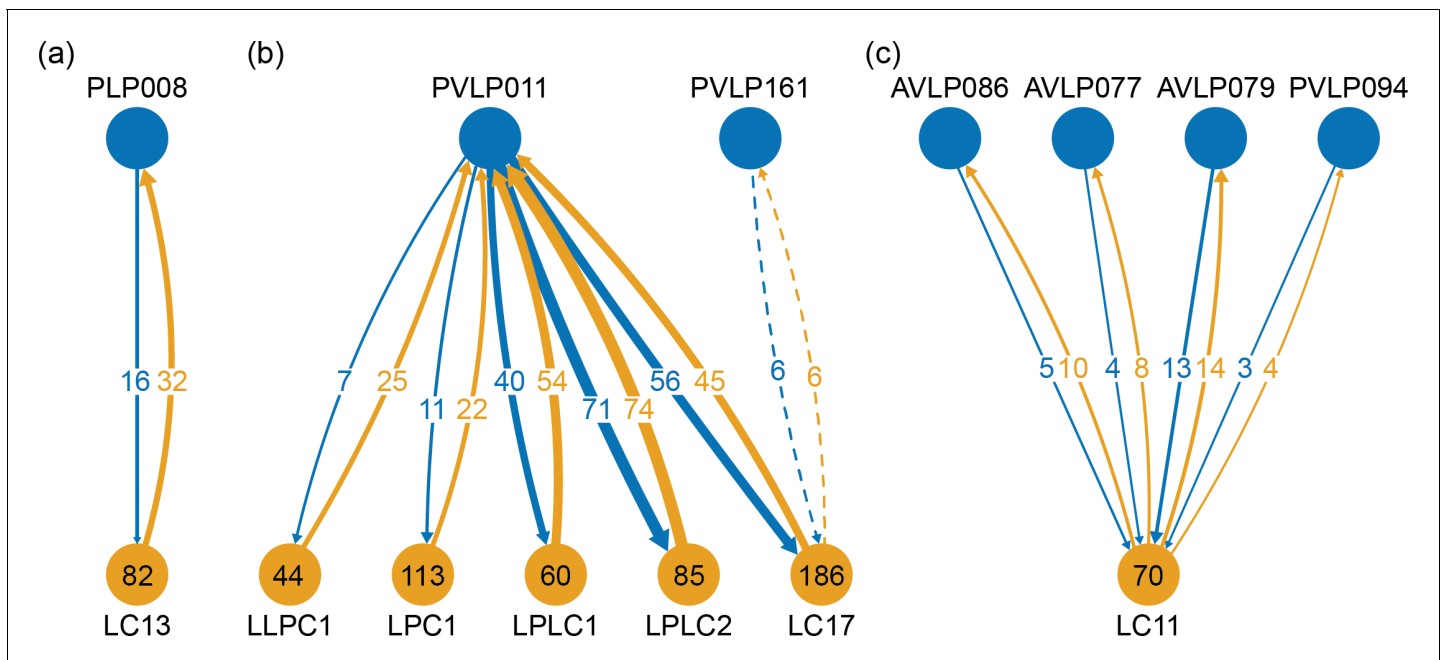

**Figure 23.** One to many motifs found in the optic circuits. Cell types consisting of a single cell, or a left-right pair, are shown at the top of the diagram. Corresponding cell type, each with many instances, are shown at the bottom of the diagram, with the number of cells per type shown inside. The arrows show the average count of synaptic connections per one cell of the bottom group. (a) An example of the most common case is shown. Here one cell, PLP008, has bidirectional connections to all 82 cells of type LC13. (b) It shows a single cell with exhaustive connections to several types. (c) It shows an alternative motif where several cells form these one-to-many connections.

## Brain regions and electrical response

How does the compartmentalization of the fly brain affect neural computation? In a few cases this has been established. For example, the CT1 neuron performs largely independent computations in each branch (*Meier and Borst, 2019*), whereas estimates show that within the medulla, the delays within each neuron are likely not significant for single column optic lobe neurons, and hence the neurons likely perform only a single computation (*Takemura et al., 2013*). Similarly, compartments of PEN2 neurons in the protocerebral bridge have been shown to respond entirely differently from their compartments in the ellipsoid body (*Green et al., 2017*; *Turner-Evans et al., 2019*).

Our detailed skeleton models allow us to construct electrical models of neurons. (In what follows, we use the word 'compartment' to mean a named physical region of the brain, as shown in *Table 1*, as opposed to the electrical sub-divisions used in simulation.) In particular, to look more generally at the issues of intra– vs inter–compartment delays and amplitudes, we can construct a linear passive model for each neuron. Our method is similar to that elsewhere (*Segev et al., 1985*), except that instead of using right cylinders, we represent each segment of the skeleton as a truncated cone. This is then used to derive the axonic resistance, the membrane resistance, and membrane capacitance for each segment. To analyze the effect of compartment structure on neuron operation, we inject the neuron at a postsynaptic density (input) with a signal corresponding to a typical synaptic input (1 nS conductance, 1 ms width, 0.1 ms rise time constant, 1 ms fall time constant, 60 mV reversal potential). We then compute the response at each of the T-bar sites (outputs). Since the synapses, both input and output, are annotated by the brain region that contains them, this allows us to calculate the amplitudes and delays from each synapse (or a sample of synapses) in each compartment to each output synapse in all other compartments.

In general, we find the compartment structure of the neuron is clearly reflected in the electrical response. Consider, for example, the EPG neuron (*Figure 24(a)*) with arbors in the ellipsoid body, the protocerebral bridge, and the gall (the gall is a sub-compartment of the LAL, the lateral accessory lobe). *Figure 25(a)* shows the responses to synaptic input in the gall. Within the gall, the delays are very short, and the amplitude relatively high and variable, depending somewhat on the input and output synapse within the gall. From the gall to other regions, the delays are longer (typically a few milliseconds) and the amplitudes much smaller and nearly constant, largely independent of the exact transmitting and receiving synapse. There is a very clean separation between the within-

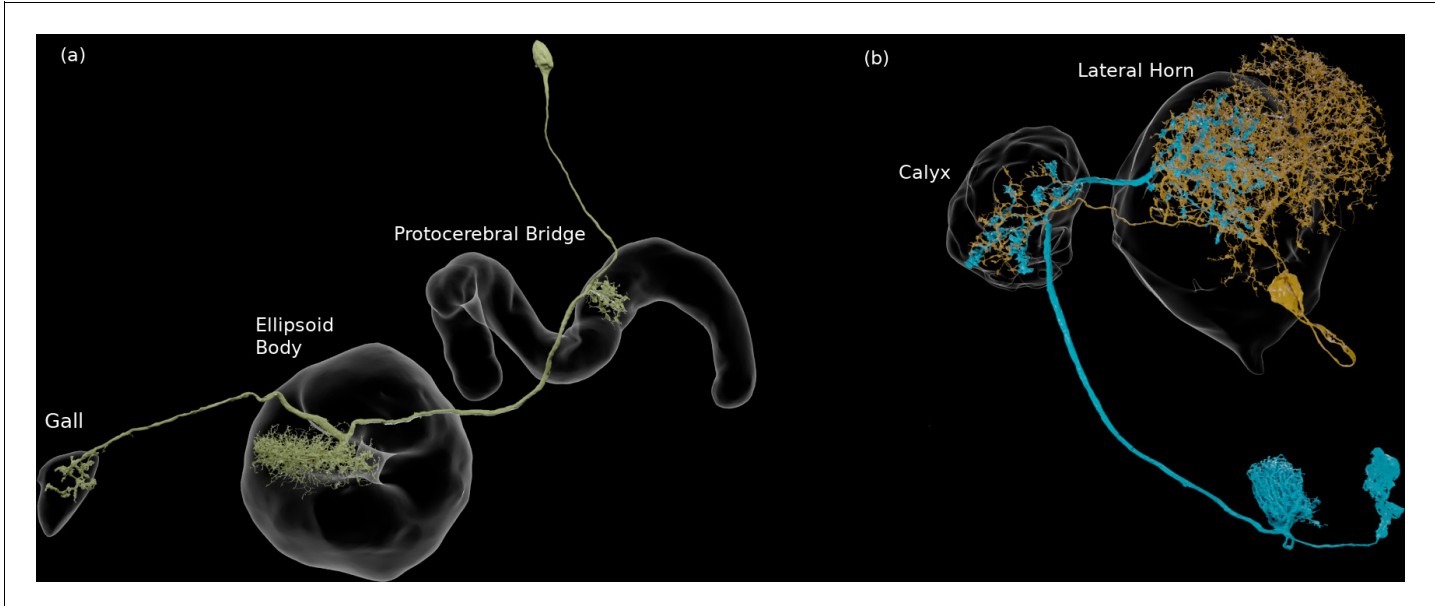

**Figure 24.** Neural connection patterns. (a) An EPG neuron, with arbors in three compartments. (b) Two neurons that connect in more than one compartment, in this case the calyx and the lateral horn. They are each pre- and postsynaptic to each other in both compartments.

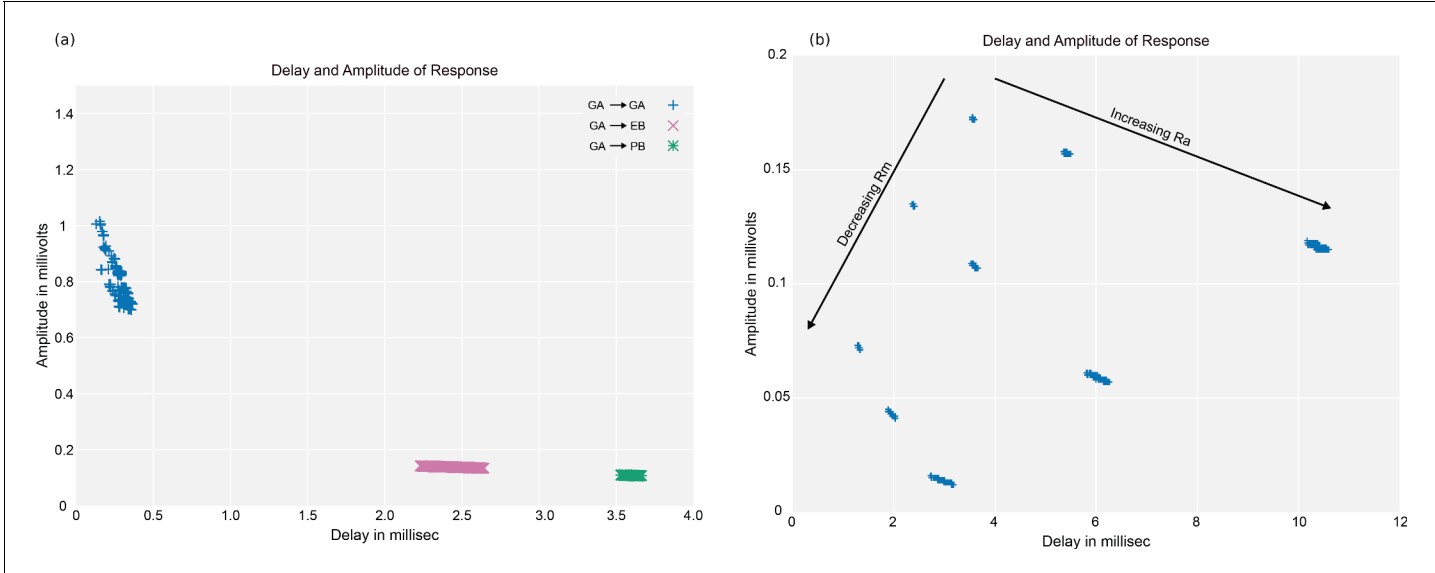

**Figure 25.** Delay versus amplitude plots for a neuron. (**a**) The linear response to inputs in the gall (GA) for an EPG neuron, which also has arbors in the ellipsoid body (EB) and the protocerebral bridge (PB). Each point in the modeled plot shows the time each response reached its peak amplitude (the delay), and the amplitude at that time, for an input injected at one of the PSDs in the gall. (**b**) Delays and amplitudes for gall to PB response, for all combinations of three values of cytoplasmic resistance $R_A$ and three values of membrane resistance $R_M$. Data available in *Figure 25—source datas 1–4*.

The online version of this article includes the following source data for figure 25:

**Source data 1.** Data for *Figure 25A* (ellipsoid body).
**Source data 2.** Data for *Figure 25A* (gall).
**Source data 3.** Data for *Figure 25A* (protocerebral bridge).
**Source data 4.** Data for *Figure 25B*.

compartment and across-compartment delays and amplitudes, as shown in *Figure 25(a)*. The same overall behavior is true for inputs into the other regions - short delays and strong responses within the compartment, with longer delays and smaller amplitudes to other compartments.

This simple pattern motivates a model that describes delays and amplitudes not as a single number, but as an $N \times N$ matrix, where $N$ is the number of compartments. Each row contains the estimated amplitude and delay, measured in each compartment, for a synaptic input in the given compartment. This gives a much improved estimate of the linear response. For the example EPG neuron above, with nominal values for $R_a$, $R_m$, and $C_m$, if we represent all delays by a single number then the standard deviation of the error is 0.446 ms. If instead we represent the delays as a $3 \times 3$ matrix indexed by the compartment, the average error is 0.045 ms, for 10x greater accuracy. Similarly, the average error in amplitude drops from 0.168 mv to 0.021 mv, an eightfold improvement. While the improvement in error will depend on the neuron topology, in all cases it will be more accurate than a point model, for relatively little increase in complexity.

The absolute values of delay and amplitude are strongly dependent on the electrical parameters of the cell, however. A wide range of electrical properties has been reported in the fly literature (see *Table 6*) and it is plausible that these vary on a cell-to-cell basis. In addition gap junctions, which are not included in our model, could affect the apparent value of $R_m$. In light of these uncertainties, we simulate with minimum, medium, and maximal values of $R_a$ and $R_m$, for a total of 9 cases, as shown in *Figure 25(b)*. All are needed since the resistance parameters interact non-linearly. We fix the value of $C_m$ at 0.01 F/m$^2$ since this value is determined by the membrane thickness and is not expected to vary from cell to cell (*Kandel et al., 2000*). The results over the parameter range are shown in *Figure 25(b)* for the case of the EPG neuron above for delay from the gall to the PB. The intra-compartment and between-compartment values are well separated for any value of the parameters (not shown).

**Table 6.** Values reported in the literature.

| Reference | $R_a, \Omega \cdot \mathrm{m}$ | $R_m, \Omega/\mathrm{m}^2$ | $C_m$, **F/m²** |
|---|---|---|---|
| Borst (*Borst and Haag, 1996*), CH cells | 0.60 | 0.25 | 0.015 |
| Borst (*Borst and Haag, 1996*), HS cells | 0.40 | 0.20 | 0.009 |
| Borst (*Borst and Haag, 1996*), VS cells | 0.40 | 0.20 | 0.008 |
| Gouwens (*Gouwens and Wilson, 2009*), DM1 cell 1 | 1.62 | 0.83 | 0.026 |
| Gouwens (*Gouwens and Wilson, 2009*), DM1 cell 2 | 1.02 | 2.04 | 0.015 |
| Gouwens (*Gouwens and Wilson, 2009*), DM1 cell 3 | 2.66 | 2.08 | 0.008 |
| Gouwens (*Gouwens and Wilson, 2009*), dendrite 1 | 2.44 | 1.92 | 0.008 |
| Gouwens (*Gouwens and Wilson, 2009*), dendrite 2 | 2.66 | 2.08 | 0.008 |
| Gouwens (*Gouwens and Wilson, 2009*), dendrite 3 | 3.11 | 2.64 | 0.006 |
| Cuntz (*Cuntz et al., 2013*), HS cells | 4.00 | 0.82 | 0.006 |
| Meier (*Meier and Borst, 2019*), CT1 cells | 4.00 | 0.80 | 0.006 |

Programs that deduce synaptic strength and sign by fitting a computed response to a connectome and measured electrical or calcium imaging data (*Tschopp et al., 2018*) may at some point require estimates of the delays within cells. If this is required, the above results suggest this could be accomplished with reasonable accuracy with a compartment-to-compartment delay table and two additional parameters per neuron, $R_A$ and $R_M$. This is relatively few new parameters in addition to the many synaptic strengths already fitted.

A number of neurons have parallel connections in separate compartments (see *Figure 24(b)*). This motif is common in the fly's brain – about 5% of all connections having a strength $\geq 6$ are spread across two or more non-adjacent compartments. Given the increased delays and lower amplitudes of cross-compartment responses, this type of interaction differs electrically from those in which all connections are contained in a single compartment. A point neuron model cannot generate an accurate response for such connections – a synapse in region A will result in a fast response in A and a slower, smaller response in B, and vice versa, even though both of these events involve communication between the same two neurons. It is not known if this configuration has a significant influence on the neurons' operation.

From these models, we conclude (a) the compartment structure of the fly brain shows up directly in the electrical response of the neurons, and (b) the compartment structure, although defined anatomically, matches that of the electrical response. From the clear separation in *Figure 25*, it is likely that the same compartment definitions could be found starting with the electrical response, although we have not tried this. (c) These results suggest a low dimensional model for neural operation, at least in the linear region. A small region-to-region matrix can represent the delays and amplitudes well. (d) Absolute delays depend strongly (but in a very predictable manner) on the values of axial and membrane resistance, which can vary both from animal to animal and from cell to cell. (e) Neurons that have parallel connections in separate compartments have a different electrical response than they would have with the same total number of synapses in a single compartment.

## Rent's rule analysis

Rent's rule (*Lanzerotti et al., 2005*) is an empirical observation that in human designed computing systems, when the system is packed as tightly as possible, at every level of the hierarchy the required communication (the number of pins) scales as a power law of the amount of contained computation, measured in gates. Rent's rule is an observed relationship, not derived from underlying theory, and the relationship is not exact and still contains scatter. A biological equivalent might be the observation that brain size tends to vary as a power law of body size (*Harvey and Krebs, 1990*), across a wide range of species occupying very different ecological and behavioral niches. Rent's rule is roughly true over many orders of magnitude in scale, and for almost every system in which it has been measured. Somewhat surprisingly, Rent's rule applies almost independently of the function performed by the computation being performed, and at every level of a hierarchical system. It also

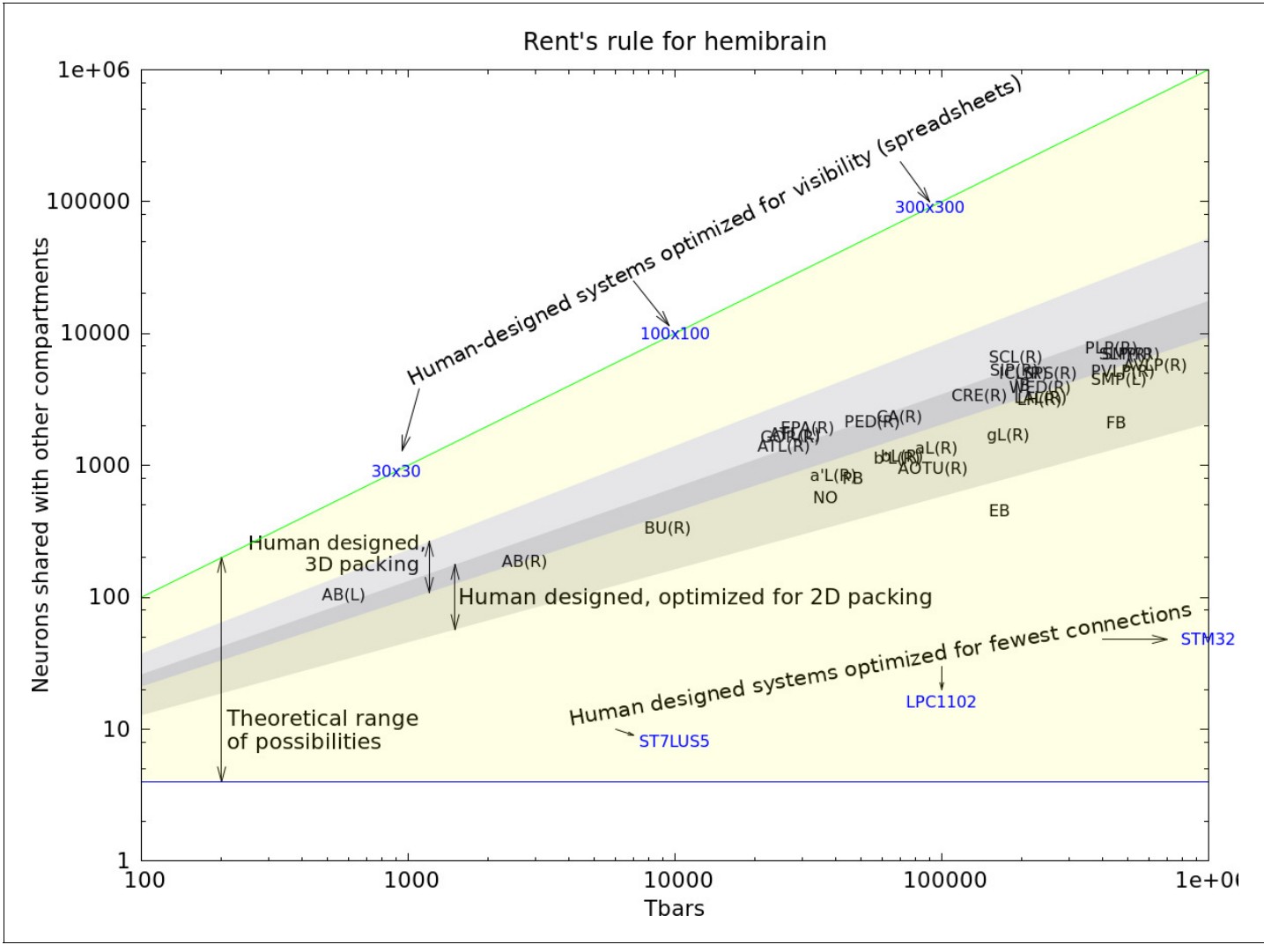

**Figure 26.** Rent's rule for the hemibrain. The yellow region encompasses the theoretical bounds for computation. Four varieties of human-designed systems are shown. Those designed for visibility into computation achieve the upper bound, while those designed for minimum communication approach the lower bounds (Microprocessors ST7LU55, LPC1102, and STM32). Human designed systems where efficient packing is the main criterion occupy the shaded area (in 2D and 3D). The characteristics of the primary compartments completely contained in the reconstructed volume are shown with alphanumeric labels. The hemibrain compartments fall very nearly in the same range as human designed systems designed for efficient packing. Data available in *Figure 26—source data 1*.

The online version of this article includes the following source data for figure 26:

**Source data 1.** Data for *Figure 26*.

applies whether the compactness criterion is minimization of communication (partitioning) or physical close packing.

Rent's rule is expressed as

$$Pins = a * (computation)^b$$

where *a* is a scale factor (typically in the range 1–4), and *b* is the 'Rent exponent' describing how the number of connections to the compartment varies as a function of the amount of computation performed in the compartment. The Rent exponent has a theoretical range of 0.0 to 1.0, where 0 represents a constant number of connections, with no dependence on the amount of computation performed, and 1.0 represents a circuit in which every computation is visible on a connection. Human designed computational systems occupy almost the full range, from spreadsheets in which

every computation is visible, to largely serial systems in which minimizing communication (pins) is critical. This relationship is shown in *Figure 26*. However, when the overriding criterion is that the system must be packed as tightly as possible, Rent observed that the exponent of the power law falls in a close range of roughly 0.5–0.7.

For electrical circuits, the computation is measured in gates, and the connections are measured by pin count. These ranges are shown in *Figure 26* for circuits that are roughly the size of the fly's brain, packed in either two (*Yang et al., 2001*) or three (*Das et al., 2004*) dimensions.

Also shown in this plot are the values for the fly's brain computational regions. In this case, the computation is measured as the number of contained T-bars, and the connection count is the number of neurons that have at least one synapse both inside and outside the compartment. (Very similar results are obtained if the computation is measured as the number of PSDs, or the number of unique connection pairs). Almost all the fly brain compartments fall well within the range of exponents expected for packing-dominated systems, while the ellipsoid body (EB) falls just outside the expected area. This is perhaps due to the large number of strongly connected clique-containing circuits in the ellipsoid body (see *Table 5*), since such circuits have relatively few connections for the amount of synapses they contain.

Both human designed and biological systems have huge incentives to pack their computation as tightly as possible. A tighter packing of the same computation yields faster operation, lower energy consumption, less material cost, and lower mass. A natural speculation, therefore, is that both the human-designed and evolved systems are dominated by packing considerations, and that both have found similar solutions.

## Conclusions and future work

In this work, we have achieved a dream of anatomists that is more than a century old. For at least the central brain of at least one animal with a complex brain and sophisticated behavior, we have a complete census of all the neurons and all the cell types that constitute the brain, a definitive atlas of the regions in which they reside, and a graph representing how they are connected.

To achieve this, we have made improvements to every stage of the reconstruction process. Better means of sample preparation, imaging, alignment, segmentation, synapse finding, and proofreading are all summarized in this work and will form the basis of yet larger and faster reconstructions in the future.

We have provided the data for all the circuits of the central brain, at least as defined by nerve cells and chemical synapses. This includes not only circuits of regions that are already the subject of extensive study, but also a trove of circuits whose structure and function are yet unknown.

We have provided a public resource that should be a huge help to all who study fly neural circuits. Finding upstream and downstream partners, a task that until now has typically taken months of challenging experiments, is now replaced by a lookup on a publicly available web site. Detailed circuits, which used to require considerable patience, expertise, and expertise to acquire, are now available for the cost of an internet query.

More widely, a dense connectome is a valuable resource for all neuroscientists, enabling novel, system-wide analyses, as well as suggesting roles for specific pathways. A surprising revelation is the richness of anatomical synaptic engagements, which far exceeds pathways required to support identified fly behaviors, and suggests that most behaviors have yet to be identified.

Finally, we have started the process of analyzing the connectome, though much remains to be done. We have quantified the difference between computational compartments, determined that the distribution of strengths is different from that reported in mammals, discovered cliques and other structures and where these occur, examined the effect of compartmentalization on electrical properties, and provided evidence that the wiring of the brain is consistent with optimizing packing.

Many of the extensions of this work are obvious and already underway. Not all regions of the hemibrain have been read to the highest accuracy possible, insofar as we have concentrated first on the regions overlapping with other projects, such as the central complex and the mushroom body. We will continue to update other sections of the brain, and distributed circuits such as clock and modulatory neurons that are not confined to one region, but spread throughout the brain.

There is much more to be learned about the graph properties of the brain, and how these relate to its function.

The two sexes of the *Drosophila* brain are known to differ (*Auer and Benton, 2016*). so that reconstructing a male fly is critical to compare the circuits of the two sexes. A ventral nerve cord (VNC) should be reconstructed, preferably attached to the brain of the same individual, since the circuits in the VNC are known to be crucial for fly motor behavior (*Yellman et al., 1997*). At least one optic lobe should be included to simplify analysis of visual inputs to the central brain. A whole brain connectome is preferable to the hemibrain, since then most cell types would have at least two examples, left and right, which would lend increased confidence to our reconstructions. It would also provide complete reconstruction to the many neurons that span the brain, especially the clock and modulatory neurons, and are incomplete in the hemibrain. These four goals are combined in a project that is currently underway, to image and reconstruct an entire male central nervous system (CNS) including the VNC and optic lobes.

We continue to improve sample preparation, imaging, and reconstruction both to decrease the efforts expended on reconstruction and to speed reconstruction of more specimens. Improvements include multi-beam imaging, etching methods (*Hayworth et al., 2020*) that can handle larger areas, and yet better reconstruction techniques. These improvements, however, will still rely on FIB-SEM technology, and additional methods will likely be required to fill in other information. Gap junctions will continue to be difficult to see in FIB-SEM, and other methods such as optical labeling, expansion microscopy, and RNA-SEQ (to find which neurons express gap junction proteins) will be required. Methods for estimating the extent of diffusion of the secreted modulatory transmitters and gaseous signal molecules such as NO remain to be established. Different staining methods (and expression driver lines) may be needed to study glia to the same extent we currently study neurons. A wide variety of techniques will be needed to understand the subcellular architecture of the neurons we have reconstructed. Finally, larger animal brains beckon, such as the brain of a mouse and eventually a human. The data we present here is only a start.

## Acknowledgements

We thank our colleagues at Janelia and the broader connectomics field for many helpful discussions and suggestions during the course of this work. We thank David Peale, Patrick Lee, and the Janelia Experimental Technology group for supporting the modifications of the FIB-SEM systems. Goran Ceric and other members of the Scientific Computing Systems and Scientific Computing Software Teams at Janelia provided critical support throughout this work. The Janelia Facilities group was essential in proving a stable environment for image collection. We thank Julia Buhmann and Jan Funke for help in implementing the synapse prediction algorithm described in *Buhmann et al., 2019*. Many colleagues at Janelia and Cambridge tested the performance of Neuprint performance prior to its release.

We thank Elizabeth Marin, Tomke Stuerner, Sridhar Jagannathan, Shahar Frechter (Cambridge University/MRC LMB) for additional contributions to the identification/typing and naming of all the neurons associated with the antennal lobe and lateral horn, Rachel Wilson and Asa Barth-Maron (Harvard University) for the typing and naming of antennal lobe local neurons, Nils Otto (Oxford University), Georgia Dempsey and Ildiko Stark (Cambridge University) for typing of some mushroom body dopaminergic neurons, Thomas Riemensperger (University of Cologne) for the identification of known aminergic and peptidergic neurons, Nik Drummond, Markus Pleijzier, Konrad Heinz (Cambridge University) and Joe Hsu (Janelia) for additional tracing of olfactory system neurons, Kaiyu Wang and Barry Dickson (Janelia) for identification of fruitless expressing neurons, and Aljosha Nern, Michael B. Reiser, and Arthur Zhao (Janelia) for the identification of optic lobe neurons.

Major financial support for this work was provided by the Howard Hughes Medical Institute and Google Research. Feng Li, Marta Costa, Philipp Schlegel, and approximately 10% of the proofreading team were supported by a Wellcome Trust Collaborative Award (203261/Z/16/Z).

## Additional information

### Competing interests

Michal Januszewski, Jeremy Maitlin-Shepard, Tim Blakely, Laramie Leavitt, Peter H Li, Larry Lindsey, Viren Jain: is an employee of Google. The other authors declare that no competing interests exist.

## Funding

| Funder | Grant reference number | Author |
| --- | --- | --- |
| Howard Hughes Medical Institute | Internal funding | Louis K Scheffer<br>C Shan Xu<br>Zhiyuan Lu<br>Shin-ya Takemura<br>Kenneth J Hayworth<br>Gary B Huang<br>Kazunori Shinomiya<br>Stuart Berg<br>Jody Clements<br>Philip M Hubbard<br>William T Katz<br>Lowell Umayam<br>Ting Zhao<br>David Ackerman<br>John Bogovic<br>Tom Dolafi<br>Dagmar Kainmueller<br>Khaled A Khairy<br>Nicole Neubarth<br>Donald J Olbris<br>Hideo Otsuna<br>Eric T Trautman<br>Masayoshi Ito<br>Jens Goldammer<br>Tanya Wolff<br>Robert Svirskas<br>Erika Neace<br>Christopher J Knecht<br>Chelsea X Alvarado<br>Dennis A Bailey<br>Samantha Ballinger<br>Jolanta A Borycz<br>Brandon S Canino<br>Natasha Cheatham<br>Michael Cook<br>Marisa Dreher<br>Octave Duclos<br>Bryon Eubanks<br>Kelli Fairbanks<br>Samantha Finley<br>Nora Forknall<br>Audrey Francis<br>Gary Patrick Hopkins<br>Emily M Joyce<br>SungJin Kim<br>Nicole A Kirk<br>Julie Kovalyak<br>Shirley A Lauchie<br>Alanna Lohff<br>Charli Maldonado<br>Emily A Manley<br>Sari McLin<br>Caroline Mooney<br>Miatta Ndama<br>Omotara Ogundeyi<br>Nneoma Okeoma<br>Christopher Ordish<br>Nicholas Padilla<br>Christopher M Patrick<br>Tyler Paterson<br>Elliott E Phillips<br>Emily M Phillips<br>Neha Rampally<br>Caitlin Ribeiro<br>Madelaine K Robertson<br>Jon Thomson Rymer<br>Sean M Ryan<br>Megan Sammons<br>Anne K Scott<br>Ashley L Scott<br>Aya Shinomiya |

| | | Claire Smith |
| | | Kelsey Smith |
| | | Natalie L Smith |
| | | Margaret A Sobeski |
| | | Alia Suleiman |
| | | Jackie Swift |
| | | Satoko Takemura |
| | | Iris Talebi |
| | | Dorota Tarnogorska |
| | | Emily Tenshaw |
| | | Temour Tokhi |
| | | John J Walsh |
| | | Tansy Yang |
| | | Jane Anne Horne |
| | | Ruchi Parekh |
| | | Patricia K Rivlin |
| | | Vivek Jayaraman |
| | | Kei Ito |
| | | Stephan Saalfeld |
| | | Reed George |
| | | Ian A Meinertzhagen |
| | | Gerald M Rubin |
| | | Harald F Hess |
| | | Stephen M Plaza |
| Google | Internal funding | Michal Januszewski |
| | | Jeremy Maitlin-Shepard |
| | | Tim Blakely |
| | | Laramie Leavitt |
| | | Peter H Li |
| | | Larry Lindsey |
| | | Viren Jain |
| Wellcome | 203261/Z/16/Z | Philipp Schlegel |
| | | Feng Li |

The funders had no role in study design, data collection and interpretation, or the decision to submit the work for publication.

## Author contributions

Louis K Scheffer, Conceptualization, Data curation, Software, Formal analysis, Investigation, Visualization, Methodology, Writing - original draft, Writing - review and editing, Image alignment, analysis software, connectivity analysis; C Shan Xu, Investigation, Methodology, Writing - original draft, Writing - review and editing, Developed imaging hardware; imaged the sample; Michal Januszewski, Software, Validation, Investigation, Visualization, Methodology, Developed and applied segmentation and tissue classification; Zhiyuan Lu, Validation, Investigation, Methodology, Developed sample preparation methods; fixed and stained the sample; Shin-ya Takemura, Formal analysis, Validation, Investigation, Visualization, Methodology, Defined brain regions and cell types; biological interpretation and analysis; Kenneth J Hayworth, Validation, Investigation, Visualization, Methodology, Writing - original draft, Developed hot-knife method; cut and images sample; Gary B Huang, Software, Validation, Investigation, Visualization, Methodology, Writing - original draft, Developed and applied methods to identify synapses; Kazunori Shinomiya, Software, Formal analysis, Validation, Investigation, Visualization, Methodology, Defined regions and cell types; biological interpretation and analysis; Jeremy Maitlin-Shepard, Software, Formal analysis, Investigation, Visualization, Methodology, Developed and applied segmentation methods, developed analysis and visualization pipeline software; Stuart Berg, Software, Formal analysis, Investigation, Visualization, Methodology, Wrote proofreading and analysis software; Jody Clements, Data curation, Software, Visualization, Methodology, Wrote analysis and visualization software; Philip M Hubbard, Software, Investigation, Visualization, Methodology, Wrote analysis and visualization software; William T Katz, Conceptualization, Data curation, Software, Investigation, Visualization, Methodology, Wrote and managed the versioned data system; Lowell Umayam, Data curation, Software, Validation, Visualization, Wrote proofreading and analysis software; Ting Zhao, Software, Investigation, Visualization, Methodology, Wrote proofreading and data analysis software; David Ackerman, Software, Investigation, Image alignment and flattening; Tim Blakely, Larry Lindsey, Software, Investigation, Visualization, Methodology, Developed analysis, visualization, and pipeline software; John Bogovic, Hideo Otsuna, Software,

Investigation, Visualization, Methodology, Developed EM-Optical mapping; Tom Dolafi, Software, Investigation, Visualization, Methodology, Developed proofreading and analysis software; Dagmar Kainmueller, Software, Investigation, Methodology, Image alignment and flattening; Takashi Kawase, Software, Investigation, Visualization, Methodology, Proofreading and analysis software; Khaled A Khairy, Software, Investigation, Visualization, Methodology, Wrote and applied image alignment software; Laramie Leavitt, Software, Investigation, Visualization, Methodology, Developed and applied tissue classification approaches and analysis, visualization and pipeline software; Peter H Li, Software, Investigation, Visualization, Methodology, Developed and applied segmentation approaches; Nicole Neubarth, Software, Investigation, Visualization, Wrote proofreading and analysis software; Donald J Olbris, Philipp Schlegel, Software, Investigation, Visualization, Methodology, Wrote proofreading and analysis software; Eric T Trautman, Software, Validation, Investigation, Visualization, Methodology, Wrote image alignment software and aligned data; wrote proofreading and analysis software; Masayoshi Ito, Software, Investigation, Visualization, Methodology, Defined regions and cell types; Alexander S Bates, Data curation, Validation, Investigation, Visualization, Defined brain regions and cell types; Jens Goldammer, Software, Investigation, Visualization, Methodology, Biological interpretation and analysis; defined brain regions and cell types; Tanya Wolff, Validation, Investigation, Visualization, Methodology, Defined brain regions and cell types; Robert Svirskas, Software, Supervision, Visualization, Methodology, Developed pipeline analysis and software; Erika Neace, Software, Validation, Investigation, Visualization, Methodology, Proofreading analytics; Christopher J Knecht, Software, Supervision, Validation, Investigation, Visualization, Methodology, Proofreading analytics; Chelsea X Alvarado, Dennis A Bailey, Samantha Ballinger, Jolanta A Borycz, Brandon S Canino, Natasha Cheatham, Michael Cook, Octave Duclos, Bryon Eubanks, Kelli Fairbanks, Samantha Finley, Nora Forknall, Audrey Francis, Gary Patrick Hopkins, Emily M Joyce, SungJin Kim, Nicole A Kirk, Julie Kovalyak, Shirley A Lauchie, Alanna Lohff, Charli Maldonado, Emily A Manley, Sari McLin, Caroline Mooney, Miatta Ndama, Omotara Ogundeyi, Nneoma Okeoma, Nicholas Padilla, Christopher M Patrick, Tyler Paterson, Elliott E Phillips, Emily M Phillips, Neha Rampally, Caitlin Ribeiro, Madelaine K Robertson, Jon Thomson Rymer, Sean M Ryan, Megan Sammons, Anne K Scott, Ashley L Scott, Aya Shinomiya, Claire Smith, Kelsey Smith, Natalie L Smith, Margaret A Sobeski, Alia Suleiman, Jackie Swift, Satoko Takemura, Iris Talebi, Dorota Tarnogorska, Emily Tenshaw, Temour Tokhi, John J Walsh, Tansy Yang, Validation, Investigation, Proofreading; Marisa Dreher, Validation, Investigation, Proofreading and figures; Christopher Ordish, Validation, Investigation, Methodology, Proofreading; Jane Anne Horne, Software, Formal analysis, Supervision, Validation, Investigation, Visualization, Methodology, Project administration, Defined brain regions and cell types; Feng Li, Software, Formal analysis, Supervision, Validation, Investigation, Visualization, Methodology, Project administration, Defined brain regions and cell types; managed proofreading; biological interpretation; Ruchi Parekh, Resources, Supervision, Validation, Investigation, Visualization, Methodology, Project administration, Biological interpretation and defined brain regions and cell types; managed proofreading; Patricia K Rivlin, Resources, Supervision, Validation, Investigation, Visualization, Methodology, Project administration, Biological interpretation; managed proofreading; Vivek Jayaraman, Conceptualization, Resources, Writing - review and editing, Biological interpretation and analysis; Marta Costa, Data curation, Supervision, Validation, Investigation, Visualization, Methodology, Writing - review and editing, Defined brain regions and cell typing; biological interpretation and analysis; Gregory SXE Jefferis, Data curation, Supervision, Investigation, Visualization, Methodology, Writing - review and editing, Defined brain regions and cell types; biological interpretation and analysis; Kei Ito, Formal analysis, Supervision, Validation, Investigation, Visualization, Methodology, Defined brain regions and cell types; managed cell typing; Stephan Saalfeld, Software, Validation, Investigation, Visualization, Methodology, Writing - original draft, Image alignment; Reed George, Conceptualization, Resources, Project administration, Managed overall effort; Ian A Meinertzhagen, Conceptualization, Formal analysis, Supervision, Validation, Investigation, Methodology, Writing - original draft, Project administration, Writing - review and editing, Biological interpretation and analysis; Gerald M Rubin, Conceptualization, Resources, Supervision, Funding acquisition, Writing - original draft, Project administration, Writing - review and editing, Biological interpretation and analysis; managed overall effort; Harald F Hess, Conceptualization, Resources, Supervision, Validation, Investigation, Visualization, Methodology, Project administration, Developed imaging hardware and imaged sample; Viren Jain, Resources, Software, Validation, Investigation, Visualization, Methodology, Writing - original draft, Project administration, Writing - review and editing,

Developed and applied tissue classification and segmentation approaches; Stephen M Plaza, Conceptualization, Resources, Software, Supervision, Funding acquisition, Validation, Investigation, Visualization, Methodology, Writing - original draft, Project administration, Writing - review and editing, Wrote proofreading and analysis software; connectivity analysis; managed overall effort

**Author ORCIDs**

Louis K Scheffer https://orcid.org/0000-0002-3289-6564

C Shan Xu http://orcid.org/0000-0002-8564-7836

Michal Januszewski https://orcid.org/0000-0002-3480-2744

Zhiyuan Lu http://orcid.org/0000-0002-4128-9774

Shin-ya Takemura http://orcid.org/0000-0003-2400-6426

Gary B Huang http://orcid.org/0000-0002-9606-3510

Kazunori Shinomiya http://orcid.org/0000-0003-0262-6421

Jeremy Maitlin-Shepard https://orcid.org/0000-0001-8453-7961

Philip M Hubbard http://orcid.org/0000-0002-6746-5035

William T Katz http://orcid.org/0000-0002-9417-6212

David Ackerman http://orcid.org/0000-0003-0172-6594

Tim Blakely https://orcid.org/0000-0003-0995-5471

John Bogovic http://orcid.org/0000-0002-4829-9457

Dagmar Kainmueller http://orcid.org/0000-0002-9830-2415

Khaled A Khairy https://orcid.org/0000-0002-9274-5928

Peter H Li https://orcid.org/0000-0001-6193-4454

Eric T Trautman https://orcid.org/0000-0001-8588-0569

Alexander S Bates http://orcid.org/0000-0002-1195-0445

Jens Goldammer http://orcid.org/0000-0002-5623-8339

Tanya Wolff http://orcid.org/0000-0002-8681-1749

Robert Svirskas http://orcid.org/0000-0001-8374-6008

Philipp Schlegel http://orcid.org/0000-0002-5633-1314

Christopher J Knecht https://orcid.org/0000-0002-5663-5967

Chelsea X Alvarado http://orcid.org/0000-0002-5973-7512

Dennis A Bailey http://orcid.org/0000-0002-4675-8373

Jolanta A Borycz http://orcid.org/0000-0002-4402-9230

Brandon S Canino http://orcid.org/0000-0002-8454-865X

Michael Cook http://orcid.org/0000-0002-7892-6845

Marisa Dreher http://orcid.org/0000-0002-0041-9229

Bryon Eubanks http://orcid.org/0000-0002-9288-2009

Kelli Fairbanks http://orcid.org/0000-0002-6601-4830

Samantha Finley http://orcid.org/0000-0002-8086-206X

Nora Forknall http://orcid.org/0000-0003-2139-7599

Audrey Francis http://orcid.org/0000-0003-1974-7174

Emily M Joyce http://orcid.org/0000-0001-5794-6321

Julie Kovalyak http://orcid.org/0000-0001-7864-7734

Shirley A Lauchie https://orcid.org/0000-0001-8223-9522

Alanna Lohff http://orcid.org/0000-0002-1242-1836

Sari McLin http://orcid.org/0000-0002-9120-1136

Christopher M Patrick http://orcid.org/0000-0001-8830-1892

Elliott E Phillips https://orcid.org/0000-0002-4918-2058

Emily M Phillips https://orcid.org/0000-0001-7615-301X

Madelaine K Robertson https://orcid.org/0000-0002-1764-0245

Jon Thomson Rymer http://orcid.org/0000-0002-4271-6774

Sean M Ryan https://orcid.org/0000-0002-8879-6108

Megan Sammons http://orcid.org/0000-0003-4516-5928

Aya Shinomiya http://orcid.org/0000-0002-6358-9567

Natalie L Smith https://orcid.org/0000-0002-8271-9873

Jackie Swift http://orcid.org/0000-0003-1321-8183
Satoko Takemura https://orcid.org/0000-0002-2863-0050
Iris Talebi https://orcid.org/0000-0002-0173-8053
Dorota Tarnogorska http://orcid.org/0000-0002-7063-6165
John J Walsh https://orcid.org/0000-0002-7176-4708
Tansy Yang http://orcid.org/0000-0003-1131-0410
Jane Anne Horne http://orcid.org/0000-0001-9673-2692
Ruchi Parekh https://orcid.org/0000-0002-8060-2807
Vivek Jayaraman http://orcid.org/0000-0003-3680-7378
Marta Costa http://orcid.org/0000-0001-5948-3092
Gregory SXE Jefferis http://orcid.org/0000-0002-0587-9355
Kei Ito https://orcid.org/0000-0002-7274-5533
Stephan Saalfeld https://orcid.org/0000-0002-4106-1761
Gerald M Rubin http://orcid.org/0000-0001-8762-8703
Harald F Hess http://orcid.org/0000-0003-3000-1533
Stephen M Plaza https://orcid.org/0000-0001-7425-8555

## Decision letter and Author response

Decision letter https://doi.org/10.7554/eLife.57443.sa1
Author response https://doi.org/10.7554/eLife.57443.sa2

# Additional files

## Supplementary files

- Supplementary file 1. Spreadsheet of instances of sparse-to-many connections.

- Transparent reporting form

## Data availability

There is no institutional resource for hosting connectome data. Therefore we host it ourselves on a publicly accessible web site, https://neuprint.janelia.org, also accessible via https://doi.org/10.25378/janelia.11676099.v2. We commit to keeping this available for at least 10 years, and provide procedures where users can copy any or all of it to their own computer. Login is via any Google account; users who wish to remain anonymous can create a separate account for access purposes only.

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

## Appendix 1

### Sensory inputs and motor outputs

The dataset covers most of the antennal lobe (AL) glomeruli, which house the presynaptic terminals of the olfactory receptor neurons (ORNs) from the antennae. The ORNs are named after their innervating glomeruli, for example ORN_DA2, and the olfactory receptors they express, as well as their ligands, and have been identified through various physiological studies (*Couto et al., 2005*; *Fishilevich and Vosshall, 2005*; *Hallem and Carlson, 2006*). The olfactory signals are then transmitted by the olfactory projections neurons (PNs) to the calyx (CA) of the mushroom body, the lateral horn (LH) and beyond.

While a large fraction of the optic lobe (OL) neuropils are missing, more than half of the lobula (LO) and small pieces of the lobula plate (LOP) and medulla (ME) are within the dataset. Many neurons connecting the OL and the central brain, called visual projection neurons (VPNs), are identified and annotated, along with their synaptic terminals in the central brain, and in the optic lobe when possible. Among them, the columnar VPNs, including the lobula columnar (LC), lobula plate columnar (LPC), lobula-lobula plate columnar (LLPC), and lobula plate-lobula columnar (LPLC) neurons (*Ache et al., 2019*; *Fischbach and Dittrich, 1989*; *Klapoetke et al., 2017*; *Otsuna and Ito, 2006*; *Wu et al., 2016*), account for the vast majority of the population and are more or less densely identified. Since the distribution of the columnar neurons in the optic lobe follows the arrangement of the photoreceptor cells in the compound eye, the retinotopy can be traced even in their terminals in the central brain in some cell types, while in others the retinotopy is apparently lost in the central brain. In most cases, these neurons terminate in synapse-rich structures called the optic glomeruli in the ventrolateral neuropils, where they relay visual information to higher-order neurons (*Panser et al., 2016*; *Wu et al., 2016*).

The antennal mechanosensory and motor center (AMMC) is located lateral and ventral to the esophagus foramen. It houses terminals of the Johnston's organ neurons (JONs), the mechanosensory neurons from the Johnston's organ in the second segment of the antennae, as well as their synaptic partners. The AMMC is subdivided into five functionally and anatomically segregated zones, A, B, C, D, and E (*Kamikouchi et al., 2006*). Since the neuropil is partially truncated, especially in the medial and ventral part corresponding to the zones D and E in the hemibrain dataset, only a limited number of the JONs innervating zones A, B, and C have been annotated, as JO-ABC.

The gustatory receptor neurons (GRNs) from the labellum and maxillary palp terminate in the gustatory sensory centers in the gnathal ganglia (GNG) and the prow (PRW) (*Hartenstein et al., 2018*; *Ito et al., 2014*; *Miyazaki and Ito, 2010*). Both of them are mostly out of the imaging range of the dataset and therefore no GRNs have been identified.

We have identified 51 types of descending neurons (out of a total of 98 types identified by the LM study) that play a key role in behavior. These neurons were annotated based on the nomenclature described in a previous study (*Namiki et al., 2018*), namely the classes of DNa, DNb, DNg, and DNp. Due to the lack of ventral region in the current dataset, we are not able to specify other cell types that run in the neck connective. In addition we identified three types of descending neurons that go out of the brain at the esophogus foramen without going via the neck connective (DNES1, 2, and 3).

### Sample preparation

We employed the Progressive Lowering of Temperature dehydration with Low temperature en bloc Staining (PLT-LTS), a modified conventional chemical fixation and en bloc staining method. This method, mentioned in our previous papers (*Hayworth et al., 2015*; *Xu et al., 2017*; *Lu et al., 2019*), is here abbreviated as 'C-PLT'. PLT-LTS is an optimization method to give tissue advanced high contrast staining and minimize artifacts such as extraction, and size and shape variation, by treating tissue under 0°C to −25°C in acetone or ethanol based uranyl acetate and osmium tetroxide after routine fixation. PLT-LTS samples show highly visible membranes with fewer deflated and collapsed profiles and conspicuous synaptic densities in FIB-SEM images.

Five-day-old adult female *Drosophila*, of the genotype Canton S G1 x w[1118], were used in this experiment. They were raised on a 12-hr day/night cycle, with dissection performed 1.5 hr after lights-on. Isolated whole brains were fixed in 2.5% formaldehyde and 2.5% glutaraldehyde in 0.1 M

phosphate buffer at pH 7.4 for 2 hr at 22°C. After washing, the tissues were post-fixed in 0.5% osmium tetroxide in double distilled $H_2O$ for 30 min at 4°C. After washing and en bloc staining with 0.5% aqueous uranyl acetate for 30 min and then further washing in water, for 20 min in 0.8% osmium tetroxide, a Progressive of Lowering Temperature (PLT) procedure started from 1°C when the tissues were transferred into 10% acetone. The temperature was progressively decreased to −25°C while the acetone concentration was gradually increased to 97%. The tissue was incubated in 1% osmium tetroxide and 0.2% uranyl acetate in acetone for 32 hr at −25°C. After PLT and low temperature incubation, the temperature was increased to 22°C, and tissues were rinsed in pure acetone following by propylene oxide, then infiltrated and embedded in Poly/Bed 812 epoxy (Luft formulation).

## Hot knife cutting
### Ultrathick sectioning

The hemibrain is too large to image by FIB-SEM without artifacts so we used our ultrathick sectioning 'hot knife' procedure (*Hayworth et al., 2015*) to first slice the brain into 20-µm-thick slabs which were better suited to FIB-SEM imaging. The Epon-embedded *Drosophila* brain block's face was trimmed to present a width of just over 1 mm to the knife during sectioning (with the brain centered in this width). The length of the blockface was trimmed to be >3 mm so that each cut section would have a large enough region of blank plastic surrounding the tissue to allow forceps to grasp it during later processing steps. All sides of the block were trimmed to be perpendicular to the face except the trailing edge which was trimmed to slope away at ≈ 45° (to prevent this trailing edge from deforming during hot knife sectioning). Hot knife sectioning was performed on our custom ultrathick sectioning testbed (*Hayworth et al., 2015*). The block was cut at a speed of 0.1 mm/s into a total of 37 slices, each 20 µm thick, using an oil-lubricated (filtered thread cutting oil, Master Plumber) diamond knife (Cryo 25° from Diatome). The knife temperature was adjusted at the beginning of the run to ensure sections flowed smoothly across the knife surface without curling (too cold) or buckling (too hot). The knife temperature was measured to be 61°C at the end of the run. The knife was forced to oscillate via a piezo at 39 kHz during sectioning. A laser vibrometer (Polytec CLV-2534) was used to measure the amplitude of vibration at 0.5 µm peak-to-peak. Each thick section was collected individually from the knife surface by pressing a vacuum aspirator (extended fine tip plastic transfer pipette, Samco Scientific, attached to lab vacuum) onto the surface of the section. Each section was transferred to an individual well in the top of a 96-well microplate (Costar) into an awaiting oil drop. Once all sections were collected, they were transferred via forceps under a dissection microscope to a glass slide. The slide was placed on a hot plate (200°C) long enough (≈ 10 s) to flatten any residual curl in the sections. Each section was then imaged in a 20x light microscope to evaluate its quality.

### Flat embedding

Each of the 20-µm-thick Epon-embedded fly brain sections was re-embedded in Durcupan resin to allow high quality FIB-SEM imaging. Durcupan re-embedding was required because FIB milling of Epon-embedded tissue without a Durcupan front covering resulted in milling streaks which mar the SEM images (*Xu et al., 2017*). Residual oil left over from the cutting process was first removed from each thick sections by dipping the section in Durcupan resin. Four drops of Durcupan resin were spaced out in sequence on a fresh glass slide. Each section was manually grasped with forceps (under a dissecting microscope) and dipped and lightly agitated sequentially in each Durcupan drops. Sections were gently wiped against the glass slide between each dipping to remove excess Durcupan and oil. After the final dipping, each section was placed (blockface side up) onto the heat-sealable side of a strip of 25 µm thick PET film (PP24I, Polymex Clear one side heat sealable/one side untreated polyester film, Polyester Converter Ltd.). Flat embedding tissue sections against this PET backing provided the strength needed for later mounting and handling. The PET film had been previously affixed to a glass slide for support, separated from the slide by a thin Kapton film designed to allow easy stripping of the PET. A gasket made from 50 µm thick adhesive-backed Kapton was positioned so as to surround all of the sections making a well for Durcupan resin to be poured into. This arrangement of sections was placed in a 65°C oven for ≈ 1 hr to partially cure the

Durcupan so as to 'tack' the sections into position against the PET film. Then fresh Durcupan was poured to fill the well to its brim, and several large area pieces of 20-µm-thick Durcupan (previously cut from a blank block) were placed above the tissue sections to act as spacers during flat embedding to ensure that at least a 20 µm layer of Durcupan would exist in front of each tissue section during FIB milling. A piece of 25 µm Kapton film was laid on top of the Durcupan along with a glass slide and a weight was placed on top to press excess Durcupan out of the well. This flat embedding stack up was cured at 65°C for 2 days.

## Tab mounting, laser trimming, X-ray imaging

Each individual brain slab to be FIB-SEM imaged was cut out of this flat embedding using a scalpel, and the resulting 'tab' was affixed with cyanoacrylate (Super Glue) to a metal stud. An ultraviolet laser (LaserMill, New Wave Research) was used to trim away excess blank resin to minimize the FIB-milling time required. An X-ray micro-CT scan (Versa 520, Zeiss) was then performed on each tab prior to FIB-SEM imaging.

## Imaging

For the hemibrain, thirteen such slices were imaged using two customized enhanced FIB-SEM systems, in which an FEI Magnum FIB column was mounted at 90° onto a Zeiss Merlin SEM. Three different imaging conditions were used for different sections with details listed in *Appendix 1—table 1*. In general, SEM images were acquired at 8 nm XY pixel size with a 4-nA beam with 1.2 kV landing energy, but other parameters were tuned for best imaging quality. Slices 24 to 27 were imaged with the specimen biased at + 600 V to prevent secondary electrons from reaching the detector, so that only backscattered electrons were collected. The electron beam energy was lowered to 600 V accordingly to maintain the same 1.2 kV landing energy. The remaining slices were imaged with specimen grounded at 0 V, and both secondary and backscattered electrons were collected to improve signal-to-noise ratio. As a result, SEM scanning rates were set at 2 MHz for slabs with specimen bias and 4 MHz for those without specimen bias. FIB milling was carried out by a 7-nA 30 kV Ga ion beam. Since optic lobes are typically more heavily stained than the central brain, the FIB milling step size in sections 22 to 30 was set to 2 nm, while the step size on sections 31 to 34 was set at 4 nm, to compensate for staining nonuniformity while preserving throughput and signal-to-noise ratio. The total FIB-SEM imaging time for the entire hemibrain was roughly four FIB-SEM-years: two years of on and off operation with two machines.

**Appendix 1—table 1.** FIB-SEM imaging conditions.

| Sample ID | Electron beam energy (kV) | Sample bias (kV) | Landing energy (kV) | SEM current (nA) | SEM scan rate (MHz) | x-y pixel (nm) | z-step (nm) |
|---|---|---|---|---|---|---|---|
| Z0115-22_Sec22 | 1.2 | 0 | 1.2 | 4 | 4 | 8 | 2 |
| Z0115-22_Sec23 | 1.2 | 0 | 1.2 | 4 | 4 | 8 | 2 |
| Z0115-22_Sec24 | 0.6 | 0.6 | 1.2 | 4 | 2 | 8 | 2 |
| Z0115-22_Sec25 | 0.6 | 0.6 | 1.2 | 4 | 2 | 8 | 2 |
| Z0115-22_Sec26 | 0.6 | 0.6 | 1.2 | 4 | 2 | 8 | 2 |
| Z0115-22_Sec27 | 0.6 | 0.6 | 1.2 | 4 | 2 | 8 | 2 |
| Z0115-22_Sec28 | 1.2 | 0 | 1.2 | 4 | 4 | 8 | 2 |
| Z0115-22_Sec29 | 1.2 | 0 | 1.2 | 4 | 4 | 8 | 2 |

*Continued on next page*

*Appendix 1—table 1 continued*

| Sample ID | Electron beam energy (kV) | Sample bias (kV) | Landing energy (kV) | SEM current (nA) | SEM scan rate (MHz) | x-y pixel (nm) | z-step (nm) |
|---|---|---|---|---|---|---|---|
| Z0115-22_Sec30 | 1.2 | 0 | 1.2 | 4 | 4 | 8 | 2 |
| Z0115-22_Sec31 | 1.2 | 0 | 1.2 | 4 | 4 | 8 | 4 |
| Z0115-22_Sec32 | 1.2 | 0 | 1.2 | 4 | 4 | 8 | 4 |
| Z0115-22_Sec33 | 1.2 | 0 | 1.2 | 4 | 4 | 8 | 4 |
| Z0115-22_Sec34 | 1.2 | 0 | 1.2 | 4 | 4 | 8 | 4 |

## Slab alignment

From each of the flattened sections, we generated a multi-scale pyramid of the section faces. The highest resolution pyramid level sat exactly at the surface plane, had a thickness of 1 pixel and showed a significant amount of cutting artifacts. Lower levels of the pyramid were increasingly thicker, projecting deeper into the volume and showed larger structures.

The alignment was initialized with a regularized affine alignment for the complete series of face pairs using the feature based method by *Saalfeld et al., 2010*. The pyramid of section face pairs was then used to robustly calculate pairwise deformations between adjacent sections. The faces are of notable size (>30k$^2$ pixels) and expose many preparation artifacts such that off the shelf registration packages failed to process them reliably. We therefore developed a custom pipeline that was able to robustly align the complete series without manual corrections. Using the same feature-based method as above, an increasingly fine grid of local affine transformations was calculated and converted into a smooth and increasingly accurate interpolated deformation field. The resulting deformation field was further refined using a custom hierarchical optic flow method down to a resolution of 2 pixels. Optic flow minimizing the normalized cross correlation (NCC) was calculated for a pyramid of square block-sizes. For each pixel, the translation vector with the highest number of votes from all block-sizes was selected, and the resulting flow-field was further smoothed with an adaptive Gaussian filter that was weighted by the corresponding NCC.

The deformation fields were then applied to each section volume by smoothly interpolating between the deformation field at the top face and the affine transformation at the bottom face.

The block-based N5 format (https://github.com/saalfeldlab/n5; *Saalfeld, 2020a*; copy archived at https://github.com/elifesciences-publications/n5) was used to store volumes, multi-scale face pyramids, deformation fields, meta-data, and to generate the final export. Apache Spark was used to parallelize on a compute cluster. The pipeline is open source and available on GitHub (https://github.com/saalfeldlab/hot-knife; *Saalfeld, 2020b*; copy archived at https://github.com/elifesciences-publications/hot-knife).

## Segmentation
### Image adjustment with CycleGANs

To reduce photometric variation, we first normalized the contrast of the aligned EM images at full resolution ([8 nm]$^3$/voxel) with CLAHE in planes parallel to the hot-knife cuts. In experiments targeted to small subvolumes we observed that segmentation quality decreased in certain areas of the hemibrain volume due to variations in the image content arising from, for example, fluctuations in staining quality as well as reduced contrast near the boundaries of the physically distinct 13 hot-knife 'tabs' that partitioned the original tissue volume. To compensate for these irregularities, we trained and applied CycleGAN (*Zhu et al., 2017*) models. This unsupervised machine learning method was originally introduced to adjust the appearance of images from one set A (e.g. photos) to be similar to those from another set B (e.g. paintings), without being given any explicit pairings between elements of both sets. Here, we extended this method to 3D volumes, and used model architectures

and training hyperparameters as previously described (*Januszewski and Jain, 2019*), but without utilizing the flood-filling module.

We trained separate CycleGAN models to make data from every tab visually similar to that of a reference area spanning tabs 26 and 27 at $[32 \text{ nm}]^3$ and $[16 \text{ nm}]^3$ voxel sizes (i.e. using 4x, and 2x downsampled images, respectively), yielding a total of 20 CycleGAN models (no model was trained for tabs 26 and 27 at 32 nm and for tabs 23, 24, 26, and 27 at 16 nm). The reference area was chosen based on similarity to the region in which training data for segmentation models was located. The images in tabs 26 and 27 were sufficiently similar that no additional adjustment was required. The bounding boxes within the hemibrain volume used for training the CycleGAN models are specified in *Appendix 1—table 2*.

During training, a snapshot of network weights ('checkpoint') was saved every 30 min. CycleGAN inference was performed over a tab- and resolution-specific region of interest (ROI; see *Appendix 1—table 3*) with every saved checkpoint from the tab- and resolution-matched model. We then segmented the resulting volumetric images with a resolution-matched flood-filling network (FFN) model, and screened the segmentations for merge errors. Merge errors were identified by visually inspecting the largest objects (by the number of voxels) in the segmentations using a 3D mesh viewer (Neuroglancer). For every CycleGAN model, we selected checkpoints resulting in the minimum number of mergers, and then among these, selected the checkpoint corresponding to a segmentation with the maximum number of labeled voxels in objects containing at least 10,000 voxels.

We then performed CycleGAN inference with the selected checkpoint for every tab-resolution pair over the part of the aligned hemibrain volume corresponding to that tab. The stitched inference results were used as input volumes for tissue classification and neuron segmentation. CycleGAN normalization was not done at the native $[8 \text{ nm}]^3$/voxel resolution because there was insufficient evidence that the 8 nm FFN model could generalize well to different tabs.

**Appendix 1—table 2.** Bounding boxes within the hemibrain volume used for training CycleGAN models.

Coordinates and sizes are given for $[32 \text{ nm}]^3$ voxels. The same physical area of the hemibrain volume was used to train both 32 nm and 16 nm CycleGAN models.

| Tab | Start | | | Size | | |
|---|---|---|---|---|---|---|
| | X | Y | Z | X | Y | Z |
| reference | 4633 | 3792 | 2000 | 1374 | 2000 | 2000 |
| 22 | 8089 | 4030 | 1744 | 518 | 2000 | 2000 |
| 23 | 7435 | 3925 | 2101 | 654 | 2000 | 2000 |
| 24 | 6713 | 2939 | 4094 | 722 | 2000 | 2000 |
| 25 | 6017 | 2895 | 3635 | 694 | 2000 | 2000 |
| 28 | 3980 | 4944 | 3495 | 638 | 2000 | 2000 |
| 29 | 3307 | 2414 | 4094 | 666 | 2000 | 2000 |
| 30 | 2649 | 2519 | 4094 | 657 | 2000 | 2000 |
| 31 | 1979 | 2750 | 4094 | 670 | 2000 | 2000 |
| 32 | 1312 | 3065 | 4094 | 667 | 2000 | 2000 |
| 33 | 668 | 3101 | 3520 | 663 | 2000 | 2000 |
| 34 | 1 | 3112 | 3520 | 660 | 2000 | 2000 |

## Tissue classification

We manually labeled voxels in 4 tabs of the hemibrain volume as belonging to one of 7 classes: 'broken white tissue', trachea, cell bodies, glia, large dendrites, neuropil, or 'out of bounds'. We used these labels to train a 3D convolutional network that receives as input a field of view of $65 \times 65 \times 65$ voxels at $(16 \text{ nm})^3$/voxel resolution. The network uses 'valid' convolution padding and 'max' pooling operations with a kernel and striding shape of $2 \times 2 \times 2$, with convolution and pooling

operations interleaved in the following sequence: convolution with 64 features maps and a 3 × 3 × 3 kernel shape, max-pooling, convolution with 64 feature maps, max-pooling, convolution with 64 feature maps, max-pooling, convolution with 3 × 3 × 3 kernel size and 16 feature maps, convolution with 4 × 4 × 4 kernel shape 512 feature maps (i.e. fully connected layer), and finally a logistic layer output with eight units (the first unit was unused in the labeling scheme). The network was trained with data augmentation in which the order of the three spatial axes was randomly and uniformly permuted for each example during construction of the 16-example minibatch. For each example, the order of voxels along each spatial axis was also inverted at random with 50% probability. Examples from the seven classes were sampled randomly with equal probability. The model was implemented in TensorFlow and training was performed with asynchronous SGD on eight workers using NVIDIA P100 GPUs. The results can be viewed using NeuroGlancer at https://hemibrain-dot-neuroglancer-demo.appspot.com/#!gs://flyem-views/hemibrain/v1.0/mask-view.json.

The resulting classifier output was, on certain slices of the hemibrain, manually proofread using a custom tool (''Armitage''). The inference and proofreading process was then iterated seven times in order to expand and improve the set of ground truth voxels, resulting in a final ground truth set with the following number of examples in each class (sizes in Mxv, or megavoxels): 9.7 Mvx broken white tissue, 22.9 Mvx trachea, 42.1 Mvx cell bodies, 5.6 Mvx glia, 17.7M Mvx large dendrites, 71.4 Mvx neuropil, and 208.1 Mvx out of bounds.

## Mitochondria classification

We detected and classified mitochondria within the hemibrain volume using the same neural network architecture and training setup as that used for tissue classification. Ground truth data was collected through iterative annotation (two rounds) in Armitage, in which voxels within hemibrain were manually annotated as belonging to one of 4 classes: 'background' (33.7 Mvx), 'regular' (0.7 Mvx), 'special' (0.5 Mvx), and 'intermediate' (0.5 Mvx).

**Appendix 1—table 3.** ROIs within the hemibrain volume used for CycleGAN checkpoint selection.

| Tab | Voxel Res. [nm] | Start | | | Size | | |
|---|---|---|---|---|---|---|---|
| | | X | Y | Z | X | Y | Z |
| 22 | 32 | 8092 | 4392 | 5447 | 500 | 936 | 936 |
| 23 | 32 | 7435 | 2479 | 4979 | 500 | 936 | 936 |
| 24 | 32 | 6717 | 5414 | 4873 | 500 | 936 | 936 |
| 25 | 32 | 6010 | 3960 | 6235 | 500 | 936 | 936 |
| 28 | 32 | 3971 | 2591 | 2954 | 500 | 936 | 936 |
| 29 | 32 | 3471 | 4252 | 2224 | 500 | 936 | 936 |
| 30 | 32 | 2650 | 2995 | 4875 | 500 | 936 | 936 |
| 31 | 32 | 1982 | 3196 | 4875 | 500 | 936 | 936 |
| 32 | 32 | 1311 | 3141 | 4873 | 500 | 936 | 936 |
| 33 | 32 | 664 | 2850 | 4875 | 500 | 936 | 936 |
| 34 | 32 | 0 | 1900 | 4500 | 500 | 5000 | 2500 |
| 22 | 16 | 16080 | 8353 | 9871 | 1034 | 936 | 936 |
| 25 | 16 | 11900 | 12657 | 12636 | 1406 | 936 | 936 |
| 25 | 16 | 11900 | 5266 | 10578 | 1408 | 936 | 936 |
| 28 | 16 | 7900 | 9279 | 4613 | 1297 | 936 | 936 |
| 29 | 16 | 6550 | 8520 | 4613 | 1333 | 936 | 936 |
| 30 | 16 | 5250 | 7997 | 7510 | 1315 | 936 | 936 |
| 31 | 16 | 3860 | 7749 | 7510 | 1340 | 936 | 936 |
| 32 | 16 | 2550 | 9482 | 4225 | 1334 | 936 | 936 |
| 33 | 16 | 1280 | 7176 | 12265 | 1298 | 936 | 936 |
| 34 | 16 | 0 | 7587 | 12265 | 1328 | 936 | 936 |

## Automated neuron segmentation with FFNs

We trained three FFN models composed of the same architecture as detailed in previous work (*Januszewski et al., 2018*) for FIB-SEM volumes, targeted specifically for 8 nm, 16 nm, and 32 nm voxel resolution data. For the 8 nm model we used manually generated ground truth spread over six subvolumes ($520^3$ voxels each) located within the ellipsoid body, fan-shaped body and protocerebral bridge. The 16 nm and 32 nm models were trained with a proofread segmentation contained within a $8600 \times 3020 \times 9500$ voxel region spanning tabs 26 and 27. For the 32 nm model, training examples were sampled from objects comprising 5000 or more labeled voxels at 32 nm/voxel resolution. In total, 4.2 Gvx of labeled data were used for the 16 nm model and 423 Mvx for the 32 nm model.

We split the training examples into 'probability classes' similarly to *Januszewski et al., 2018*. Classes 13–17 were not sampled when training the 8 nm model in order to bias it toward small-diameter neurites. For 16 nm and 32 nm models fewer classes were used and the first class comprising all initial training examples with the fraction of voxels set to $0.95f_a<0.05$. Other than the changes regarding the probability classes, we followed the same procedures for training example sampling, seed list generation, field-of-view movement, and distributed inference as detailed previously (*Januszewski et al., 2018*).

FFN checkpoints were selected in a screening process. We generated tab 24 segmentations at 16 and 32 nm voxel resolution for every available checkpoint. We then screened these segmentations for merge errors, annotating every such error with two points, one in each distinct neurite. The segmentation generated with an FFN checkpoint that avoided the most errors was selected. For the 8 nm segmentation, we followed the same procedure but restricted to a $500^3$ subvolume within tab 24, located at 23284, 1540, 12080.

## Pipeline for segmentation of hemibrain with flood-filling networks

### Multi-resolution and oversegmentation consensus

We built the hemibrain segmentation with a coarse-to-fine variant of the FFN pipeline (*Januszewski et al., 2018*) combining partial segmentations generated at different resolutions. First, we used the 16 nm and 32 nm FFN models to segment the dataset at the corresponding resolution, with voxels identified by the tissue classifier as glia and out-of-bounds excluded from FFN FOV movement ('tissue masking'), and voxels classified as 'broken white tissue' excluded from seed generation. Voxels located within 128 nm from every hot knife plane were removed from the image data, and segmentation proceeded as if these regions did not exist. The resulting segmentation was extended back to the original coordinate system by nearest neighbor interpolation to fill the unsegmented spaces.

We then removed objects smaller than 10,000 voxels from the 32 nm segmentation (we will refer to the resulting segmentation as S32), isotropically upsampled it 2x, and combined it with the 16 nm segmentation using oversegmentation consensus (*Januszewski et al., 2018*). The resulting segmentation (S16) was used as the initial state for 8 nm FFN inference. In addition to tissue masking which was applied in the same way as in the case of lower resolution segmentations, we also masked areas within 32 voxels (at 8 nm/voxel resolution) from each hot-knife plane.

FlyEM proofreaders analyzed the roughly 200,000 largest objects in the segmentation, and manually split supervoxels identified as causing merge errors. This was done in three iterations – two targeting neuropil supervoxels, and one targeting cell bodies. The resulting corrected segmentation (S8) was used as the base segmentation for further work.

### Agglomeration

For agglomeration, we modified the scheme described in *Januszewski et al., 2018* for use with resolution-specific FFN models. First, we established a class for every segment by performing a majority vote of the tissue classification model predictions over the voxels covered by the segment. For every S16 segment (A, B), we also identified the maximally overlapping segment in S32 (denoted respectively $A_{max}$, $B_{max}$ below). For each of the S32, S16, and S8 segmentations, we then computed candidate object pairs and agglomeration scores, restricting object pairs to ones involving both segments

classified as either neuropil or 'large dendrite'. For S8, the object pairs were additionally restricted to those that included at least one object not present in S16.

For every evaluated segment pair (A, B) and the corresponding segments (A*, B*) generated during agglomeration, we computed the scores originally defined in *Januszewski et al., 2018* that is the recovered voxel fractions ($f_{AA}$, $f_{AB}$, $f_{BA}$, and $f_{BB}$, where $f_{AB}$ is the fraction of B found in A*, and so on), the Jaccard index JAB between A* and B*, and the number of voxels contained in A* or B* that had been 'deleted' (i.e., during inference their value in the predicted object mask fell from >0.8 to <0.5) during one of the runs (dA, dB).

We then used the following criteria to connect segments A and B. In S32, we connected segments that were scored as $(f_{**} \geq 0.6 \wedge J_{AB} \geq 0.4) \vee (f_{A*} \geq 0.8) \vee (f_{B*} \geq 0.8)$. In S16, we connected segments that either (a) were scored as $d_* \leq 0.02$ or were both classified as neuropil, and $f_{**} \geq 0.6 \wedge J_{AB} \geq 0.4$, or (b) were both classified as neuropil, $A_{max} = 0$ or $B_{max} = 0$ and $(f_{A*} \geq 0.9) \vee (f_{B*} \geq 0.9)$. In S8, we connected segments that were scored as $(d_A \leq 0.02 \vee d_B \leq 0.02) \wedge f_{**} \geq 0.6 \wedge J_{AB} \geq 0.8$.

Given the application of oversegmentation consensus in the process of building S16, objects created in S32 could have a different shape in S8. To compensate for this possibility, when agglomeration scores were being computed for S32 segments A and B, for each we computed up to eight maximally overlapping objects (A', B') in a downsampled version of S8 with matching voxel resolution, subject to a minimum overlap size of 1000 voxels and considered the agglomeration decision to apply to all combinations of A' and B'.

## Agglomeration constraints

From the procedure above, we used the agglomeration scores to organize segment connection decisions into priority groups and assign them a single numerical priority score (see *Appendix 1— table 4*). The decisions were then sorted in ascending order of the priority score, and sequentially processed, removing any decisions that would cause two cell bodies (as defined by manual annotations), or two segments previously separated manually in S8 proofreading to be connected was removed. Additionally, once all decisions with score <10 were processed, we also disallowed any remaining decisions that connected together any objects larger than 100 Mvx.

**Appendix 1—table 4.** Criteria for agglomerating priority groups.
If an agglomeration decision fulfills the criteria for multiple priority groups, it is assigned to the one with the lowest resulting score.

| Group | Segmentation | Criterion | Score |
|---|---|---|---|
| 1 | S32 | $(d_A \leq 0.02 \vee d_B \leq 0.02) \wedge (f_{**} \geq 0.6 \wedge J_{AB} \geq 0.8)$ | $1 - J_{AB}$ |
| 2 | S16 | $(d_A \leq 0.02 \vee d_B \leq 0.02) \wedge (f_{**} \geq 0.6$ $\wedge J_{AB} \geq 0.8) \wedge (A_{max} = 0 \vee B_{max} = 0) \wedge$ A and B are classified as neuropil | $2 - J_{AB}$ |
| 3 | S16 | $(d_A \leq 0.02 \vee d_B \leq 0.02) \wedge (f_{**} \geq 0.6$ $\wedge J_{AB} \geq 0.8) \wedge (A_{max} = 0 \vee B_{max} = 0)$ | $3 - J_{AB}$ |
| 4 | S16 | $(d_A \leq 0.02 \vee d_B \leq 0.02) \wedge (f_{**} \geq 0.6 \wedge J_{AB} \geq 0.8) \wedge$ A and B are classified as neuropil | $4 - J_{AB}$ |
| 5 | S16 | $(d_A \leq 0.02 \vee d_B \leq 0.02) \wedge (f_{**} \geq 0.6 \wedge J_{AB} \geq 0.8)$ | $5 - J_{AB}$ |
| 6 | S32 | $(f_{**} \geq 0.6 \wedge J_{AB} \geq 0.4) \wedge (A_{max} = 0 \vee B_{max} = 0) \wedge$ A and B are classified as neuropil | $6 - J_{AB}$ |
| 7 | S16 | $(f_{**} \geq 0.6 \wedge J_{AB} \geq 0.4) \wedge (A_{max} = 0 \vee B_{max} = 0)$ | $7 - J_{AB}$ |
| 8 | S16 | $(f_{**} \geq 0.6 \wedge J_{AB} \geq 0.4) \wedge$ A and B are classified as neuropil | $8 - J_{AB}$ |
| 9 | S16 | $(f_{**} \geq 0.6 \wedge J_{AB} \geq 0.4)$ | $9 - J_{AB}$ |
| 10 | S8 | None | $11 - J_{AB}$ |
| 11 | S32 | None | $12 - max(min(f_{AA}, f_{AB)}, min(f_{BA}, f_{BB}))$ |
| 12 | S16 | None | $13 - max(min(f_{AA}, f_{AB}), min(f_{BA}, f_{BB}))$ |

## Speculative agglomeration

Any body (or set of segments connected by the agglomeration graph) larger than 10 Mvx was considered to be an 'anchor' body. We connected smaller bodies to these anchor bodies in a greedy procedure to further reduce the total number of bodies in the agglomerated segmentation. We formed body pair scores using segment pair agglomeration scores as $max(min(f_{AA}, f_{AB}), min(f_{BA}, f_{BB}))$. We then merged every body with its highest scoring candidate partner, as long as this would not connect two anchor bodies, and the body pair score was >0.1. This procedure was repeated seven times.

## Synapse prediction

### Ground truth

For training and validation, we collected dense synapse annotations within small cubes, spread through different brain regions. In total, we collected 122 such cubes, using 25 for classifier training, and the remaining 97 for validation. At each cube location, proofreaders manually annotated all T-bars within a $400^3$ window, and further annotated all PSDs attached to T-bars within a smaller $256^3$ sub-window. In total, 7.6k T-bars were annotated, split between 1.8k for training and 5.8k for validation, and 11.7k PSDs were annotated, split between 3k for training and 8.7k for validation.

### Method

Details of the T-bar and PSD detection algorithms we used can be found in *Huang et al., 2018*. For reference, the T-bar classifier is a 3D CNN using a U-Net architecture (*Ronneberger et al., 2015*), with a receptive field size of $40^3$ voxels and 770 k parameters.

At inference, we leverage the tissue classification results mentioned above by discarding any predictions that fell outside of tissue categories of large dendrites or neuropil.

As mentioned in the main text, after collecting ground-truth throughout additional brain regions, we found that our initial T-bar classifier was giving lower than desired recall in certain areas. Therefore, we trained a new classifier, and combined the results in a cascade fashion, which we found gave better results than simply replacing the initial predictions. Specifically, we added any predictions above a given confidence threshold made by the new classifier for synapses that were not near an existing prediction, and removed any existing predictions that were far from predictions made by the new classifier at a second lower/conservative threshold.

One difficulty in placing a single T-bar annotation at each presynaptic location is a certain ambiguity with respect to 'multi T-bars', cases in which two distinct T-bar pedestals lay in close proximity, within the same neuron. Such a case can be difficult to distinguish from a single large synapse, both for manual annotators as well as the automated prediction algorithm. To make such a distinction reliably would require obtaining many training examples for both cases (multi T-bar versus single large synapse), and would only have a slight effect on the final weights of the connectome (but not the unweighted connectivity). Therefore, we make no attempt to predict multi T-bars, and instead as a final post-processing step, collapse to a single annotation any T-bar annotations that are in close proximity and in the same segmented body.

Finally, we observed that in certain brain regions, there are instances of T-bars in separate bodies but in close proximity to one another. These often form a 'convergent T-bars' motif, in which multiple T-bars closely situated in distinct bodies form a synapse onto the same PSD body. The proximity of such T-bars is often less than the distance threshold used in the non-maxima suppression (NMS) that is applied to generate the T-bar annotations from the pixel-wise U-Net predictions. Given the NMS, a number of these types of T-bars would be missed by our predictor.

To address this issue, we modified the post-processing of pixel-wise predictions so as to use a 'segmentation-aware NMS'. Specifically, we constrain the NMS applied to each pixel-wise local maxima to largely be limited to the specific segment in which the maxima occurs. Each segment is dilated slightly to avoid additional predictions that only fall a very small number of voxels outside the segment containing the maxima. (Note that unlike standard NMS, this procedure does require that the automated segmentation be available prior to inference.) We apply the segmentation-aware

NMS only in brain regions where convergent T-bars were observed, as occurs in the mushroom body and fan-shaped body.

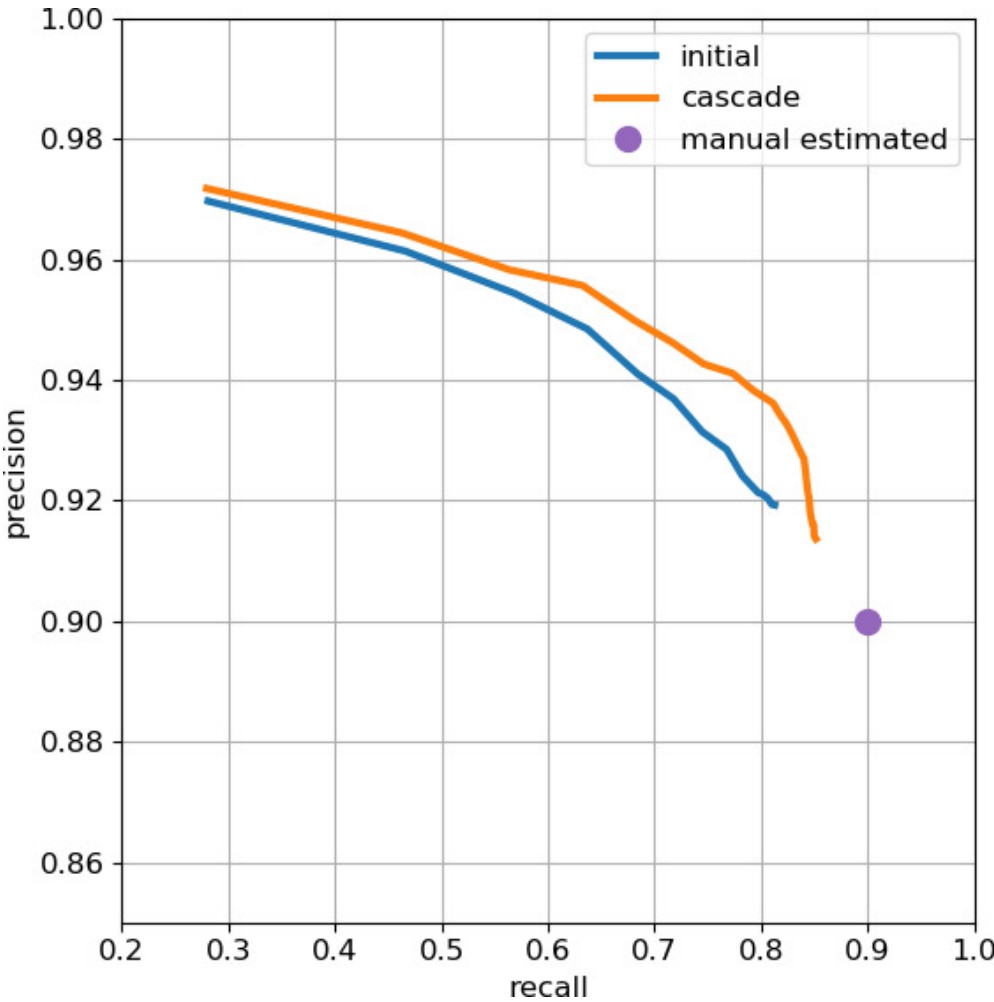

**Appendix 1—figure 1.** Precision-recall plot of T-bar prediction. The purple intercept indicates estimated manual agreement rate of 0.9. Data available in *Appendix 1—figure 1—source data 1*.

The online version of this article includes the following source data is available for figure 1:

**Appendix 1—figure 1—source data 1.** Data for *Appendix 1—figure 1*.

## Evaluation

*Appendix 1—figure 1* gives the precision-recall plot for T-bar prediction, averaged over all the available ground-truth validation cubes. As mentioned above, we do not attempt to predict multi T-bars; therefore, for the purposes of evaluation, we also collapse any ground-truth T-bars within close proximity in the same body to a single annotation. As can be seen from the figure, the cascade predictions are able to increase recall while maintaining precision. One of the primary error modes that leads to a difference between automated accuracy and manual agreement rate is the case of convergent T-bars, noted above. For instance, in *Figure 5* of the main text, the brain region with lowest recall is b'L in the mushroom body; closer analysis revealed many convergent T-bars in the annotated ground-truth cubes for b'L.

*Appendix 1—figure 2* below in the next subsection gives the corresponding precision-recall plot for end-to-end synapse prediction, averaged over all the available ground-truth validation cubes. As with both (*Huang et al., 2018*) and (*Buhmann et al., 2019*), we do not attempt to predict autapses, and remove any predicted connections that lie within the same neuron. For evaluation, any occasional ground-truth autapses are filtered out.

## Additional classifier

As an independent check on synapse quality, we also trained a separate classifier proposed by Buhmann (*Buhmann et al., 2019*), using the 'synful' software package provided. We additionally made several modifications to the code, including: adding an 'ignore' region around synapse blobs where predictions were not penalized, using focal loss (*Lin et al., 2017*) to help with class imbalance, using batch normalization (*Ioffe and Szegedy, 2015*) and residual layers (*He et al., 2016*), and adding explicit T-bar prediction as an additional network output. We found this multi-task learning (adding explicit T-bar prediction to PSD prediction and partner direction prediction) to be beneficial, similar to the use of cleft prediction in *Buhmann et al., 2019*, most likely due to the T-bar pedestals being a more reliable and prominent signal in our hemibrain preparation/staining than the PSDs. We refer to this network and its resulting synapse predictions as 'synful+'.

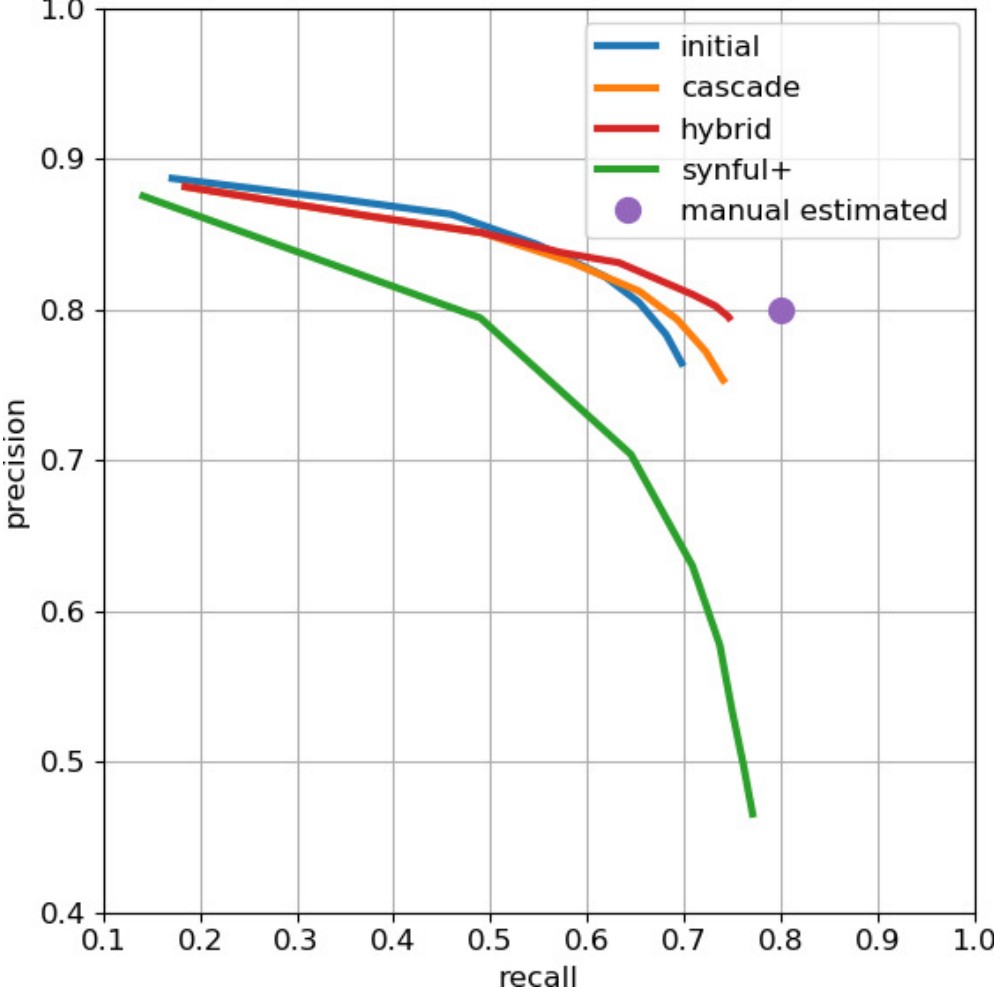

**Appendix 1—figure 2.** Precision-recall plot of end-to-end synapse prediction. The purple intercept indicates estimated manual agreement rate of 0.8. Data available in *Appendix 1—figure 2—source data 1*.

The online version of this article includes the following source data is available for figure 2:

**Appendix 1—figure 2—source data 1.** Data for *Appendix 1—figure 2*.

*Appendix 1—figure 2* shows the overall end-to-end precision-recall plots for each of the classifiers. As mentioned in the main text, we combined the predictions from the cascade and synful+ classifiers to yield a 'hybrid' classifier that achieved both better recall and precision than the two individual classifiers. Specifically, we modified the cascade predictions by (1) adding any PSDs that were predicted with strong confidence by synful+ and attached to existing T-bars, and (2) removing

any PSDs that were predicted with weak confidence by the cascade classifier and not predicted by synful+ even at a very low confidence threshold.

## Pathway analysis

Given two independent sets of synapse predictions (cascade and synful+), we further conduct an analysis of their respective connectivity graphs. We construct connectomes from each set of synapse predictions, limited to the 21,000+ traced bodies. At the level of individual synapses, the two sets of predictions have an agreement rate of about 80%.

However, we can look at connections of a given strength in one set of predictions, and see whether the other set of predictions gives a corresponding connection of any strength. For instance, among bodies that are connected with at least five synapses in the cascade predictions, less than 1% have no connection in the synful+ predictions, and similarly, among bodies that are connected with at least five synapses in the synful+ predictions, less than 2% have no connection in the cascade predictions. This suggests some level of stability in edges with a stronger connection, so that using a different classifier would be still likely to maintain that edge.

We also further manually assessed the small percentage of outlier edges. We sampled 100 synapses from the strongest of the edges in the cascade predictions that are not present in the synful+ predictions, and similarly 100 synapses from the synful+ predictions. For the cascade predictions, we find an overall accuracy of 64%, lower than the general accuracy of the cascade predictor, but we did not observe a pathway in which all sampled synapses were false positives. For the synful+ predictions, we found that all sampled synapses were false positives, resulting from improper placement of the T-bar annotation, thereby assigning the T-bar to an incorrect body. This suggests another use for such pathway analysis, in potentially discovering particular error modes of a classifier and allowing for re-training/refining to address such errors.

As a related measure of connectome stability, we also looked at how often the magnitude of the pathway connections were comparable. For instance, we can examine connections consisting of at least 10 synapses in one prediction set, and see how often those connections are within a factor of 2 in the other prediction set. We find that this holds for 93% of the connections of strength greater than 10. *Appendix 1—figure 3* shows a plot comparing pathway connection strength between the two sets of predictions.

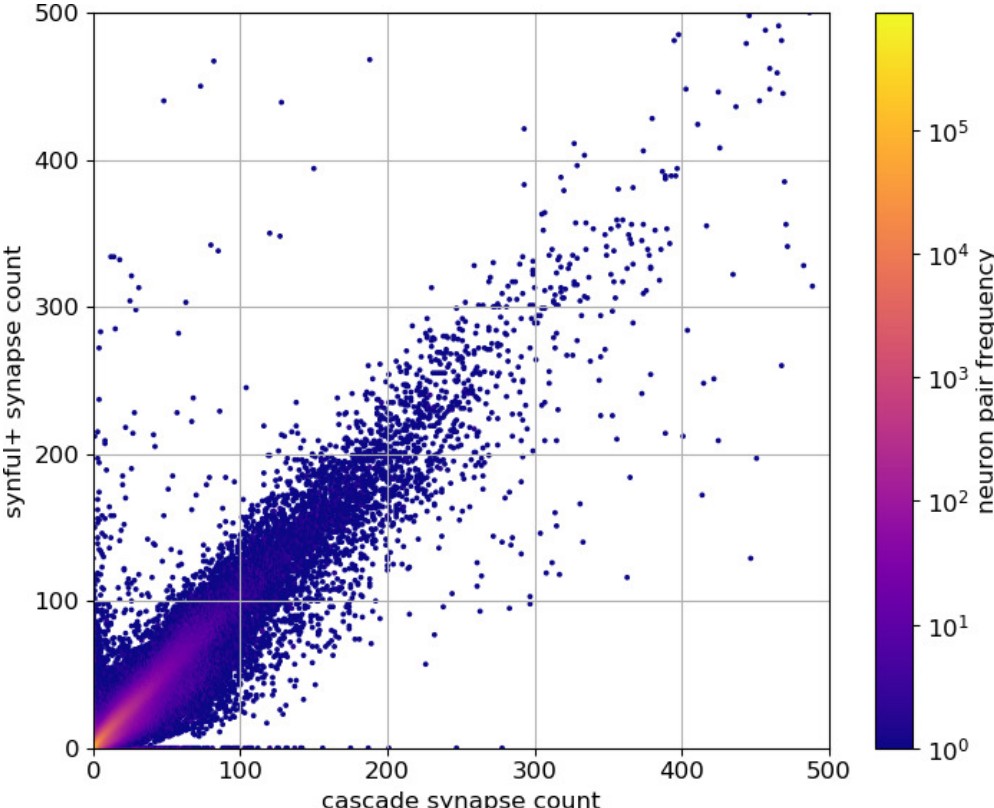

**Appendix 1—figure 3.** Comparison of synful+ connection strength versus cascade connection strength (truncated at a connection strength of 500 for clarity, omitting 40 edges from each prediction set). Data available in *Appendix 1—figure 3—source data 1*.

The online version of this article includes the following source data is available for figure 3:

**Appendix 1—figure 3—source data 1.** Data for *Appendix 1—figure 3*.

## Logistics and management

The hemibrain reconstruction required a large-scale effort involving several research labs, Janelia shared services, about ten staff scientists, and about 60 proofreaders. The overall initiative planning, including the choice of biological regions to image and reconstruct, timeline, and budget, was orchestrated by the FlyEM project team at the Janelia Research Campus with a guiding steering committee composed of several experts within the institute. The Connectomics Group at Google Research collaborated extensively with FlyEM developing key technology to segment the hemibrain volume.

Extensive orchestration by project staff and Janelia shared services was required to manage the team of proofreaders and the reconstruction effort. Our proofreading team consisted of full-time technicians hired specifically for proofreading. To satisfy the ambitious reconstruction goals of the hemibrain effort, we hired close to 30 people in a few months to augment the existing proofreading resources, requiring a streamlined system of recruitment and training. We found that the average proofreader required around 2 months of training to become reasonably proficient in EM tracing, which entailed working on carefully designed training modules and iterative feedback with more experienced proofreaders or managers. Ongoing training was necessary for both new and experienced proofreaders to meet the needs of different reconstruction tasks. The team of proofreaders had frequent meetings, and a Slack channel, with the software staff to improve proofreading software. We found that for a project of this size, several additional software personnel were required for data management, monitoring, orchestrating, and streamlining proofreading assignments.

The hemibrain reconstruction involved several different reconstruction steps or workflows, many discussed in the paper. The primary workflows were cleaving, false split review, focused proofreading, and orphan linking. Cleaving is the task of splitting a falsely merged segment. False split review

entails examining a neuron, using 3D morphology, for potential false splits. Focused proofreading is a 'merge' or 'don't merge' protocol based on automated suggestions from the segmentation algorithm. Orphan linking is fixing small detached segments that should either be annotated as exiting the hemibrain dataset, or be merged to a larger, already proofread body. Overall, we estimate that we undertook ≈ 50–100 proofreading years of reconstruction effort.

## Anatomical names in the central complex

*Appendix 1—table 5* provides two names for CX neurons: a short name useful for searching databases and an anatomical name that reveals morphological insight indicated by the input and output neuropils in the name. Previously published neurons (e.g. PB, NO, AB, EB ring neurons) now have short names, but their anatomical names are largely unchanged. Slight modifications were made to two neuron names to eliminate duplications with names in other brain regions or species: LN was changed to LNO1 and LGN to LGNO. In addition, the hyphens used in the abbreviated neuron names in *Wolff and Rubin, 2018* have been eliminated. The new anatomical names are limited to three neuropils: two input followed by an output brain region. Neurons that arborize in only two structures are named by the input followed by output neuropil. Two-letter abbreviations are used for the CX brain regions: PB, FB, EB, NO, and AB. All remaining neuropils follow the three-letter abbreviations established in *Ito et al., 2014*.

For fan-shaped body columnar and tangential cell anatomical names, numbers that follow 'FB' indicate layers, and layer numbers followed by a lower case 'd' or 'v' indicate the arbor is restricted to the dorsal or ventral half of the indicated layer (e.g. FB6d). Many FB arbors extend vertically to span more than one layer, a form that is indicated by sequential numbers separated by commas (e.g. FB2,3,4 indicates a single, vertical arbor that extends across layers 2, 3, and 4 of the FB). Some FB neurons have two distinct arbors in different layers; these are indicated by a gap in the layer numbers (e.g. FB2,6 has one arbor in layer 2 and a second in layer 6).

Inevitably, the large number of cell types and three-neuropil limit for anatomical names results in redundancy in neuron names. To give a unique identity to each name, '_#' is appended to otherwise indistinguishable names (e.g. LALCREFB2_1 and LALCREFB2_2).

The nomenclature system for the anatomical names was designed to enable visualization of a neuron's morphology. High synaptic density generally correlates with more prominent arbors in light level images and therefore provides a visual depiction of the neuron's overall shape. In most cases, synaptic density was therefore used as the primary metric in naming neurons. Synapse counts were retrieved from the neuPrint database. Only fully traced neurons from the right hemisphere were used in this analysis.

Different criteria were applied to FB tangential and columnar neurons when selecting neuropils to include in neuron names. The FB tangential neurons exhibit the greatest number of both presynaptic and postsynaptic terminals within a single FB layer, so naming these neurons based on neuropils with highest synaptic density would limit morphological insight. For example, FB layer 2 tangential neurons would be named FB2-neuropil X-FB2, FB2-neuropil Y-FB2, etc. Furthermore, since only a small number of neuropils are frequently arborized by the FB tangential neurons (e.g., SLP, SIP, SMP, CRE), names would be highly redundant. Instead, a more complete visual representation of a cell's morphology is achieved by using the brain regions with the second and third greatest number of input synapses (the first and second neuropils in the name). FB layer information is included in the third, or output, neuropil since presynaptic arbors are greatest in the FB in tangential neurons.

Occasional exceptions were made to include neuropils that would provide deeper insight into morphology. For example, according to the rules above, CRENO2FB4_2 (FB4M) should be named CRELALFB4 or CRESMPFB4 since 10–12% of input synapses are in the LAL and in the CRE, whereas only 6% are in NO2. An exception was made in this cell's name since arbors in the noduli distinguish this cell from other CRELALFB4 and CRESMPFB4 cells. Similarly, for neurons with equivalent synaptic density in several neuropils, those neuropils that best portrayed the neuron's unique morphology were chosen (e.g. SIPSCLFB2, or FB2H_a and FB2H_b).

**Appendix 1—table 5.** Corresponding short and anatomical names for cell types in the central complex.

These types were determined by different methods and different researchers, using different criteria.

| Short | Long | Short | Long | Short | Long | Short | Long |
|---|---|---|---|---|---|---|---|
| vDeltaA_a | AF | FB3B | EBCREFB3 | FB6C_a | SIPSMPFB6_1 | FC2B | FB1d,3,5,6CRE |
| vDeltaA_b | FB1D0FB8 | FB3C | LALSMPFB3 | FB6C_b | SIPSMPFB6_1 | FC2C | FB1d,3,6,7CRE |
| vDeltaB | FB1D0FB7_1 | FB3D | LALCREFB3 | FB6D | SMPFB6 | FC3 | FB2,3,5,6CRE |
| vDeltaC | FB1D0FB7_2 | FB3E | SMPLALFB3 | FB6E | SIPSMPFB6_2 | FR1 | FB2-5RUB |
| vDeltaD | FB1D0FB6 | FB4A | CRESMPFB4_1 | FB6F | SMPSIPFB6_3 | FR2 | FB2-4RUB |
| vDeltaE | FB1,2,3D0FB6v | FB4B | NO2LALFB4 | FB6G | SIPSMPFB6_3 | FS1A | FB2-6SMPSMP |
| vDeltaF | FB1,2,3D0FB5d | FB4C | CRENO2FB4_1 | FB6H | SMPSIPFB6_4 | FS1B | FB2,5, SMPSMP |
| vDeltaG | FB1,2D0FB5d | FB4D | CRESMPFB4_2 | FB6I | SMPSIPFB6_5 | FS2 | FB3,6SMP |
| vDeltaH | FB1,2D0FB5 | FB4E | CRELALFB4_1 | FB6J | FB6_1 | FS3 | FB1d,3,6,7SMP |
| vDeltaI | FB1D0FB5 | FB4F_a | CRELALFB4_2 | FB6K | SMPSIPFB6_6 | FS4A | FB3,8ABSMP |
| vDeltaJ | FB1D0FB5v | FB4F_b | CRELALFB4_2 | FB6L | FB6_2 | | FB1,3,8SMP |
| vDeltaK | FB1vD0FB4d5v | FB4G | CRELALFB4_3 | FB6M | WEDLALFB6 | FS4B | FB2,8ABSMP |
| vDeltaL | FB1vD0FB4 | FB4H | CRELALFB4_4 | FB6N | CRESMPFB6_1 | | FB1,2,8SMP |
| vDeltaM | FB1vD0FB4 | FB4I | LALCREFB4 | FB6O | SIPSMPFB6_4 | FS4C | FB2,6,7SMP |
| hDeltaA | FB4D5FB4 | FB4J | CRELALFB4_5 | FB6P | SMPCREFB6_1 | GLNO | LGNO |
| hDeltaB | FB3,4vD5FB3,4v | FB4K | CRESMPFB4_3 | FB6Q | SIPSMPFB6_5 | lbSpsP | lbSpsP |
| hDeltaC | FB2,6D7FB6 | FB4L | LALSIPFB4 | FB6R | SMPSIPFB6_7 | LCNOp | LCNp |
| hDeltaD | FB1,8D3FB8 | FB4M | CRENO2FB4_2 | FB6S | SIPSMPFB6_6 | LCNOpm | LCNpm |
| hDeltaE | FB1,7D3FB7 | FB4N | SMPCREFB4 | FB6T | SIPSMPFB6_7 | LNO1 | LNO1 |
| hDeltaF | FB1,6d,7D2FB6,7 | FB4O | CRESMPFB4d | FB6U | SMPCREFB6_2 | LNO2 | LNO2 |
| hDeltaG | FB2,3,5d6vD3FB6v | FB4P_a | CRESMPFB4_4 | FB6V | SMPCREFB6_3 | LNO3 | LNO3 |
| hDeltaH | FB2d,4D3FB5 | FB4P_b | CRESMPFB4_4 | FB6W | CRESMPFB6_2 | LNOa | LNa |
| hDeltaI | FB2,3,4,5D5FB4,5v | FB4Q_a | CRESMPFB4_5 | FB6X | SMPCREFB6_4 | LPsP | LPsP |
| hDeltaJ | FB1,2,3,4D5FB4,5 | FB4Q_b | CRESMPFB4_5 | FB6Y | SMPSIPFB6_8 | Delta7 | Delta7 |
| hDeltaK | EBFB3,4D5FB6 | FB4R | CREFB4 | FB6Z | SMPSIPFB6_9 | EL | EBGAs |
| hDeltaL | FB2,6D5FB6d | FB4X | CRESIPFB4,5 | FB7A | SIPSLPFB7 | EPG | EPG |
| hDeltaM | FB2,4D3FB5 | FB4Y | EBCREFB4,5 | FB7B | SMPSLPFB7 | EPGt | EPGt |
| FB1A | SMPSIPFB1,3 | FB4Z | FB4d5v | FB7C | SMPSIPFB7_1 | P1-9 | PBPB |
| FB1B | SMPSLPFB1d | FB5A | LALCREFB5 | FB7D | FB7,6 | P6-8P9 | P6-8P9 |
| FB1C | LALNOmFB1 | FB5AA | SMPCREFB5_10 | FB7E | SMPSIPFB7_2 | PEG | PEG |
| FB1D | SLPFB1d | FB5AB | SIPCREFB5d | FB7F | SMPSIPFB7_3 | PEN_a (PEN1) | PEN1 |
| FB1E_a | SIPSMPFB1d | FB5B | SMPSIPFB5d_1 | FB7G | SMPFB7,8 | PEN_b (PEN2) | PEN2 |
| FB1E_b | SLPSIPFB1d | FB5C | SMPCREFB5_1 | FB7H | SMPFB7 | PFGs | PFGs |
| FB1F | SMPSIPFB1d | FB5D | CRESMPFB5_1 | FB7I | SMPSIPFB7,6 | PFL1 | PFLC |
| FB1G | SMPSIPFB1d,3 | FB5E | CRESMPFB5_2 | FB7J | FB7,8 | PFL2 | PB1-4FB1,2,4,5LAL |
| FB1H | CRENO2,3FB1-4 | FB5F | SMPCREFB5_2 | FB7K | SLPSIPFB7 | PFL3 | PB1-7FB1,2,4,5LAL |
| FB1I | SMPSIPFB1d,7 | FB5G | SMPSIPFB5,6 | FB7L | SMPSIPFB7_4 | PFNa | PFNa |
| FB1J | SLPSIPFB1,7,8 | FB5H | CRESMPFB5_3 | FB7M | SMPSIPFB7_5 | PFNd | PFNd |
| FB2A | NOaLALFB2 | FB5I | SMPCREFB5_3 | FB8A | SLPSMPFB8_1 | PFNm_a | PFNm_a |
| FB2B_a | LALCREFB2_1 | FB5J | SMPFB5 | FB8B | PLPSLPFB8 | PFNm_b | PFNm_b |

*Continued on next page*

*Appendix 1—table 5 continued*

| Short | Long | Short | Long | Short | Long | Short | Long |
|---|---|---|---|---|---|---|---|
| FB2B_b | LALCREFB2_1 | FB5K | CREFB5 | FB8C | SMPFB8 | PFNp_a | PFNp_a |
| FB2C | SMPCREFB2_1 | FB5L | CRESMPFB5_4 | FB8D | SLPSMPFB8_2 | PFNp_b | PFNp_b |
| FB2D | LALCREFB2_2 | FB5M | CRESMPFB5_5 | FB8E | SMPSIPFB8_1 | PFNp_c | PFNp_c |
| FB2E | SCLSMPFB2 | FB5N | SMPCREFB5_4 | FB8F_a | SIPSLPFB8 | PFNp_d | PFNp_d |
| FB2F_a | SIPSMPFB2 | FB5O | SMPCREFB5_5 | FB8F_b | SIPSLPFB8 | PFNp_e | PFNp_e |
| FB2F_b | SIPSMPFB2 | FB5P | SMPCREFB5_6 | FB8G | SMPSIPFB8_2 | PFNv | PFNv |
| FB2F_c | SIPSMPFB2 | FB5Q | SMPCREFB5d | FB8H | SMPSLPFB8 | PFR_a | PFR_a |
| FB2G_a | SMPSIPFB2 | FB5R | FB5 | FB8I | SMPSIPFB8_3 | PFR_b | PFR_b |
| FB2G_b | SIPLALFB2 | FB5S | FB5d,6v | FB9A | SLPFB9_1 | SA1_a | SlpA |
| FB2H_a | SIPSCLFB2 | FB5T | CRESMPFB5_6 | FB9B_a | SLPFB9_2 | SA1_b | SlpA |
| FB2H_b | SIPSCLFB2 | FB5U | FB5d | FB9B_b | SLPFB9_2 | SA1_c | SlpA |
| FB2I_a | SMPATLFB2 | FB5V | CRELALFB5 | FB9B_c | SLPFB9_2 | SA2_a | SlpA |
| FB2I_b | SMPATLFB2 | FB5W | SMPCREFB5_7 | FB9B_d | SLPFB9_2 | SA2_b | SlpA |
| FB2J | SMPPLPFB2 | FB5X | SMPCREFB5_8 | FB9B_e | SLPFB9_2 | SA3 | SlpA |
| FB2K | LALSMPFB2 | FB5Y | SMPSIPFB5d_2 | FB9C_a | SLPFB9_2 | SAF | SlpAF |
| FB2L | SMPCREFB2_2 | FB5Z | SMPCREFB5_9 | FB9C_b | SLPFB9_2 | SpsP | SpsP |
| FB2M | SIPCREFB2 | FB6A | SMPSIPFB6_1 | FC1 | FB2CRE | | |
| FB3A | LALNO2FB3 | FB6B | SMPSIPFB6_2 | FC2A | FB1-5CRE | | |

In contrast to the FB tangential neurons, the FB columnar neurons project terminals to generally more than one layer of the FB. In addition, unlike the nine glomeruli in the PB, there is not a fixed number of vertical columns in the FB. Rather, column number is a function of cell type, so column number is an important feature of each cell type. Finally, a subset of the columnar neurons is intrinsic to the FB, whereas the remaining columnar neurons project terminals to additional neuropils. Nomenclature rules differ for these classes of neurons.

Intrinsic columnar FB neurons have multiple arbors in the FB. While most arbors comprise a mixture of dendrites and axons, one arbor type usually predominates. The predominantly input or output arbors are either vertically arranged within a single column of the FB, in which case they include the prefix 'v' in the short name and D0 in the anatomical name (see below), or horizontally distributed across different columns, in which case they include the prefix 'h' in the short name and D# in the anatomical name (see below for details). The horizontal class of neurons includes one or more input arbors vertically arranged within a single column that are separated by a given number of columns from an output arbor on the contralateral side of the FB. The distance between the input and output arbors, measured as the difference of column numbers, is unique to each cell type but is always half the width of the FB. The number of columns between the input and output arbors is referred to as 'delta' and is indicated in these neuron's names by a capital 'D' followed by the number of skipped columns between horizontally distributed input and output arbors. Two, three, five or seven columns have been documented to separate input from output arbors. As with the FB tangential neurons, input neuropils are indicated first in the neuron's name, followed by output neuropils. The total number of columns per brain for a given cell type equals (Delta + 1) x 2. For example, FB2,3,5d,6vD3FB6v (hDeltaG) has input arbors in FB layers 2, 3 and 1 that spans the dorsal layer 5 and ventral layer 6. A gap of 3 columns (D3) separates the input arbors from the output arbor, which is located in the ventral portion of layer 6. For this cell type, there are (3+1) x 2 or eight columns per brain. In some instances, output arbors were easier to count than input (dendritic) arbors, so column counts were based on output arbors. While columns for some cell types are unambiguous, in other cases, best guesses were made based on anatomy and connectivity. For cells with arbors that overlap, column number was defined by the minimal number of arbors (in other words, non-overlapping) that achieved full coverage of the cell's layer.

Neurons with input and output arbors that are vertically arranged within the same column have a displacement of zero columns, or a Delta0 (D0). This vertical alignment is reflected in the neuron's anatomical name by D0 and in the short name with the prefix 'v'. Column numbers were not calculated for these cell types.

The remaining columnar neurons exhibit both dendritic and axonal arbors within the FB as well as axonal arbors in additional neuropils. Although the vast majority of synapses are in the FB, it is the axonal synapses outside the FB that provide the best insight into gross morphology and are therefore indicated as the output neuropil. For example, while only 1% of the output for the FB2,5,6CRE (FC3) neuron is in the CRE, the neurite that projects to the CRE is distinctive and informs morphology. Column numbers were also not calculated for these cell types.

## Sparse to many motifs

The attached spreadsheet 'SparseToMany.xlsx' describes the sparse-to-many motifs illustrated in *Figure 22(B)*. Shown are all instances where sparse type (N ≤ 2) connects bidirectionally to at least 90% of all instances of an abundant type (N ≥ 20). The two sheets have identical data, but one is sorted by the name of the sparse type, and one the abundant type. The data contained is:

- Column A: The brain region and threshold used. All entries here are for the full brain and threshold 1.
- Column B: The name of the sparse type.
- Column C: The number of instances of the sparse type. This is most commonly 2, as most neurons are bilaterally symmetric. However there are cases where only a single instance was reconstructed in our volume.
- Column D: The bodyID of the sparse type.
- Column E: The instance name of the sparse type. This normally distinguishes the left and right examples.
- Column F: The name of the abundant type.
- Column G: The count of the abundant type.
- Column H: The number of connections from the sparse to the abundant type.
- Column I: The average strength of such connections.
- Column J: The number of connections from the abundant type to the sparse type.
- Column K: The average strength of such connections.

## Supplementary neuron type naming tables

**Appendix 1—table 6.** Naming scheme for neurons.
The neuron types that are known to exist but are not yet identified conclusively in the hemibrain data are not shown in the list.

| |
|---|
| **Connectivity types** |
| _a, _b, _c, _d, etc. at the end of the morphology type names shown below |
| **Morphology types** |
| **Central complex neuropil neurons** |
| Delta7 (protocerebral bridge Delta seven between glomeruli) |
| vDeltaA-M (fan-shaped body vertical Delta within a single column [type ID]) |
| hDeltaA-M (fan-shaped body horizontal Delta across columns [type ID]) |
| EL (Ellipsoid body - Lateral accessory lobe) |
| EPG (Ellipsoid body - Protocerebral bridge - Gall) |
| EPGt (Ellipsoid body - Protocerebral bridge - Gall tip) |
| ER1-6 (Ellipsoid body Ring neuron [type ID]) |
| ExR1-8 (Extrinsic Ring neuron [type ID]) |
| FB1A-9C (Fan-shaped Body [layer ID][type ID]) |

*Continued on next page*

FC1A-3 (Fan-shaped body - Crepine [type ID])

FR1, 2 (Fan-shaped body - Rubus [type ID])

FS1A-4C (Fan-shaped body - Superior medial protocerebrum [type ID])

IbSpsP (Inferior bridge - Superior posterior slope - Protocerebral bridge)

LCNOp, pm (Lateral accessory lobe - Crepine - NOduli [compartment ID])

LNOa (Lateral accessory lobe - NOduli [compartment ID])

LNO1-3 (Lateral accessory lobe - NOduli [type ID])

GLNO (Gall - Lateral accessory lobe - Noduli)

LPsP (Lateral accessory lobe - Posterior slope - Protocerebral bridge)

P1-9 (Protocerebral bridge [glomerulus ID])

P6-8P9 (Protocerebral bridge [glomerulus ID1] Protocerebral bridge [glomerulus ID2])

PEG (Protocerebral bridge - Ellipsoid body - Gall)

PEN_a(PEN1), _b(PEN2) (Protocerebral bridge - Ellipsoid body - Noduli [subtype ID])

PFGs (Protocerebral bridge - Fan-shaped body - Gall surrounding region)

PFL1-3 (Protocerebral bridge - Fan-shaped body - Lateral accessory lobe [type ID])

PFNa, d, m, p, v (Protocerebral bridge - Fan-shaped body - Noduli [compartment ID])

PFR (Protocerebral bridge - Fan-shaped body - Round body)

SA1-3 (Superior medial protocerebrum - Asymmetrical body [type ID])

SAF (Superior medial protocerebrum - Asymmetrical body - Fan-shaped body)

SpsP (Superior posterior slope - Protocerebral bridge)

**Mushroom body neuropil neurons**

KCab-c, m, p, s (Kenyon Cell alpha-beta lobe - [layer ID])

KCa'b'-ap1, ap2, m (Kenyon Cell alpha'-beta' lobe - [layer ID])

KCg-d, m, s, t (Kenyon Cell gamma lobe - [layer ID])

MBON01-35 (Mushroom Body Output Neuron [type ID])

APL (Anterior Paired Lateral)

DPM (Dorsal Paired Medial)

MB-C1 (Mushroom Body - Calyx [type ID])

PAM01-15 (MB-associated DAN, Protocerebral Anterior Medial cluster [type ID])

PPL101-106 (MB-associated DAN, Protocerebral Posterior Lateral 1 cluster [type ID])

**Dopaminergic neurons (DANs)**

PPL107, 08 (Protocerebral Posterior Lateral 1 cluster [type ID])

PPL201-04 (Protocerebral Posterior Lateral 2 cluster [type ID])

PPM1201-05 (Protocerebral Posterior Medial 1/2 clusters [type ID])

PAL01-03 (Protocerebral/paired Anterior Lateral cluster [type ID])

**Octopaminergic neurons**

OA-ASM1-3 (OctopAmine - Anterior Superior Medial [type ID])

OA-VPM3, 4 (OctopAmine - ventral paired median [type ID])

OA-VUMa1-7 (OctopAmine - ventral unpaired median anterior [type ID])

**Serotonergic (5HT) neurons**

5-HTPLP01 (5-HT Posterior lateral protocerebrum [type ID])

5-HTPMPD01 (Posterior medial protocerebrum, dorsal [type ID])

5-HTPMPV01, 03 (Posterior medial protocerebrum, ventral [type ID])

CSD (Serotonin-immunoreactive Deutocerebral neuron)

**Peptidergic and secretory neurons**

*Continued on next page*

AstA1 (Allatostatin A)

CRZ01, 02 (Corazonin [type ID])

DSKMP1A, 1B, 3 (Drosulfakinin medial protocerebrum [type ID])

NPFL1-l (Neuropeptide F lateral large)

NPFP1 (Neuropeptide F dorso median)

PI1-3 (Pars Intercerebralis [type ID] Insulin Producing Cell candidates)

SIFa (SIFamide)

**Circadian clock neurons**

DN1a (Dorsal Neuron 1 anterior)

DN1pA, B (Dorsal Neuron 1 posterior [type ID])

l-LNv (large Lateral Neuron ventral)

LNd (Lateral Neuron dorsal)

LPN (Lateral Posterior Neuron)

s-LNv (small Lateral Neuron ventral)

**Fruitless gene expressing neurons**

aDT4 (anterior DeuTocerebrum [type ID])

aIPg1-4 (anterior Inferior Protocerebrum [type ID])

aSP-f1-4, g1-3B (anterior Superior Protocerebrum [type ID])

aSP8, 10A-10C (anterior Superior Protocerebrum [type ID])

pC1a-e (doublesex-expressing posterior Cells [type ID])

oviDNa, b (Oviposition Descending Neuron [type ID])

oviIN (Oviposition Inhibitory Neuron)

SAG (Sex peptide Abdominal Ganglion)

vpoDN (vaginal plate opening descending neuron)

vpoEN (vaginal plate opening excitatory neuron)

**Visual projection neurons and intrinsic neurons of the optic lobe**

aMe1-26 (accessory Medulla [type ID])

CT1 (Complex neuropils Tangential [type ID])

LC4, 6, 9–46 (Lobula Columnar [type ID])

LLPC1-3 (Lobula - Lobula Plate Columnar [type ID])

LPC1, 2 (Lobula Plate Columnar [type ID])

LPLC1-4 (Lobula Plate - Lobula Columnar [type ID])

LT1, 11, 33–47, 51–87 (Lobula Tangential [type ID])

MC61-66 (Medulla Columnar [type ID])

DCH (Dorsal Centrifugal Horizontal)

H1, 2 (Horizontal [type ID])

HSN, E, S (Horizontal System North, Equatorial, South)

VS (Vertical System)

VCH (Ventral Centrifugal Horizontal)

Li11-20 (Lobula intrinsic [type ID])

HBeyelet (Hofbauer-Buchner eyelet)

**Descending neurons**

DNa01-10 (Descending Neuron cell body anterior dorsal [type ID])

DNb01-06 (Descending Neuron cell body anterior ventral [type ID])

DNd01 (Descending Neuron outside cell cluster on the anterior surface [type ID])

*Continued on next page*

DNg30 (Descending Neuron cell body in the gnathal ganglion [type ID])

DNp02-49 (Descending Neuron cell body on the posterior surface of the brain [type ID])

DNES1-3 (Descending Neuron going out to ESophagus [type ID])

Giant_Fiber descending neuron

MDN (Moonwalker Descending Neuron)

**Sensory associated neurons**

ORN_D, DA1-4, DC1-4, DL1-5, DM1-6, DP1l, m, V, VA1-7m, VC1-5, VL1-2p, VM1-7v (Olfactory Receptor Neuron_ [glomerulus ID])

TRN_VP1m, 2, 3 (Thermo-Receptor Neuron_ [glomerulus ID])

HRN_VP1d, 1 l, 4, 5 (Hygro-Receptor Neuron_ [glomerulus ID])

JO-ABC (Johnston's Organ auditory receptor neuron- [AMMC zone ID])

OCG01-08 (OCellar Ganglion neuron [type ID])

**Antennal lobe neuropil neurons**

D_adPN, DA1_lPN, DC2_adPN, DL3_lPN, DM4_vPN, DP1l_adPN, VA1d_adPN, VC2_lPN, VL2p_vPN, VM7d_adPN, VP2_l2PN, etc. (uniglomerular [glomerulus ID] _ [cell cluster ID] Projection Neuron)

VP1l+_lvPN, VP3+_vPN, etc. (uni+glomerular [glomerulus ID]+ _ [cell cluster ID] Projection Neuron, arborizing in a glomerulus and a few neighboring areas)

VP1m+VP2_lvPN1, 2, VP4+VL1_l2PN, etc. (biglomerular [glomerulus ID1]+[glomerulus ID2] _ [cell cluster ID] Projection Neuron, arborizing in two glomeruli)

M_smPNm1, 6t2, adPNm3-8, spPN4t9, 5t10, lPNm11A-13, l2PNm14-16, 3t17, 10t18, l19-22, m23, lvPNm24-48, lv2PN9t49, vPNml50-89, ilPNm90, 8t91, imPNl92 (Multiglomerular_ [cell cluster ID] Projection Neuron [antennal lobe tract ID][type ID])

MZ_lvPN, lv2PN (Multiglomerular and subesophageal Zone _ [cell cluster ID] Projection Neuron)

Z_lvPNm1, Z_vPNml1 (subesophageal Zone only _ [cell cluster ID] Projection Neuron [antennal lobe tract ID][type ID])

lLN1, 2, 7–17, v2LN2-5, 30–50, il3LN6, l2LN18-23, vLN24-29 ([cell cluster ID] Local Neuron [type ID])

mAL1-6, B1-5, C1-6, D1-4 (mediodorsal Antennal Lobe neuron [type ID])

AL-AST1 (Antennal Lobe - Antenno-Subesophageal Tract [type ID])

AL-MBDL1 (Antennal Lobe - Median BunDLe [type ID])

ALBN1 (Antennal Lobe Bilateral Neuron [type ID])

ALIN1-3 (Antennal Lobe INput neuron [type ID])

**Lateral horn neuropil neurons**

LHAD1a1-4a1 (Lateral Horn Anterior Dorsal cell cluster [cell cluster ID][anatomy group ID][type ID])

LHAV1a1-9a1 (Lateral Horn Anterior Ventral cell cluster [cell cluster ID][anatomy group ID][type ID])

LHPD1a1-5f1 (Lateral Horn Posterior Dorsal cell cluster [cell cluster ID][anatomy group ID][type ID])

LHPV1c1-12a1 (Lateral Horn Posterior Ventral cell cluster [cell cluster ID][anatomy group ID][type ID])

LHCENT1-14 (Lateral Horn CENTrifugal [type ID])

LHMB1 (Lateral Horn - Mushroom Body [type ID])

**Anterior optic tubercle neuropil neurons**

AOTU001-065 (Anterior Optic TUbercle [type ID])

TuBu01-10, A, B (anterior optic Tubercle - Bulb [type ID])

**Antler neuropil neurons**

ATL001-045 (Antler [type ID])

**Anterior ventrolateral protocerebrum neuropil neurons**

AVLP001-596 (Anterior VentroLateral Protocerebrum [type ID])

**Clamp neuropil neurons**

CL001-364 (CLamp [type ID])

**Crepine neuropil neurons**

*Continued on next page*

| | |
|---|---|
| CRE001-108 (CREpine [type ID]) | |

**Inferior bridge neuropil neurons**

| IB001-119 (Inferior Bridge [type ID]) |
|---|

**Lateral accessory lobe neuropil neurons**

| LAL001-204 (Lateral Accessory Lobe [type ID]) |
|---|

**Posterior lateral protocerebrum neurons**

| PLP001-255 (Posterior Lateral Protocerebrum [type ID]) |
|---|

**Posterior slope neuropil neurons**

| PS001-303 (Posterior Slope [type ID]) |
|---|

**Posterior ventrolateral protocerebrum neuropil neurons**

| PVLP001-151 (Posterior VentroLateral Protocerebrum [type ID]) |
|---|

**Saddle neuropil and antennal mechanosensory and motor center neurons**

| SAD001-095 (SADdle [type ID]) |
|---|
| AMMC-A1 (Antennal Mechanosensory and Motor Center- [type ID]) |

**Superior lateral protocerebrum neuropil neurons**

| SLP001-468 (Superior Lateral Protocerebrum [type ID]) |
|---|

**Superior intermediate protocerebrum neuropil neurons**

| SIP001-90 (Superior Intermediate Protocerebrum [type ID]) |
|---|

**Superior medial protocerebrum neuropil neurons**

| SMP001-604 (Superior Medial Protocerebrum [type ID]) |
|---|
| DGI (Dorsal Giant Interneuron) |

**Vest neuropil neurons**

| VES001-84 (VESt [type ID]) |
|---|

**Wedge neuropil neurons**

| WED001-183 (WEDge [type ID]) |
|---|
| WEDPN1-19 (WEDge Projection Neuron [type ID]) |

