## [Decision Letter]

**Acceptance summary:**

We consider this work to be a tour de force achievement on several fronts. Technologically, it ties together nearly a decade of advances in sample preparation, imaging, data management, and image analysis. It also is a very complete automated reconstruction of an EM volume that allows the authors to carefully begin the process of labeling subregions of the neuropil, derive cell types on the basis of both structure and connectivity, and identify circuit motifs, and is a demonstration of what connectomics has always promised to deliver: a reference atlas for biologists and a springboard for theoreticians and modelers working anywhere between the single-cell and whole network levels. We anticipate that this paper and its tools will facilitate the work from numerous laboratories around the world.

**Decision letter after peer review:**

Thank you for submitting your article "A Connectome and Analysis of the Adult *Drosophila* Central Brain" for consideration by *eLife*. Your article has been reviewed by two peer reviewers, and the evaluation has been overseen by a Reviewing Editor and Michael Eisen as the Senior Editor. The following individuals involved in review of your submission have agreed to reveal their identity: Jason Pipkin (Reviewer #1) and Chris Q Doe (Reviewer #2).

The reviewers have discussed the reviews with one another and the Reviewing Editor has drafted this decision to help you prepare a revised submission.

Summary:

This paper is viewed as a landmark contribution to the methodologies of EM connectomics and its use to characterize the *Drosophila* brain. The manuscript is extensive and well-illustrated, and the reviewers and editors are pleased to help to make this work available to the public. I am taking the unusual (for *eLife*) action to include the two reviewers in their entirety, as they include constructive comments that were intended by these two careful readers to make the paper more accessible and more useful for the community. I hope that you will take into consideration these comments, and make those editorial changes that will strengthen the paper. In particular, reviewer 2's major request for additional information seems critical for the paper to be maximally useful to the community,

Title: Reviewer 2 suggests a change in the title for your consideration.

Reviewer #1:

The work presented by Scheffer et al. here is a tour de force achievement on several fronts. Technologically, it ties together nearly a decade of advances in sample preparation, imaging, data management, and image analysis. Most impressively, this represents – to my knowledge – the densest and most complete automated reconstruction of an EM volume of this size. While at least one larger volume has been generated from the adult fly brain (Davi Bock's TEMCA work), it has not been segmented (yet) to the level of completion presented here. (Though I am curious to hear the authors' thoughts on to what extent the overall automated segmentation strategy used herein is truly dependent on the isotropic voxels or if a similar set of networks could be retrained on anisotropic data from other existing volumes. One can imagine the value in validating connectivity in another sample that's already been imaged.)

The completeness of the hemibrain connectome enables the authors to carefully begin the process of labeling subregions of the neuropil, derive cell types on the basis of both structure and connectivity, and identify circuit motifs. They also show that the segmented skeletons enable a first pass at building detailed neuronal models at the single-cell level. Therefore this work is not just the presentation of a volume of data (itself impressive) but also a demonstration of what connectomics has always promised to deliver: a reference atlas for biologists and a springboard for theoreticians and modelers working anywhere between the single-cell and whole network levels.

I have no major critiques of this manuscript. Some of the figures could be more striking – or at least not set to Matlab defaults in terms of colors and box ticks (Figures 17, 20, 21 and 25). Others are beautiful (Figures 8 and 10, e.g.).

Finally, I commend the authors for building out the online portal for others to interact with their data. This is an achievement on its own, and probably the most important one for yielding the greatest scientific returns from their efforts.

Reviewer #2:

This massive work describes new methods for generating EM data on large chunks of nervous system – 250 x 250 μm adult central brain – which includes all of one side of the bilateral brain plus all of the central brain midline structures such as the central complex. Thus, it has an n = 1 for most brain neurons. It excludes most of the optic lobe, and all of the ascending/descending neurons, SEZ and VNC. The paper contains comprehensive analyses of the data set, including motif structure, classifying cell types, and adjusting brain neuropil boundaries. The Neuprint software is elegant and intuitive.

Importantly, this data set and associated software provide a method to transition from a light level neuron morphology (e.g. from a FlyLight neuron to a Neuprint neuron). While this needs further development (see comment below), it has the potential to save years of experimental analysis to reach the same point.

This data set will be the gold standard until the full CNS reconstruction is finished in the future. The quality of the EM data are extremely high based on images shown and data in Neuroglancer. As mentioned above, this is a massive work in many regards.

My only required major comment is to expand the section "Matching EM and light microscopy data" as this is an extremely important advance, and perhaps one of the most useful aspects of the entire manuscript. I think the most useful improvement would be to give an example from beginning (FlyLight neuron) to end (matching neuron in Neuprint). This can be another figure, or perhaps better as a numbered text instructions with full URLs for each required step. Or a third option, provide an example workflow on a Janelia page and link to it here. As it stands, I was unable to perform this function with the available information in the paper.

---

## [Author Response]

Reviewer #1:The work presented by Scheffer et al. here is a tour de force achievement on several fronts. Technologically, it ties together nearly a decade of advances in sample preparation, imaging, data management, and image analysis. Most impressively, this represents – to my knowledge – the densest and most complete automated reconstruction of an EM volume of this size. While at least one larger volume has been generated from the adult fly brain (Davi Bock's TEMCA work), it has not been segmented (yet) to the level of completion presented here. (Though I am curious to hear the authors' thoughts on to what extent the overall automated segmentation strategy used herein is truly dependent on the isotropic voxels or if a similar set of networks could be retrained on anisotropic data from other existing volumes. One can imagine the value in validating connectivity in another sample that's already been imaged.)The completeness of the hemibrain connectome enables the authors to carefully begin the process of labeling subregions of the neuropil, derive cell types on the basis of both structure and connectivity, and identify circuit motifs. They also show that the segmented skeletons enable a first pass at building detailed neuronal models at the single-cell level. Therefore this work is not just the presentation of a volume of data (itself impressive) but also a demonstration of what connectomics has always promised to deliver: a reference atlas for biologists and a springboard for theoreticians and modelers working anywhere between the single-cell and whole network levels.I have no major critiques of this manuscript. Some of the figures could be more striking – or at least not set to Matlab defaults in terms of colors and box ticks (Figures 17, 20, 21 and 25). Others are beautiful (Figures 8 and 10, e.g.).

We had someone with stronger graphic artist skills work on these figures, and a few others. She unified the fonts and the colors, changed the backgrounds to be clearer, and substituted color-blind friendly colors for the originals. We hope these are more striking.

Finally, I commend the authors for building out the online portal for others to interact with their data. This is an achievement on its own, and probably the most important one for yielding the greatest scientific returns from their efforts.Reviewer #2:This massive work describes new methods for generating EM data on large chunks of nervous system – 250 x 250 μm adult central brain – which includes all of one side of the bilateral brain plus all of the central brain midline structures such as the central complex. Thus, it has an n = 1 for most brain neurons. It excludes most of the optic lobe, and all of the ascending/descending neurons, SEZ and VNC. The paper contains comprehensive analyses of the data set, including motif structure, classifying cell types, and adjusting brain neuropil boundaries. The Neuprint software is elegant and intuitive.Importantly, this data set and associated software provide a method to transition from a light level neuron morphology (e.g. from a FlyLight neuron to a Neuprint neuron). While this needs further development (see comment below), it has the potential to save years of experimental analysis to reach the same point.This data set will be the gold standard until the full CNS reconstruction is finished in the future. The quality of the EM data are extremely high based on images shown and data in Neuroglancer. As mentioned above, this is a massive work in many regards.My only required major comment is to expand the section "Matching EM and light microscopy data" as this is an extremely important advance, and perhaps one of the most useful aspects of the entire manuscript. I think the most useful improvement would be to give an example from beginning (FlyLight neuron) to end (matching neuron in Neuprint). This can be another figure, or perhaps better as a numbered text instructions with full URLs for each required step. Or a third option, provide an example workflow on a Janelia page and link to it here. As it stands, I was unable to perform this function with the available information in the paper.

We completely agree with this comment – this is one of the most useful things to do with the data, and it was not easy upon our initial data release. We have now addressed this – there is a new web application, https://neuronbridge.janelia.org, devoted explicitly to EM to light matching and vice versa. Furthermore there is now a button, shown as NB, on the tabular format for neurons in Neuprint. This neuron brings up NeuronBridge with the particular neuron pre-selected. There is also a demonstration video showing how to use this software, with examples.

There will be a separate paper on this process, with more examples and description of the algorithms. For example, when going from EM to light, it’s helpful if the software can create a search mask to help pull the specific neuron out of a not-so-sparse GAL4 line. When going from light to EM, the similarity function needs to know that only a portion of a brain-spanning neuron can be expected to match, and so on. Unfortunately, this paper is not out yet, even in bioRxiv form, so we cannot cite it. We are encouraging the authors to get this out as quickly as possible. Meanwhile we describe the process, and at least point out there is an upcoming paper.